# Hematopoietic stem and progenitor cell membrane-coated vesicles for bone marrow-targeted leukaemia drug delivery

Jinxin Li[1,2,3,9], Honghui Wu[4,5,6,9], Zebin Yu[1,2,3,9], Qiwei Wang [1,2,3,9], Xin Zeng[1,2,3,9], Wenchang Qian [1,2,3], Siqi Lu[1,2,3], Lingli Jiang[1,2,3], Jingyi Li[1,2,3], Meng Zhu[1,2,3], Yingli Han[1,2,3], Jianqing Gao [4,5,6,7,8] ✉ & Pengxu Qian [1,2,3] ✉

Leukemia is a kind of hematological malignancy originating from bone marrow, which provides essential signals for initiation, progression, and recurrence of leukemia. However, how to specifically deliver drugs to the bone marrow remains elusive. Here, we develop biomimetic vesicles by infusing hematopoietic stem and progenitor cell (HSPC) membrane with liposomes (HSPC liposomes), which migrate to the bone marrow of leukemic mice via hyaluronic acid-CD44 axis. Moreover, the biomimetic vesicles exhibit superior binding affinity to leukemia cells through intercellular cell adhesion molecule-1 (ICAM-1)/integrin β2 (ITGB2) interaction. Further experiments validate that the vesicles carrying chemotherapy drug cytarabine (Ara-C@HSPC-Lipo) markedly inhibit proliferation, induce apoptosis and differentiation of leukemia cells, and decrease number of leukemia stem cells. Mechanically, RNA-seq reveals that Ara-C@HSPC-Lipo treatment induces apoptosis and differentiation and inhibits the oncogenic pathways. Finally, we verify that HSPC liposomes are safe in mice. This study provides a method for targeting bone marrow and treating leukemia.

Leukemia is an aggressive hematopoietic malignancy that has a poor prognosis and a high mortality rate, which poses a serious health risk to people[1]. In AmL (acute myeloid leukemia), chemotherapy is still the first-line treatment, including the classical chemotherapeutic "3 + 7" regimen (7 days of cytarabine and 3 days of anthracycline or anthracenedione) and the recently reported DAV (doxorubicin, cytarabine and venetoclax) regimen, but these treatments still fail in certain patients[2]. Besides, chemotherapy cannot completely clear the residual cancer cells and has severe side effects. Hematopoietic stem cell transplantation (HSCT) during the window period when chemotherapy achieves remission is an important strategy for the treatment of leukemia. Nevertheless, HSCT usually encounters difficulties with human leukocyte antigen (HLA) matching and challenges with graft-versus-host disease (GVHD), and generally faces the risk of relapse after transplantation. In recent years, immunotherapies such as immune checkpoint inhibitors and chimeric antigen receptor T cell (CAR-T) therapy have shown great promise in the treatment of B cell-derived malignancies, but nearly half of the patients still do not

[1]Center for Stem Cell and Regenerative Medicine and Bone Marrow Transplantation Center of the First Affiliated Hospital, Zhejiang University School of Medicine, Hangzhou 310058, China. [2]Liangzhu Laboratory, Zhejiang University, 1369 West Wenyi Road, Hangzhou 311121, China. [3]Institute of Hematology, Zhejiang University & Zhejiang Engineering Laboratory for Stem Cell and Immunotherapy, Hangzhou 310058, China. [4]State Laboratory of Advanced Drug Delivery Systems of Zhejiang Province, College of Pharmaceutical Sciences, Zhejiang University, Hangzhou 310058, China. [5]Institute of Pharmaceutics, College of Pharmaceutical Sciences, Zhejiang University, Hangzhou 310058, PR China. [6]Jinhua Institute of Zhejiang University, Jinhua 321002 Zhejiang, PR China. [7]Department of Pharmacy, The Second Affiliated Hospital, Zhejiang University School of Medicine, Hangzhou 310009, PR China. [8]Zhejiang University Cancer Center, Zhejiang University, Hangzhou 310058, PR China. [9]These authors contributed equally: Jinxin Li, Honghui Wu, Zebin Yu, Qiwei Wang, Xin Zeng. ✉e-mail: gaojianqing@zju.edu.cn; axu@zju.edu.cn

respond to immunotherapy or even relapse after treatment[3]. Therefore, better therapeutic strategies need to be explored in leukemia treatment.

Nanoparticles have great advantages in the targeted delivery of anti-tumor drugs, particularly in the treatment of leukemia[4–6]. Liposomal drug delivery system is widely used in vaccine development and cancer treatment, but the unmodified liposomes have poor targeting ability and are easy to be cleared by rapid metabolism due to short internal circulation time. In recent years, cell membrane-coated biomimetic nanoparticles show good prospects in the targeted delivery to specific tissue or organ[7–11]. The cell membrane-coated nanoparticles exhibit good homing ability due to their own tissue-specific antigens or receptors, and have the advantages of lower immunogenicity and longer internal circulation time. Recent studies have shown that the combination of liposomes and mesenchymal stem cell (or neural stem cell) membranes to prepare engineered biomimetic vesicles can significantly improve the targeting ability and therapeutic efficacy of diseases[12,13].

In the pathogenesis, leukemia originates from the mutation and malignant clonal expansion of hematopoietic stem cells in the bone marrow[14,15]. Thus, bone marrow is the critical site where leukemic cells generate and accumulate[16]. According to the "two hit" theory of leukemogenesis, the genome of a pre-leukemic cell undergoes two hits, the first hit creating the genomic instability and the second hit introducing an oncogene mutation to transform into the leukemia stem cell. The leukemic cells first accumulate in the bone marrow, subsequently impair normal hematopoiesis, and eventually infiltrate into the blood circulation and invade other organs. In addition, the residual leukemia stem cells in the bone marrow after drug treatment are also important factors for relapse. Current chemotherapy drugs cannot sufficiently accumulate in bone marrow; therefore, targeted delivery of drugs to bone marrow is a promising strategy for the treatment of leukemia. It is well known that hematopoietic stem cells naturally exist in the bone marrow

microenvironment and have the inherent characteristics of bone marrow homing, which play an important role in maintaining normal hematopoiesis and immune system functions[17,18]. Therefore, we propose to prepare biomimetic vesicles by combining HSPC cell membrane with liposomes to deliver anti-leukemia drugs to bone marrow.

In this work, we prepare bone marrow targeted biomimetic vesicles by infusing HSPC cell membranes with liposomes. In leukemia-bearing mice, the nanoparticles home to and accumulate in the bone marrow. After loading the anti-leukemic chemotherapy drug cytarabine (Ara-C), the apoptosis and differentiation of leukemic cells are significantly increased, and the proportion of leukemic stem cells and the ability of colony formation are markedly reduced. In animal experiments, the survival of leukemic mice is significantly prolonged. Mechanistically, through analysis of surface proteins by mass spectrometry, the interactions between CD44-hyaluronic acid and ITGB2-ICAM-1 mediate the targeting of biomimetic vesicles to the bone marrow and leukemia cells, respectively. Finally, we verify that HSPC liposomes are safe in mice.

## Results

### Synthesis and characterization of HSPC-Lipo

The preparation process of the biomimetic vesicles was summarized in Fig. 1. Briefly, we isolated HSPCs (Lin⁻cKit⁺) by magnetic-activated cell sorting (MACS) from bone marrow of 6-8-week-old C57BL/6 mice, collected HSPC cell membranes by repeatedly freeze-thaw cycles, and fused with liposomes (HSPC-Lipo). We then carried out the characterization and detection of HSPC-Lipo vesicles. The transmission electron microscopy (TEM) showed the uniform cup-shaped morphology, with smooth surfaces of the biomimetic vesicles (Fig. 2a). The dynamic light scattering (DLS) measurement revealed that HSPC-Lipo had an average size of ≈178.8 nm (Fig. 2b), which was consistent with the results of TEM. Besides, HSPC-Lipo kept stable at different storage

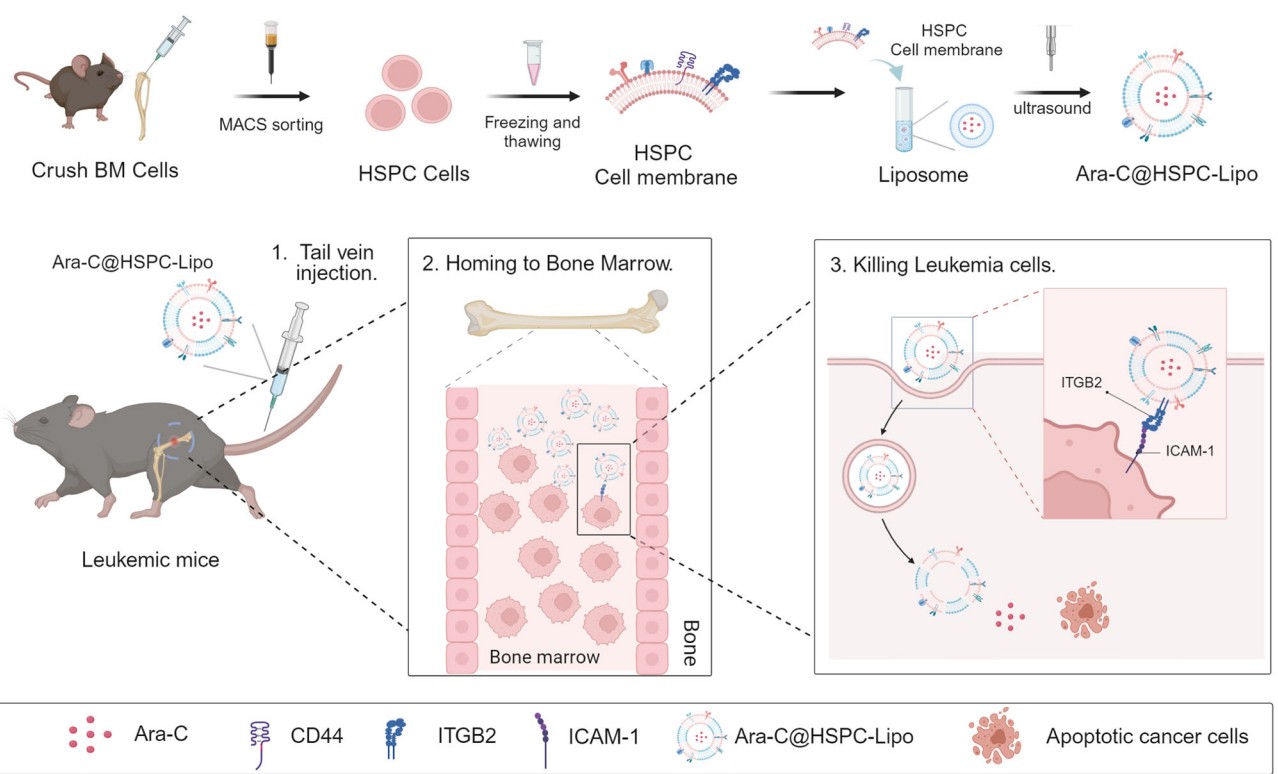

**Fig. 1 | Schematic of hematopoietic stem and progenitor cell membrane-coated vesicles (HSPC-Lipo) production and anti-leukemic treatment.** Schematic diagram of the preparation process for Ara-C@HSPC-Lipo. The drug loaded HSPC-Lipo biomimetic vesicles were injected into leukemic mice through the tail vein and targeted to the bone marrow. Ara-C@HSPC-Lipo targeting leukemia cells through receptor-ligand mediated specific binding and delivering chemotherapy drugs to kill leukemia cells in bone marrow.

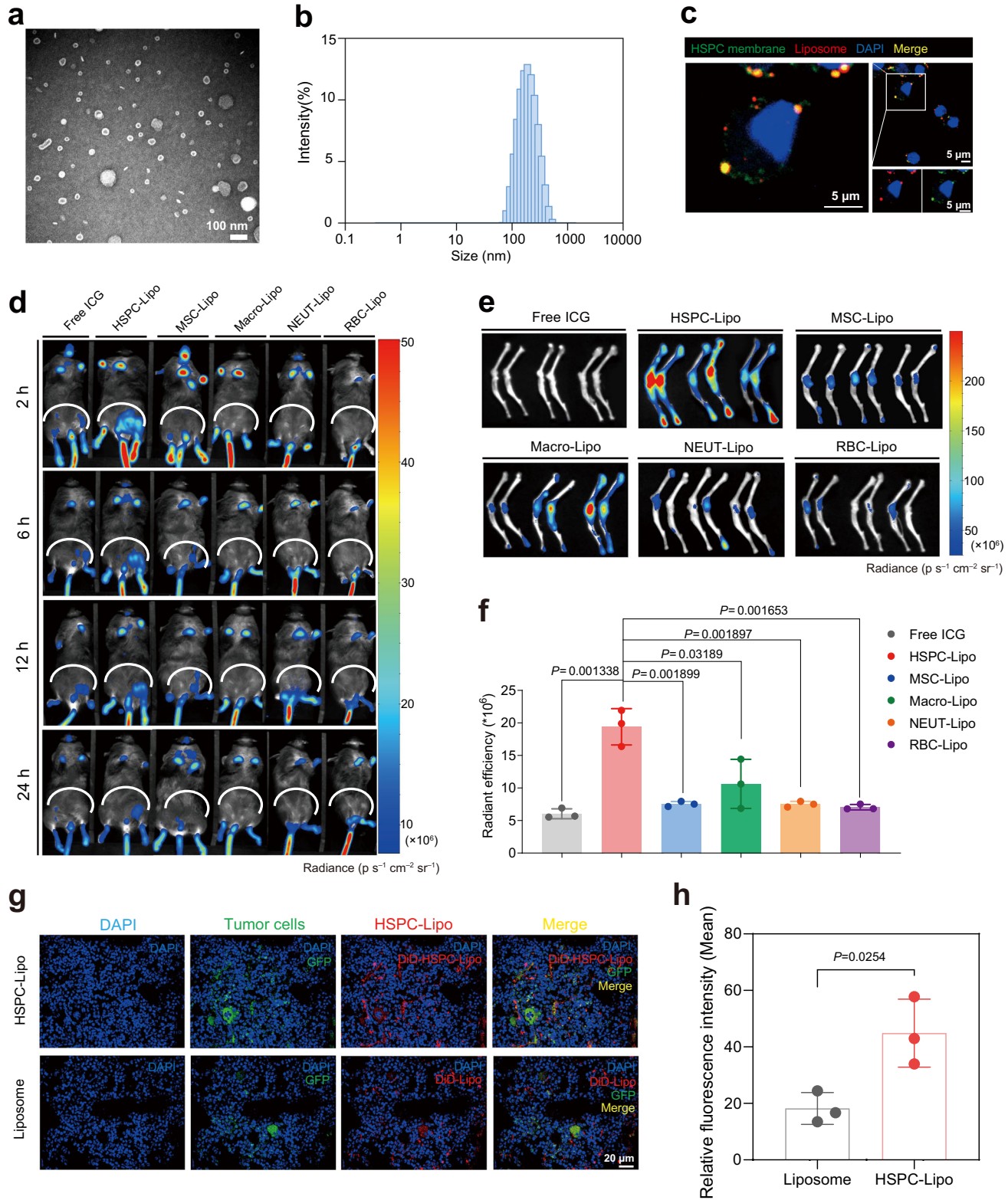

temperatures and medium conditions (Supplementary Fig. 1a). The zeta potential results indicated HSPC-Lipo had a negative surface charge (Supplementary Fig. 1b), which facilitated the long-term stability of vesicles (Supplementary Fig. 1c). Further, the fourier-transform infrared spectroscopy (FTIR) showed a similar spectrum to HSPC membrane, and the amide bond was detected at 1500 cm$^{-1}$ of wavelength (Supplementary Fig. 1d). Finally, to verify the fusion structure of

HSPC cell membrane and liposome, we carried out immunofluorescence staining, and found that liposome and HSPC cell membrane exhibited excellent co-localization (Fig. 2c). To avoid cross-staining of two components with hydrophobic dyes, we utilized anti-CD44 antibody and secondary antibodies (Cy3-labeled Goat Anti-Rabbit IgG (H + L)) to label the HSPC membrane (represented in green color), and used DiD dye to label the liposomes. The CD44 expressed

**Fig. 2 | Characterization and bone marrow targeting of HSPC-Lipo. a** The representative TEM images of HSPC-Lipo. Scale bar, 100 nm. **b** The average particle size distribution of HSPC-Lipo. Measured by Dynamic Light Scattering. **c** The representative immunofluorescence confocal images of co-localization of HSPC cell membrane and liposome. HSPC cell membrane was labeled with DiO (green), liposome was labeled with DiD (red). Scale bar, 5 μm. **d** Representative fluorescence images of ICG fluorescent labeled different leukocyte membrane-coated liposomes in leukemic mice (Ka539 model). Mice were subjected to in vivo imaging detection at different time intervals (2, 6, 12, 24 hours) after tail vein injection of different liposomes. Each mouse received approximately 20 μg of liposomes and 10 μg of cell membrane (indicated by total cell membrane protein weight). $n = 3$ mice.

**e** Representative fluorescence images of different leukocyte membrane-coated liposomes in the tibia and femur of mice. Mice were sacrificed at 24 h after tail vein injection of different liposomes, the tibia and femur were taken for in vivo imaging detection. $n = 3$ mice. **f** Quantitative analysis of different liposomes in mouse tibia and femur. Data were presented as mean ± s.d. ($n = 3$ mice). **g** Representative immunofluorescence images of HSPC-Lipo in mouse bone marrow. Scale bar, 20 μm. **h** Quantitative analysis of immunofluorescence results. The data were presented as mean ± s.d. ($n = 3$ mice). Statistical significance of $P$ values was calculated via a two-tailed, unpaired Student's t test and were indicated as *$P < 0.05$, **$P < 0.01$ and ***$P < 0.001$. Source data are provided as a Source Data file.

on the HSPC membrane but not on the liposomes, suggesting the specific labeling of HSPC membranes. Moreover, green and red fluorescence signals have achieved co-localization in HSPC-Lipo materials (Supplementary Fig. 2a) and when combined with leukemia cells (Supplementary Fig. 2b). Together, these data demonstrate that the HSPC cell membrane has been successfully embedded into the HSPC-Lipo.

## Bone marrow homing and biodistribution of HSPC-Lipo in vivo

We next investigated the bone marrow targeting ability of HSPC-Lipo in vivo. To demonstrate the advantages of using HSPC cell membranes, we compared the bone marrow targeting ability of the membranes derived from four blood or stromal cells with that from HSPCs. Specifically, we used four different sources of cell membranes, including mesenchymal stem cells, bone marrow-derived macrophages, neutrophils, and red blood cells, as control. After injection of approximately 20 μg of liposome and 10 μg of cell membrane (indicated by total cell membrane protein weight) into each mouse, the fluorescence imaging results showed that compared with the other four different sources of cell membranes, the carrier prepared by HSPC cell membrane exhibited the most significant bone marrow targeting ability (Fig. 2d, e). Besides, we found that the nanoparticles promptly (2 hours after tail vein injection) homed to the bone marrow of mice (Fig. 2d). Compared with free ICG probes or other ICG-labeled liposomes, ICG-labeled HSPC-Lipo exerted a higher targeted delivery efficacy to the bone marrow (Fig. 2e, f). We further evaluated the in vivo biodistribution of the vesicles by dissecting various organs of mice. Consistent with previous studies, the nanoparticles from all three groups were highly distributed in the liver and spleen after tail vein injection. However, compared with free probes, the HSPC-Lipo showed decreased distribution in spleen and liver, but increased delivery rate to the bone marrow in both normal and leukemic mice (Supplementary Fig. 3), indicating the specific bone marrow targeting property of HSPC-Lipo. Finally, we performed in vivo immunofluorescence staining and validated that HSPC-Lipo showed excellent homing and delivery efficacy to the bone marrow of leukemic mice (Fig. 2g, h). Collectively, these data suggest that HSPC-Lipo vesicles specifically home to and distribute in the bone marrow of leukemic mice.

## HSPC-Lipo facilitates targeting the bone marrow of leukemic mice via the CD44-hyaluronic acid interaction

To investigate the cellular and molecular mechanisms underlying the specific targeting of HSPC-Lipo nanoparticles to the bone marrow of leukemic mice, we performed mass spectrometry to map the proteomics of HSPC-Lipo nanoparticles and HSPC cell membrane (Fig. 3a and Supplementary Data 1). The classification of protein subtypes and proportion were similar between HSPC-Lipo and HSPC cell membrane (Fig. 3b and Supplementary Fig. 4a, b). Intriguingly, the pathway enrichment analysis showed that although HSPC-Lipo vesicles have been repeatedly freeze-thawed and fused with liposomes, they exhibited and enriched the similar genes and pathways as HSPC cell membranes, particularly the pathways related to cell adhesion (Fig. 3c, d). To explore the molecular mechanism of HSPC-Lipo

targeting bone marrow, we verified the classical adhesion molecules CD44 and CXCR4, locating on the cell membrane and HSPC-Lipo (Fig. 3e and Supplementary Fig. 4c). CD44 is a cell-surface glycoprotein on HSPCs and mediates their bone marrow homing behavior by interacting with its ligand hyaluronic acid (HA) in the bone marrow microenvironment[19,20]. Previous study has reported that the content of hyaluronic acid is upregulated in the bone marrow of leukemia patients compared to healthy control[21]. Accordingly, we detected the expression level of hyaluronic acid in the bone marrow of leukemic mice and found that the content of HA in bone marrow of leukemic mice was significantly higher than that of normal mice (Fig. 3f). The immunofluorescence staining further confirmed that HA co-localized with the HSPC-Lipo, whereas not the free liposomes (Fig. 3g). To further validate the specificity of CD44 mediated bone marrow targeting of HSPC-Lipo, we constructed CD44 knockdown by inducing HSPC with lentiviruses carrying CD44 shRNA, then collected HSPC cell membranes and conducted in vivo distribution experiments in mice. Firstly, we validated the efficiency of shRNA knockdown of CD44 through western blot (Supplementary Fig. 5a), and the results showed a significant decrease in CD44 expression. Further in vivo distribution results showed that compared with the control group, the CD44 knockdown group significantly reduced the bone marrow targeting ability of HSPC-Lipo (Fig. 3h, i). In summary, CD44 on HSPC-Lipo mediates the specific targeting to the bone marrow of leukemic mice by interacting with hyaluronic acid, which is highly distributed in the leukemic bone marrow niche.

## ITGB2 on the HSPC-Lipo directly binds to ICAM-1 and targets leukemia cells

We next sought to illustrate the specific targeting of HSPC-lipo to leukemia cells. Firstly, to demonstrate the specific advantage of HSPC cell membranes, we used four different leukocyte membrane-coated liposomes as controls. The results showed that compared to other cell membranes, the vesicles prepared by HSPC cell membranes had the highest targeting ability to leukemia cells (Fig. 3j). In contrast, the binding ability to mouse progenitor cell line (32D) was significantly lower than that of leukemia cells (Fig. 3j). To verify the mechanism by which HSPC-Lipo targets leukemia cells, we further analyzed the mass spectrometry data and found that the adhesion-related proteins were present in both the cell membrane and HSPC-Lipo (Fig. 3c, d and Supplementary Fig. 4a, b). We further performed KEGG pathway enrichment analysis among different groups and found that adhesion-related pathways, such as focal adhesion, were both found in HSPC-Lipo and co-enriched groups (Supplementary Fig. 4d–f; Supplementary Tables 1–3). By further analyzing the proteins enriched by the cell membrane and HSPC-Lipo vesicles, we identified ITGB2 (CD18), the classic proteins involved in cell adhesion. We further verified the mass spectrometry data by performing western blot to demonstrate that the adhesion molecules ITGB2 was expressed on the cell membrane and HSPC-Lipo (Fig. 3e). ITGB2 is a classical adhesion ligand, and its receptor is ICAM-1 (CD54), which is highly expressed in leukemia cells[22]. Thus, we detected the expression of ICAM-1 on the surface of leukemic cells and confirmed that ICAM-1 was highly expressed on the

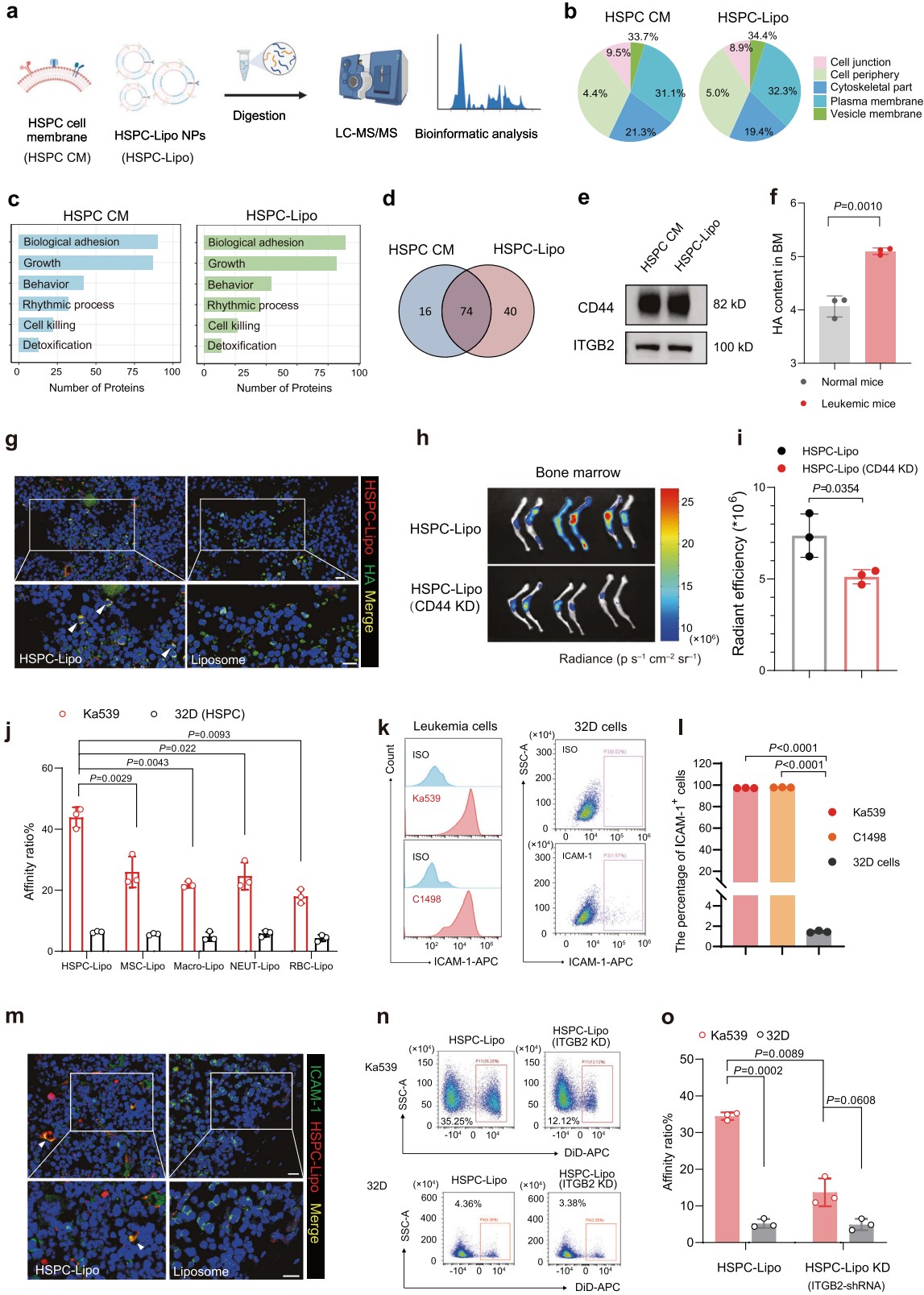

both lymphocytic and myelocytic leukemia cells (Fig. 3k, l and Supplementary Fig. 6a-f). We also validated the expression level of ICAM-1 using data from the online database of blood diseases (Supplementary Fig. 6g). HSPC-lipo showed stronger binding ability with leukemic cells compared with the control lineage cells in BM (Supplementary Fig. 6h, i). Finally, the immunofluorescence staining showed that the HSPC-lipo nanoparticles and ICAM-1 were co-localized (Fig. 3m). In addition, we also constructed ITGB2 knockdown HSPC cells and the

knockdown efficiency of ITGB2 was shown in Supplementary Fig. 5b. Compared with the control group, the carrier prepared using ITGB2 knockdown HSPC cell membranes significantly reduced its affinity with leukemic cells (Fig. 3n, o). HSPC-Lipo increased affinity of liposomes for leukemia (Ka539) cell line ~2.7-fold in vitro (from a baseline of ~13% when ITGB2 knocked down, up to ~35%) (Fig. 3o). In brief, these results suggest that HSPC-Lipo binds to leukemic cells through the ICAM-1-ITGB2 interaction.

**Fig. 3 | HSPC-Lipo exhibits higher affinity to leukemic cells. a** Schematic illustration of Liquid chromatography tandem mass spectrometry (LC-MS/MS) sequencing design of HSPC cell membrane and HSPC-Lipo. The HSPCs cells were isolated from C57BL/6 mice. **b** Protein type analysis diagram. **c** Protein pathway enrichment analysis. **d** Proteins that coexist in HSPC cell membrane and HSPC-Lipo. **e** Western blot analysis of CD44 and ITGB2. **f** Determination of hyaluronic acid content in mouse bone marrow. Data were presented as mean ± s.d. (*n* = 3 mice). **g** Representative immunofluorescence images of co-localization of hyaluronic acid and HSPC-Lipo. Scale bar, 10 μm. **h** Representative fluorescence images of CD44 knockdown HSPC-Lipo at 24 hours after tail vein injection (cell membrane derived from progenitor cell line 32D cells). (*n* = 3 mice). **i** Quantitative analysis of fluorescence images of CD44 knockdown HSPC-Lipo. Data were presented as mean ± s.d. (*n* = 3 mice). **j** Flow cytometry detection of leukemic cell targeting ability of different leukocyte membrane-coated liposomes. Leukemia cells were harvested for

detection at 0.5 hours after incubation with different liposomes. *n* = 3 experimental replicates. **k** Representative flow cytometry histograms and cytometry plots of ICAM-1 expression on mouse leukemia cells and mouse progenitor cell line (32D). **l** Quantitative analysis of ICAM-1 expression. *n* = 3 experimental replicates. **m** Representative immunofluorescence images of co-localization of ICAM-1 and HSPC-Lipo. Scale bar, 10 μm. **n** Representative flow cytometry plots of the leukemic cells targeting ability of ITGB2 knockdown HSPC-Lipo (cell membrane derived from mouse progenitor cell line 32D). Leukemic cells were harvested for detection 0.5 hours after incubation with different liposomes. **o** Quantitative analysis of the leukemic cells targeting ability of ITGB2 knockdown HSPC-Lipo. *n* = 3 experimental replicates. The data were presented as mean ± s.d. Statistical significance of *P* values was calculated via a two-tailed, unpaired Student's t test and were indicated as * *P* < 0.05, ** *P* < 0.01 and *** *P* < 0.001. Source data are provided as a Source Data file.

## Drug loading and in vitro anti-leukemia effect of HSPC-Lipo vesicles

Next, we loaded the HSPC-Lipo vesicles with chemotherapy drug Ara-C, and measured the drug loading efficiency of Ara-C@HSPC-Lipo and its anti-leukemia effect in vitro. We measured the drug loading efficiency and drug release kinetics of HSPC-Lipo at 48 h and 72 h, and found that the drug release ratio of HSPC-Lipo was significantly lower than that of other control groups, showing sustained drug release capacity (Fig. 4a, Supplementary Fig. 7a, b and Supplementary Table 4). Next, we verified the anti-leukemia effect of Ara-C@HSPC-Lipo in vitro. After Ara-C@HSPC-Lipo treatment at 48 h, the cell viability of leukemic cells was significantly decreased (Fig. 4b). Correspondingly, we tested the apoptosis of leukemic cells after 48 hours of different treatments, and the results indicate that the frequency of Annexin V positive cells was significantly increased compared with the control group (Fig. 4c, d). Furthermore, we measured the frequency of leukemia stem cells after different treatments in C1498 cells. And the results showed that the frequency of LSCs was significantly decreased after the treatment of Ara-C@HSPC-Lipo at 48 (Fig. 4e, f) and 72 hours (Supplementary Fig. 7e, f) respectively. Moreover, the frequency of mature myeloid cells after Ara-C@HSPC-Lipo treatments was significantly increased in C1498 cells both at 48 (Fig. 4g, h) and 72 hours (Supplementary Fig. 7c, d). Similarly, in Ka539 cells, previous literature has reported B220 as a marker for leukemia stem cells and used it as a target for nanomedicine[23,24]. Our results showed a significant decrease in the frequency of B220 high population leukemia stem cells after treatment with Ara-C@HSPC-Lipo (Supplementary Fig. 7g, h). Collectively, these above results prove that the Ara-C@HSPC-Lipo shows excellent anti-leukemia effect in vitro.

## Anti-leukemia effect of Ara-C@HSPC-Lipo in vivo

Next, we verified the anti-leukemia effect of Ara-C@HSPC-Lipo in two different leukemia mouse models. We utilized a wildly-used and more naturally-forming mLL-AF9 leukemia mouse model to validate the therapeutic effect in vivo. The animal experiment setup was shown in Fig. 5a. The results showed that the survival period of leukemic mice treated with Ara-C@HSPC-Lipo was significantly prolonged compared with the other control groups (Fig. 5b). After Ara-C@HSPC-Lipo treatment, the frequency of leukemic cells (Fig. 5c, d) and leukemia stem cells (Fig. 5e, f) were both markedly reduced compared with the other control groups in the bone marrow. Similarly, the frequency of leukemic cells in spleen (Fig. 5g, h) and peripheral blood (Fig. 5i) also significantly decreased. The above results suggest the addition of HSPC membrane to Ara-C loading liposomes potentially promoted a - 5-fold greater targeting specificity for LSC in the BM, and approximately 1.4-fold greater specificity in peripheral blood for leukemic cells. Correspondingly, the weight of spleens was remarkably reduced in mice treated with Ara-C@HSPC-Lipo (Fig. 5j, k). Further, we have performed the H&E staining on the sections of mouse bone marrow in mLL-AF9 AML model after treatment with different drugs, and found

that the group of Ara-C@HSPC-Lipo showed the strongest inhibitory effect on leukemogenesis (Supplementary Fig. 8). In addition, we also investigated the anti-leukemia effect of Ara-C@HSPC-Lipo in a B-ALL leukemia mouse model (Fig. 6a). After drug treatment, the frequency of leukemic cells in the bone marrow was significantly reduced in Ara-C@HSPC-Lipo group compared to other control groups (Fig. 6b, c). Correspondingly, the frequency of leukemic cells in the spleen (Fig. 6d) and the weight of the spleen (Fig. 6e) were significantly reduced. The survival curve analysis also showed that the survival of leukemic mice in Ara-C@HSPC-Lipo group was prolonged compared with other different treatment groups (Fig. 6f). In addition, our results demonstrate that Ara-C@HSPC-Lipo also exhibits good therapeutic effects when using low-dose free chemotherapy drugs as control treatment (Supplementary Fig. 9 and Fig. 10). To sum up, the above results show that the Ara-C@HSPC-Lipo shows good anti-leukemia effect in vivo.

## Transcriptome analysis reveals the signaling pathways after Ara-C@HSPC-Lipo treatment

To demonstrate the mechanisms of Ara-C@HSPC-Lipo in killing leukemia cells, we performed bulk-cell RNA sequencing in Ka539 and C1498 leukemia cells after treatment with Ara-C-loaded HSPC-Lipo or free liposomes. The results of Principal Components Analysis of different samples were shown in Supplementary Fig. 11a, b. After Ara-C@HSPC-Lipo treatment, the volcano plot of differentially expressed genes and clustering results were shown in the Fig. 7a–d. After treatment with Ara-C@HSPC-Lipo, expression levels of oncogenes such as Idh2 and Slc6a2 were significantly downregulated (Fig. 7a), while levels of tumor suppressor genes such as Ptpn2 and Runx3 were significantly upregulated (Fig. 7c). KEGG and GO enrichment analysis showed that pathways related to apoptosis, ferroptosis and P53 signaling pathway were markedly upregulated, whereas pathways involved in oncogenesis, such as PI3K-Akt, cell adhesion, oxidative phosphorylation and Rap1 signaling pathways, were significantly down-regulated in Ara-C@HSPC-Lipo treated group related to control group (Fig. 7e–h and Supplementary Fig. 11c, d). Particularly, KEGG pathway analyzes revealed that cell differentiation, apoptosis, negative regulation of cell cycle and negative regulation of TOR signaling were enriched in Ara-C@HSPC-Lipo treated group (Fig. 7i), which was in line with our flow cytometry results (Fig. 4c–h). Overall, these data indicate that Ara-C@HSPC-Lipo treatment induces apoptosis and differentiation and reduces cell cycle related and other oncogenic pathways.

## Biosafety evaluation of HSPC-Lipo in vivo

Finally, we evaluated the biological safety of HSPC-Lipo in BALB/c mice (Fig. 8a). Compared with the control group, the body weight of mice injected with HSPC-Lipo was unchanged (Fig. 8b). We also tested the peripheral blood hemogram of mice, and found that HSPC-Lipo had no significant toxicity on all the parameters of peripheral blood hemogram

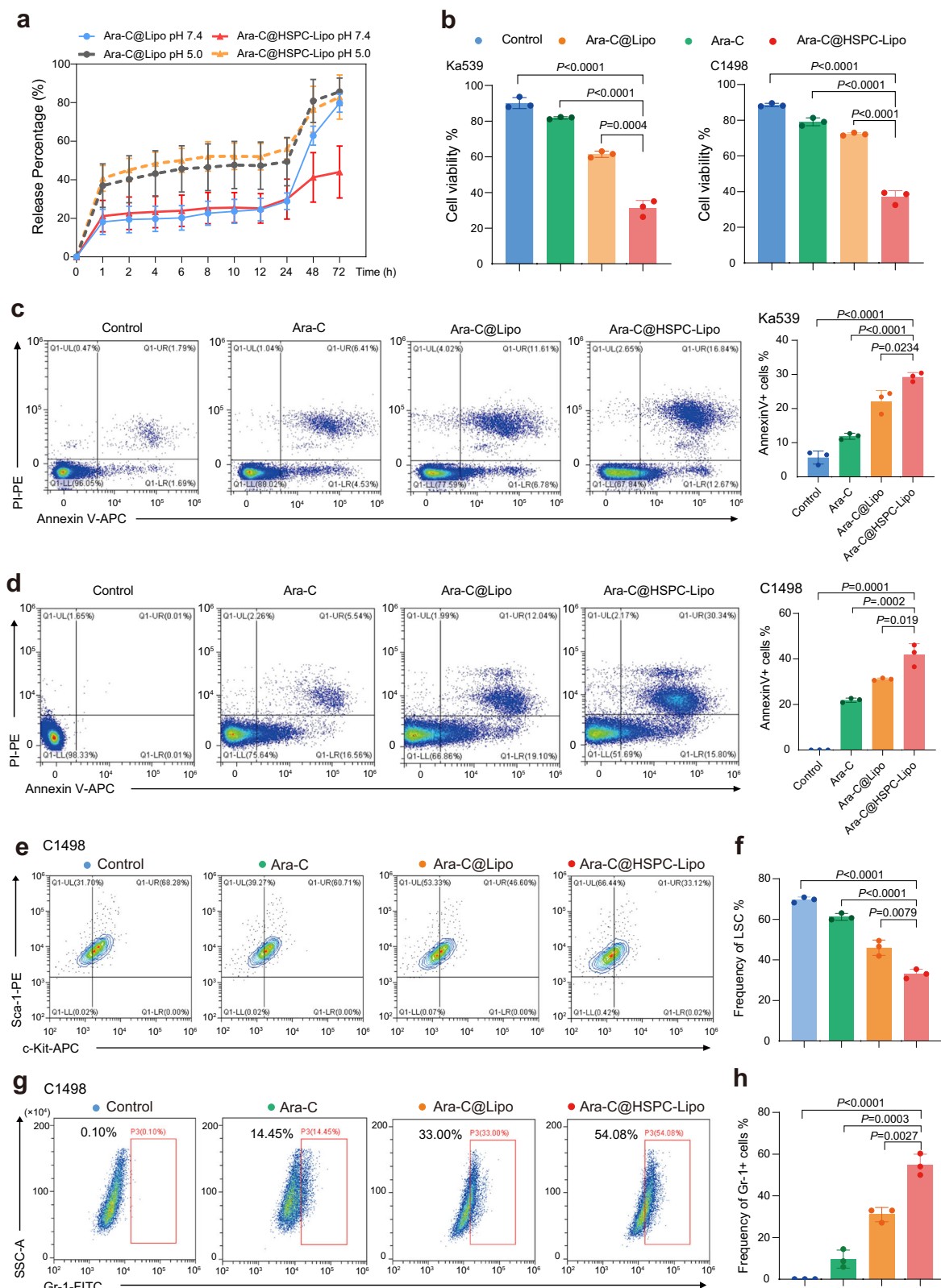

of mice (Supplementary Fig. 12). Further, we examined the proportions of hematopoietic stem cells, progenitor cells, and lineage cells in the bone marrow and found no significant difference between the HSPC-Lipo treatment and control groups either in the frequency of total mononuclear cells (Fig. 8c–e) or the absolute number (Fig. 8f–h). Finally, the H&E staining results of organs also proved the biosafety of HSPC-Lipo since no significant pathological damage was observed in the main

visceral organs of both male and female mice (Supplementary Fig. 13). Taken together, these results show that the HSPC liposomes are safe in mice.

## Discussion

Bone marrow targeting delivery drugs is a promising strategy in the treatment of leukemia[25,26]. Hong et al. reported a leukemia cell

**Fig. 4 | Drug loading and in vitro toxicity against leukemia cells. a** Drug release kinetics of HSPC-Lipo. Data were presented as mean ± s.d. ($n = 3$ experimental replicates). **b** Cell viability assay of leukemic cells after different treatments at 48 hours in vitro. Data were presented as mean ± s.d. ($n = 3$ experimental replicates). **c** Representative flow cytometry plots and quantitative analysis of Annexin V positive cells after 48 hours of different treatment in Ka539 leukemia cells. Data were presented as mean ± s.d. ($n = 3$ experimental replicates). **d** Representative flow cytometry plots and quantitative analysis of Annexin V positive cells after 48 hours of different treatment by in C1498 leukemia cells. The leukemia cells were harvested at 48 h after different treatments and displays the same number of cells in

each flow cytometry result. Data were presented as mean ± s.d. ($n = 3$ experimental replicates). **e** Representative flow cytometry plots of leukemia stem cells (Sca-1+c-Kit + ) after 48 hours of different treatment in C1498 cells. **f** Quantitative analysis of LSC in C1498 cells. Data were presented as mean ± s.d. ($n = 3$ experimental replicates). **g** Representative flow cytometry plots of mature cells after 48 hours of different treatment in C1498 cells. **h** Quantitative analysis of mature cells in C1498 cells. $n = 3$ experimental replicates. The data were presented as mean ± s.d. Statistical significance of $P$ values was calculated via a two-tailed, unpaired Student's t test and were indicated as $*P < 0.05$, $**P < 0.01$ and $***P < 0.001$. Source data are provided as a Source Data file.

membrane-coated nano-drug which responds to bone marrow hypoxia microenvironment for leukemia treatment[27]. Gu et al. reported utilization of living HSCs or liquid nitrogen treated C1498 cells to deliver drugs to bone marrow and to treat leukemia[28,29]. In this study, we prepared HSPC biomimetic vesicles to specifically deliver anti-leukemia drugs to bone marrow. As one of the most widely used delivery systems in clinic, liposomes have high biocompatibility, flexible drug loading capability and controllable drug release. This study combines the advantages of liposome and targeting ability of HSPCs cell membrane, and constructs a biomimetic vesicle by infusing HSPC cell membrane into liposomes. The biomimetic vesicles were significantly accumulated in the bone marrow of leukemic mice and displayed specific affinity to the leukemia cells. Thus, HSPC-Lipo enhanced the toxicity to leukemia cells and prolonged the survival of leukemia-bearing mice after effective carrying and delivering Ara-C.

Adhesion molecules or receptors on the surface of HSPCs play a pivotal role in mediating homing process. In our study, the existence of CD44 in HSPC-Lipo was verified by results of WB experiments. Hyaluronic acid is a classic ligand of CD44, and studies have shown that the content of HA in the bone marrow of leukemia patients is significantly increased[21,30]. Concordantly, we verified that the content of HA in the bone marrow of leukemia-bearing mice was significantly higher than that of normal mice, confirming that CD44-HA axis is involved in mediating bone marrow targeting of HSPC-Lipo. We identified cell adhesion molecules in HSPC-Lipo, ITGB2, which is involved in cell adhesion and surface signal transduction, and plays important roles in cell adhesion and recognition. ICAM-1 and ITGB2 are documented as ligand-receptor, and previous studies have shown that ICAM-1 is highly expressed on the surface of leukemia cells[22,31–34]. We found that ICAM-1 was highly expressed in mouse and human leukemia cell lines, but was low expressed in HSCs in bone marrow, indicating that the adhesion molecules on the surface of nanoparticles are important reasons in mediating the specific targeting of leukemic cells.

In this study, the drug loading efficiency and anti-leukemia ability of HSPC-Lipo were tested both in vitro and in vivo. Cell differentiation, maturation, and the frequency of LSCs are important indicators for evaluating the effectiveness of leukemia treatment. Through in vitro toxicity experiments, we found that after the treatment of drug loading HSPC-Lipo, the apoptosis and differentiation of leukemic cells were significantly increased, and the proportion of LSC were significantly decreased, indicating the superior therapeutic effect of Ara-C@HSPC-Lipo. The subsequent animal experiment results also confirmed that the survival of leukemic mice treated with Ara-C@HSPC-Lipo was significantly prolonged. Whether HSPC-Lipo vesicles can stimulate the immune system and form long-term immune memory is worth further study.

In general, this study constructs HSPC biomimetic vesicles for targeting bone marrow, which significantly increase the accumulation in bone marrow of leukemic mice and exert specific and efficient anti-leukemia effect. This study provides a way for the treatment of leukemia and a reference for the targeted treatment of other bone marrow-derived diseases.

## Methods

### Ethical statement
All animal experiments were carried out in compliance with the 3 R principle, and were approved by Ethics Committee of Zhejiang University under approval number of ZJU20230168.

### Materials and Reagents
Liposomes were prepared from lecithin (Aladdin, Cat. L105732) by reverse phase evaporation. The fluorescent dye Did (Cat. C1039) and BCA kit (Cat. P0010) were purchased from Beyotime Biotechnology. The chemotherapy drug Ara-C was purchased from MCE Inc (Cat. No: HY-13605). The apoptosis detection kit was used Annexin V-Alexa Fluor 647/PI Apoptosis Detection Kit (Cat. 40304ES60) purchased from Hangzhou Yisheng Co., LTD.

### Cell culture
The murine cell lines C1498 (ATCC TIB-49) and 32D (ATCC CRL-3594) were obtained from the American Type Culture Collection (ATCC). The murine cell line Ka539 was a gift from Y. Liu (Sichuan University). The human AML cell lines HL-60 (ATCC CCL-240) were obtained from the American Type Culture Collection (ATCC). The human AML cell lines OCI-AML2 (DSMZ ACC 99), OCI-AML3 (DSMZ ACC 582), SKM-1 (DSMZ ACC 547) and NB-4 (DSMZ ACC 207) were purchased from the Leibniz-Institute DSMZ. Ka539 cells were mouse B-cell leukemia/lymphoma mixed cell line cultured in 45% DMEM + 45% IMDM + 10% FBS+β-Me medium. The cells were passed every two days. C1498, a mouse acute myeloid leukemia cell line, was cultured with DMEM + 10% FBS and passed every 72 hours. 32D cells, a murine hematopoietic progenitor cell line originally generated in C3H/Hej mouse strain and immortalized by murine leukemia virus, were cultured in 90% 1640 + 10% FBS + 10 ng/ml mIL-3 media, and used as source of HSPC for knockdown experiments in Fig. 3 and Supplementary Fig. 5. OCI-AML2 and OCI-AML3 were cultured in α-MEM supplemented with 20% FBS. HL-60, SKM-1 and Nalm6 were cultured in IMDM supplemented with 10% FBS. Cells were cultured in a constant temperature incubator at 37 degrees with 5% carbon dioxide.

### Animals
C57BL/6 mice, BALB/c mice and SD rats were purchased from The Jackson Laboratory, Gempharmatech Co., Ltd. and Animal Center, School of Medicine, Zhejiang University. Female mice aged 6 to 8 weeks were used for the experiment. All mice were kept in an SPF environment. All mice were kept under specific pathogen-free conditions, nurtured in an environment with proper temperature and humidity, and provided with abundant water and nourishment (25 °C, optimal humidity typically at 50%, and a 12-hour dark/light cycle).

### Isolation of HSPCs and cell membrane
The isolation of HSPCs method was provided as follow and the cell membrane isolation method originates from existing reports[35,36]. The isolation of hematopoietic progenitor cells was using the Mouse Hematopoietic Progenitor Cell Isolation Kit (MojoSort™, Cat. 480004, BioLegend). The detailed methodology followed these steps. Initially, mouse bone marrow cells were flushed out using a 3-mL syringe and

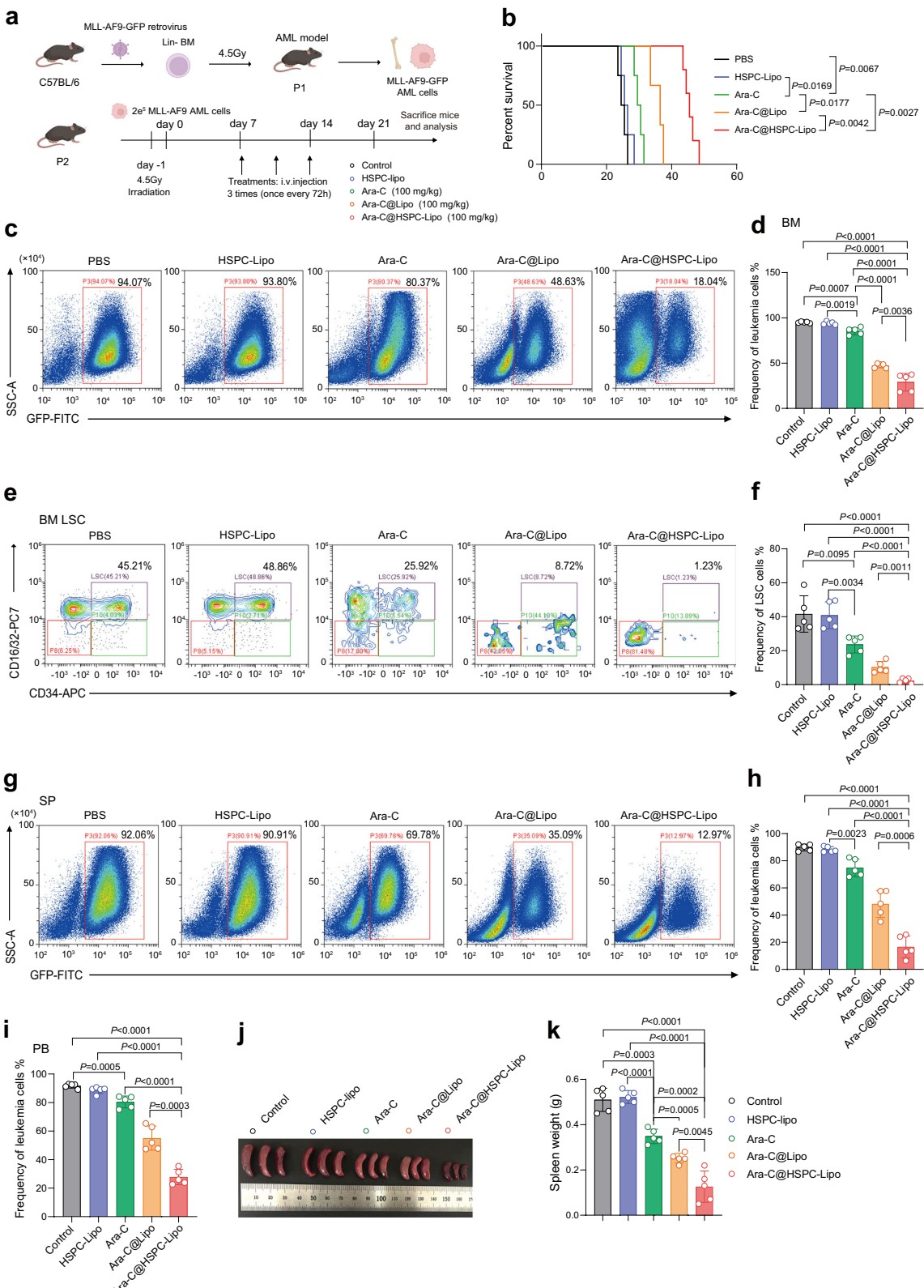

centrifuged at 300 g for 5 minutes. Subsequently, 2 mL of red blood cell lysis buffer was introduced and the mixture was lysed at room temperature for 4 minutes. Following this, they were reconstituted in PBS and subjected to cell counting. At 4°C, the above cells were centrifuged again at 300 g for an extended period of 10 minutes. Post-centrifugation, the cells were resuspended in a solution composed of precooled PBS + 2% BSA + 2 mM EDTA, with a ratio of 100 μL per every

10 million cells. Each batch of 10 million cells was then incubated with 10 μL of the Biotin-Antibody cocktail at 4 degrees for 15 minutes. Following this step, 10 μL Streptavidin Nanobeads were added per 10 million cells and the mixture was again incubated at 4°C for another 15 minutes. After incubation, cells underwent washing with 4 mL of PBS under centrifugation conditions of 300 g for 10 min. The supernatant was carefully discarded, and the cells were resuspended in 2.5 mL PBS

**Fig. 5 | The anti-leukemic effect of Ara-C@HSPC-Lipo in MLL-AF9 leukemia mouse model. a** Schematic illustration of animal experiment design. Cell membranes were derived from primary isolated HSPCs. Each mouse received approximately 20 μg of liposomes and 10 μg of cell membrane. **b** Survival curves of the leukemic mice received different treatments. The statistics and *P*-values were calculated using the Log-rank (Mantel-Cox) test. (*n* = 6 mice for control, HSPC-Lipo and Ara-C group, and *n* = 7 mice for Ara-C@Lipo group, and *n* = 11 mice for Ara-C@HSPC-Lipo group). **c** Representative flow cytometry plots of leukemia cells (GFP positive cells) in bone marrow. Leukemic mice were euthanized on the day 21 to collect bone marrow cells for flow cytometry analysis. **d** Quantitative analysis of leukemia cells in bone marrow. Data were presented as mean ± s.d. (*n* = 5 mice).

**e** Representative flow cytometry plots of leukemia stem cells (CD34 + CD16/32 + ) in bone marrow after different treatments. **f** Quantitative analysis of leukemia stem cells in BM. Data were presented as mean ± s.d. (*n* = 5 mice). **g** Representative flow cytometry plots of leukemic cells (GFP positive cells) in spleen. **h** Quantitative analysis of leukemia cells in spleen. Data were presented as mean ± s.d. (*n* = 5 mice). **i** Quantitative analysis of leukemic cells (GFP positive cells) in peripheral blood. Data were presented as mean ± s.d. (*n* = 5 mice). **j** Representative spleen images in leukemic mice after different treatment. **k** Weight of Spleen. *n* = 5 mice. The data were presented as mean ± s.d. Statistical significance of *P* values was calculated via a two-tailed, unpaired Student's t test and were indicated as **P* < 0.05, ***P* < 0.01 and ****P* < 0.001. Source data are provided as a Source Data file.

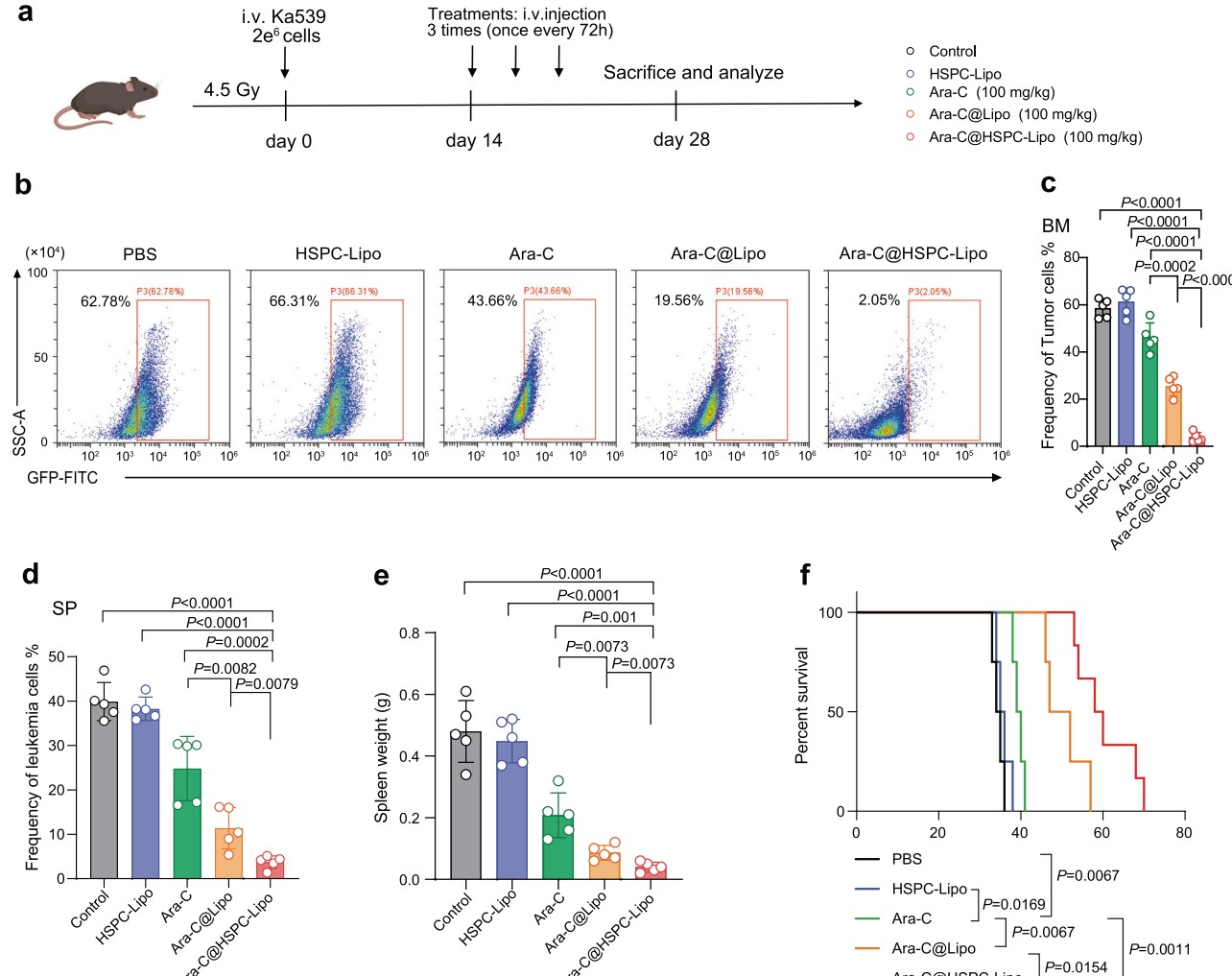

**Fig. 6 | The anti-leukemic effect of Ara-C@HSPC-Lipo in Ka539 leukemia mouse model. a** Schematic illustration of animal experiment design. Cell membranes were derived from primary isolated HSPCs. Each mouse received approximately 20 μg of liposomes and 10 μg of cell membrane. **b** Representative flow cytometry plots of leukemia cells (GFP positive cells) in bone marrow. Leukemic mice were euthanized at day 28 to collect bone marrow cells for analysis. **c** Quantitative analysis of leukemia cells in bone marrow. Data were presented as mean ± s.d. (*n* = 5 mice). **d** Quantitative analysis of leukemia cells in spleen. Data

were presented as mean ± s.d. (*n* = 5 mice). **e** Weight of Spleen. Data were presented as mean ± s.d. (*n* = 5 mice). **f** Survival curves of the leukemic mice received different treatments. The statistics and *P*-values were calculated using the Log-rank (Mantel-Cox) test. (*n* = 6 mice for control, HSPC-Lipo and Ara-C group, and *n* = 8 mice for Ara-C@Lipo, and *n* = 12 mice for Ara-C@HSPC-Lipo group). Statistical significance of *P* values was calculated via a two-tailed, unpaired Student's t test and were indicated as **P* < 0.05, ***P* < 0.01 and ****P* < 0.001. Source data are provided as a Source Data file.

within a 5 mL polypropylene tube (12 × 75 mm). This tube was then placed in a magnetic field for a period of 5 minutes. This entire process was meticulously repeated once more, culminating in a total of two separation cycles. Upon completion, the liquid was decanted, and the hematopoietic progenitor cells contained therein were collected into fresh, clean tubes for subsequent applications.

The purified HSPCs underwent freezing and thawing repeatedly from −80 °C to 37 °C three times, followed by getting enucleated in a hand-held Dounce homogenizer (20 passes while on ice) to remove the cell organelles such as nucleus. Then, HSPC membrane were resuspended with a lysis buffer containing a mixture of 75 mM NaCl, 6 mM NaHCO₃ (Fisher Chemical, Cat. MFCD00003528), 1.5 mM KCl (Fisher

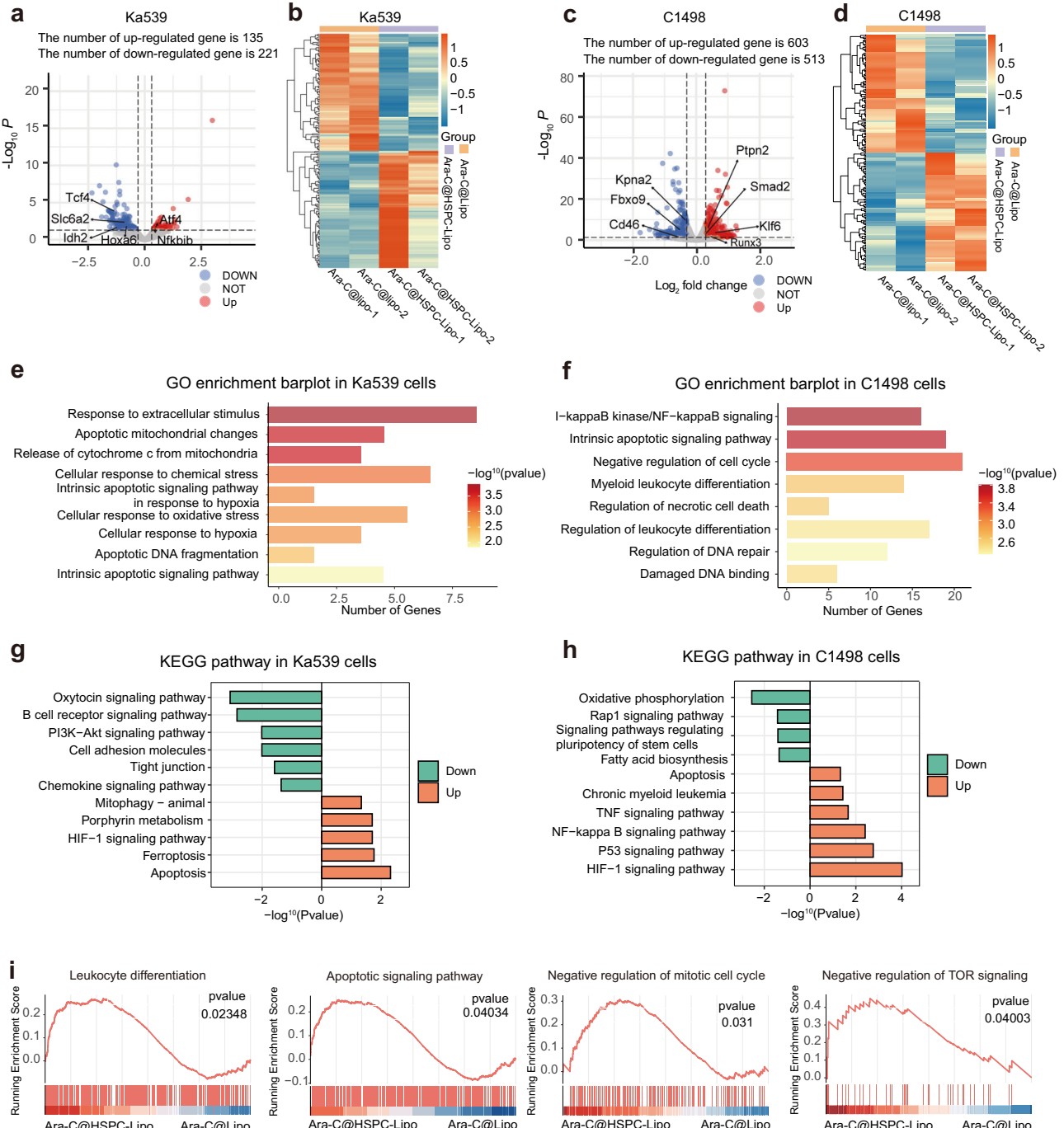

**Fig. 7 | Transcriptome analysis reveals mechanisms of Ara-C@HSPC-Lipo treatment. a** Volcano plot of differential gene expression in Ka539 cells. **b** Heat map of clustering analysis of differential gene in Ka539 cells. **c** Volcano plot of differential gene expression in C1498 cells. **d** Heat map of clustering analysis of differential gene in C1498 cells. **e** GO pathway enrichment analysis in Ka539 cells. **f** GO pathway enrichment analysis in C1498 cells. **g** KEGG enrichment analysis in Ka539 cells. **h** KEGG enrichment analysis in C1498 cells. **i** Enrichment analysis of GSEA gene set in C1498 cells. The statistical analyzes of *P* values were derived from a two-sided statistical test (**a, c**) or one-sided statistical test (**e–i**) and without adjustment for multiple comparisons.

Chemical, Cat. MFCD00011360), 0.17 mM Na$_2$HPO$_4$ (Fisher Chemical, Cat. MFCD00003496), 0.5 mM MgCl$_2$ (Aladdin, Cat. M113692), 20 mM HEPES (Beyotime Biotech, Cat. C0215), 1 mM ethylenediaminetetraacetic acid (Fisher Chemical, Cat. MFCD00150037), and protease inhibitors (Thermo Scientific, Cat. A32963). After centrifugation at 3200 g for 5 min at 4 °C, the resulting supernatant was collected and centrifuged at 211000 g for 30 min for HSPC membrane collection. The collected cell membrane was suspended and stored in distilled water at 4 °C. As for the CD44 and ITGB2 knockdown HSPC cell membranes, we constructed CD44 and ITGB2 knockdown by

transducing progenitor cell line (32D) with lentiviruses carrying CD44 and ITGB2 shRNA (https://www.sigmaaldrich.cn). The shRNA sequences were as follow: Non-target shRNA: GCGCGATAGCGCTAATAAT; ITGB2:CCAGGAATGCACCAAGTACAA; CD44:CCTCCCACTATGACA-CATATT. Then collected the cell membranes follow the above steps and conducted the subsequent experiments.

**Preparation of HSPC-Lipo**

As for the isolation of HSPC cells, mouse bone marrow cells were flushed with a 3 mL syringe, and the HSPC cells and cell membranes

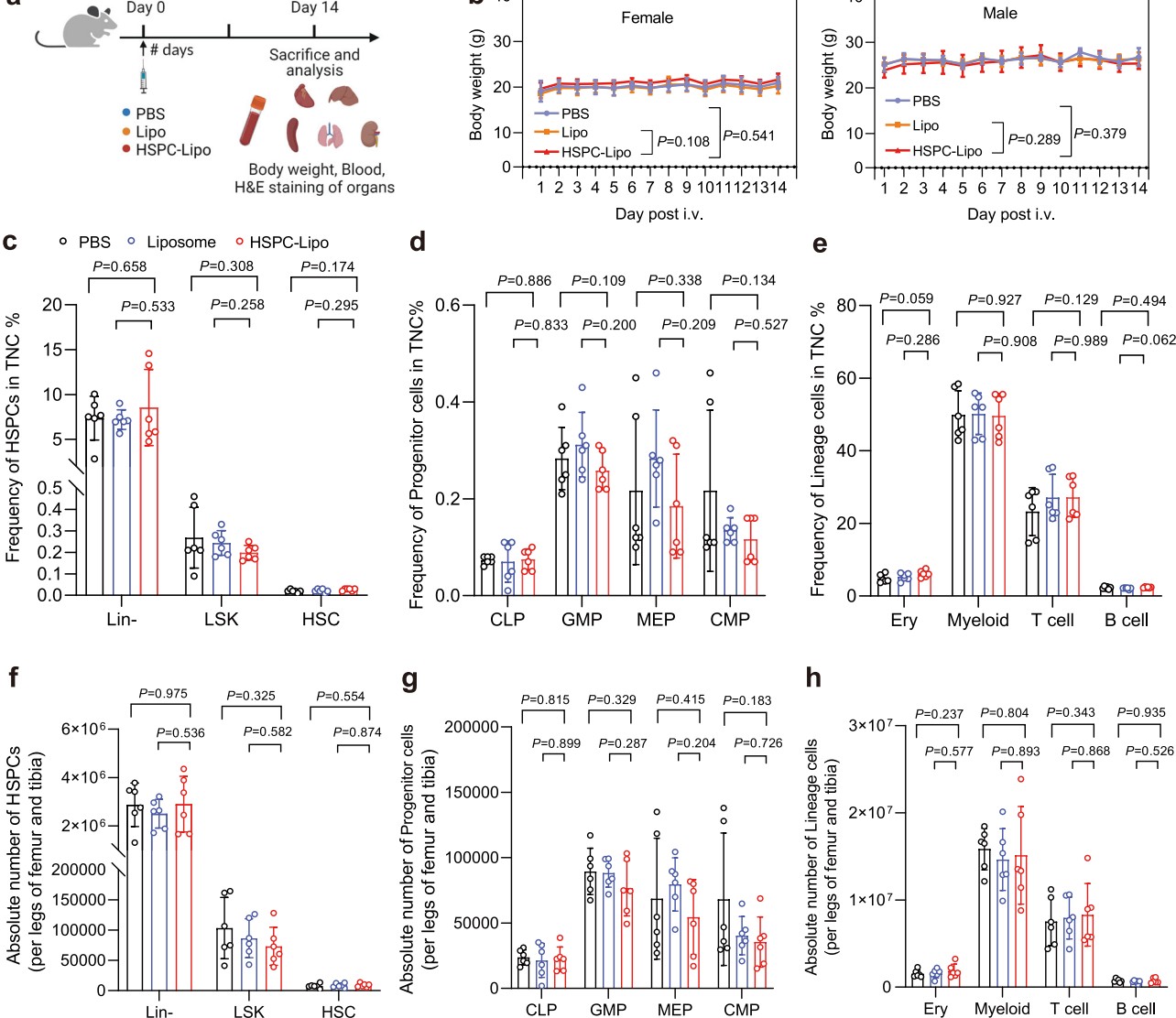

**Fig. 8 | Safety evaluation and analysis of HSPC-Lipo. a** Schematic illustration of animal experiment design. **b** Body weight of mice in different treatment groups. Data represent the mean ± s.d. ($n = 6$ mice). **c** Quantitative analysis of hematopoietic stem progenitor cells in total mononuclear cells. **d** Quantitative analysis of progenitor cells in total mononuclear cells. **e** Quantitative analysis of lineage cells in total mononuclear cells. **f** Quantitative analysis of absolute number of hematopoietic stem progenitor cells per legs of tibia and femur. **g** Quantitative analysis of absolute number of progenitor cells per legs of tibia and femur. **h** Quantitative analysis of absolute number of lineage cells per legs of tibia and femur. $n = 6$ mice (**c-h**). The data represent the mean ± s.d (**c-h**). Statistical significance of $P$ values was calculated via a two-tailed, unpaired Student's t test and were indicated as *$P < 0.05$, **$P < 0.01$ and ***$P < 0.001$. Source data are provided as a Source Data file.

were collected according to the above method. The protein quantification of the obtained cell membranes was executed using the BCA Protein Assay Kit (Beyotime Biotechnology, Catalog No. P0010). The thin-film hydration method was adopted for liposome preparation[37]. Dissolve 20 mg of lecithin in ethanol, spin dry ethanol under negative pressure at 35 °C and 120 rotation speed to prepare a thin film. The thin film was rehydrated with PBS buffer containing the cell membrane suspension, following which the phospholipid-to-cell membrane protein mass ratio was adjusted to 1:0.5 based on our established protocols[12,13]. Then, put the mixture of liposome and cell membrane into ultrasonic disrupter (SCIENTZ-IID, SCIENTZ, Chima) under ice bath and cell membrane and liposome fusion were performed by ultrasonic cavitation (60 w) for 6 min. Ultracentrifuge centrifugation at 4 °C, 150,000 $g$ conditions for 3 h, remove the supernatant, the precipitate is the HSPC-Lipo.

## Characterization of HSPC-Lipo
The average particle size, zeta potential, and PDI of HSPC vesicles and HSPC-Lipo were analyzed using dynamic light scattering DLS measurements (Mastersizer 2000, Malvern, USA). For the long-term stability test of HSPC-Lipo, 5 mg/mL HSPC-Lipo and liposome were resuscated with 90% BSA, respectively, and placed at 4°C. Particle size was tested daily. The morphological characteristics of HSPC-Lipo were measured by projective electron microscopy (G2 Spirite 120kv, FEI Technai, USA). The structure of HSPC vesicles, HSPC-Lipo, and Liposome functional groups was determined by Fourier infrared spectroscopy (ANTARIS, Thermo Fisher, USA).

## Isolation of Mesenchymal stem cells
SD mice were purchased from Animal Center, School of Medicine, Zhejiang University. SD rats aged 4 to 8 weeks were sacrificed by

cervical dislocation and fully disinfected with 75% alcohol. The tibia and femur of the mouse were isolated and removed under sterile conditions. Cut both ends of the tibia and femur, and then use a 1 mL syringe to draw complete culture medium. Wash the bone marrow cells from one end of the bone into a 50 mL sterile centrifuge tube. Repeat twice until the bones turn white. Add 5 times the volume of red blood cell lysis solution to the 50 mL centrifuge tube, pipe repeatedly with a Pasteur pipette, and let stand for 15 minutes. After standing, centrifuge at 200 $g$ for 10 min and discard the supernatant. Add an appropriate amount of DMEM complete culture medium to resuspend the cells, and then filter the cells through a 200-mesh filter. Centrifuge at 200 g for 10 min, discard the supernatant, and repeat twice. Add DMEM complete culture medium to the tube for in vitro culture to obtain MSC.

### Isolation of red blood cells
C57BL/6 mice were purchased from Animal Center, School of Medicine, Zhejiang University. Take C57BL/6 mice aged 4 to 8 weeks, and collect blood from the orbit to isolation of whole blood. Transfer 10 mL of whole blood into a 50 mL centrifuge tube, add 10 mL of PBS solution to dilute, and mix gently. Take two 15 mL centrifuge tubes and add 5 mL Ficoll solution. Then gently add the diluted blood to the upper layer of Ficoll in the two centrifuge tubes, centrifuge at 1000 g for 20 minutes, and the red blood cells will be in the bottom layer.

### Isolation of macrophage
C57BL/6 mice were purchased from Animal Center, School of Medicine, Zhejiang University. C57BL/6 mice aged 4 to 8 weeks were sacrificed by cervical dislocation and fully disinfected with 75% alcohol. The tibia and femur of the mouse were isolated and removed under sterile conditions. Cut both ends of the tibia and femur, and then use a 1 mL syringe to draw complete culture medium. Wash the bone marrow cells from one end of the bone into a 50 mL sterile centrifuge tube. Repeat it twice until the bones turn white. Add 5 times the volume of red blood cell lysis solution to the 50 mL centrifuge tube, pipe repeatedly with a Pasteur pipette, and let stand for 15 minutes. After standing, centrifuge at 200 $g$ for 10 min and discard the supernatant. Add an appropriate amount of 1640 cell culture medium to resuspend the cells, and then filter the cells with a 200-mesh filter. Centrifuge at 200 $g$ for 10 min, discard the supernatant, and repeat twice. Add 1640 complete medium containing 10 ng mL$^{-1}$ M-CSF (PeproTech, Cat. 315-02-50) to the tube to resuspend the cells. After one week of culture in vitro, bone marrow cells were induced to differentiate into macrophages.

### Isolation of neutrophils
C57BL/6 mice were purchased from Animal Center, School of Medicine, Zhejiang University. C57BL/6 mice aged 4 to 8 weeks were sacrificed by cervical dislocation and fully disinfected with 75% alcohol. The tibia and femur of the mouse were isolated and removed under sterile conditions. Cut both ends of the tibia and femur, and then use a 1 mL syringe to draw complete culture medium. Wash the bone marrow cells from one end of the bone into a 50 mL sterile centrifuge tube. Repeat it twice until the bones turn white. Pass the bone marrow cell suspension through a 200-mesh cell sieve, transfer to a 15 mL centrifuge tube, and centrifuge at 600 $g$ for 5 minutes at 4 °C. Discard the supernatant, add 3–5 times the red blood cell lysate to the cell pellet, pipet gently to mix evenly, lyse for 1-2 minutes, centrifuge at 600 g for 5 minutes at 4°C, and collect the cell pellet. Neutrophils from bone marrow cells were isolated and purified using discontinuous density gradient centrifugation. Add an appropriate amount of serum to a 15 mL centrifuge tube, rinse the tube wall and discard. Add 2 mL of 78% Percoll solution. Use sterile injection to add 2 mL of 65% and 55% Percoll solutions to the 78% Percoll solution in sequence. above the

liquid level. Pipette the cell suspension, use a sterile syringe to draw the cell suspension, slowly add it along the wall of the centrifuge tube to the top of the 55% Percoll solution, and centrifuge at 1600 $g$ for 30 minutes at 25°C. A layer of white material will appear at the interface between 78% and 65%, aspirate this layer of material and transfer it to a centrifuge tube, add an equal volume of cold PBS to wash, and centrifuge at 600 $g$ for 3 minutes at 4 °C to remove excess percoll. Finally, add an appropriate amount of RPMI 1640 and resuspend to obtain neutrophils.

### Cell membrane derivation
The purified MSCs, RBCs, macrophages and neutrophils underwent freezing and thawing repeatedly from -80 °C to 37 °C three times, followed by getting enucleated in a hand-held Dounce homogenizer (20 passes while on ice) to remove the cell organelles such as nucleus. Then, cell membranes were resuspended with a lysis buffer containing a mixture of 75 mM NaCl, 6 mM NaHCO$_3$ (Fisher Chemical, Cat. MFCD00003528), 1.5 mM KCl (Fisher Chemical, Cat. MFCD00011360), 0.17 mM Na$_2$HPO$_4$ (Fisher Chemical, Cat. MFCD00003496), 0.5 mM MgCl$_2$ (Aladdin, Cat. M113692), 20 mM HEPES (Beyotime Biotech, Cat. C0215), 1 mM ethylenediaminetetraacetic acid (Fisher Chemical, Cat. MFCD00150037), and protease inhibitors (Thermo Scientific, Cat. A32963). After centrifugation at 3200 g for 5 min at 4 °C, the resulting supernatant was collected and centrifuged at 211000 $g$ for 30 min for HSPC membrane collection. The collected cell membrane was suspended and stored in distilled water at 4 °C.

### Preparation of DiD-labeled HSPC-Lipo
DiD (Beyotime Biotech, Cat. C1039) was hydrophobic dye, the thin film dispersion method was adopted for DiD dye loading. Shortly, the dye-carrying films were prepared by precision weighing 50, 125, 250, 500, 750, and 1000 μg of DiD dye with 20 mg of lecithin (Aladdin, Cat. L105732) dissolved in anhydrous ethanol, and rotary evaporating the solvent at 35 °C and 120 rotation speed. The suspension containing HSPC membranes was added and hydrated, and adjusted the ratio of phospholipid mass to cell membrane protein mass to 1:0.5. Then, put the mixture of liposome and cell membrane into ultrasonic disrupter (SCIENTZ-IID, SCIENTZ, China) under ice bath and cell membrane and liposome fusion wereas performed by ultrasonic cavitation probe sonication (60 w) for 6 min. Ultracentrifuge centrifugation at 4 °C, 150,000 $g$ conditions for 3 h, can be separated from the free dye and the loaded drug, remove the supernatant, the precipitate is the HSPC-Lipo carrying DiD dye and suspended into 2 mL of PBS buffer. As figure S displayed, 20 mg of phospholipids can carry about 400 μg of DiD dye, which is much higher than the 50 μg (25 μg mL$^{-1}$) we used for labeling. Further, DiD encapsulation rates at different time points after carryover were examined (Supplementary Fig. 4a, b). It illustrated that the HSPC-Lipo could carry DiD stably for a long time.

### Preparation of Indocyaninegreen(ICG)-Fluorescent Labeled HSPC-Lipo
ICG (Aladdin, Cat. I107931) was loaded into HSPC-Lipo for in vivo biodistribution detection. The reverse evaporation method was adopted for ICG loading based on the water solubility of ICG. 10 mg of ICG was dissolved in 1 mL of ultrapure water, and 20 mg of lecithin (Aladdin, Cat. L105732) was dissolved in 3 mL of ether, and the microemulsion was prepared by ultrasonication. The solvent was evaporated at room temperature until the solution was viscous when PBS buffer was added, and the residual ether was removed by spin evaporation to obtain the ICG-carrying liposome. Suspension containing HSPC membranes was added and adjust the ratio of phospholipid mass to cell membrane protein mass to 1:0.5. Then, put the mixture of liposome and cell membrane into ultrasonic disrupter (SCIENTZ-IID, SCIENTZ, Chima) under ice bath and cell membrane and liposome fusion were performed by ultrasonic cavitation (60 w) for

6 min. Ultracentrifuge centrifugation at 4 °C, 150,000 $g$ conditions for 3 h, can be separated from the free ICG and the loaded ICG, remove the supernatant, the precipitate is the HSPC-Lipo carrying ICG.

### Preparation of Ara-C@HSPC-Lipo

Due to the water-solubility of cytarabine, HSPC-Lipo carrying cytarabine was prepared by reverse evaporation method. 30 mg of cytarabine (MCE) was dissolved in 1 mL of ultrapure water, and 20 mg of lecithin (Aladdin, Cat. L105732) was dissolved in 3 mL of ether, and the microemulsion was prepared by ultrasonication. The solvent was evaporated at room temperature until the solution was viscous when PBS buffer was added, and the residual ether was removed by spin evaporation to obtain the drug-carrying liposome. Suspension containing HSPC membranes was added and cell membrane fusion was performed by probe sonication (60 w) for 6 min. Ultracentrifuge centrifugation at 4 °C, 150,000 $g$ conditions for 3 h, can be separated from the free drug and the loaded drug, remove the supernatant, the precipitate is the HSPC-Lipo carrying cytarabine.

### Immunofluorescence labeling of the HSPC membrane

To avoid cross staining of two components with hydrophobic dyes, we utilized anti-CD44 antibody (Boster, Cat. A00052, dilution ratio: 1:500) and secondary antibodies (Cy3-labeled Goat Anti-Rabbit IgG (H + L), Beyotime, Cat. A0516) to label the HSPC membrane. The liposomes were labeled by DiD dye. In the immunofluorescence co-localization experiment, about $2 \times 10^5$ C1498 leukemia cells were inoculated into 12-well plates and incubated with fluorescent-labeled HSPC-Lipo for 2 hours. After washing and centrifugation with PBS, after nuclear staining with DAPI, cells were harvested and observed by laser confocal microscope.

### In vivo bone marrow targeting and distribution of HSPC-Lipo

As for the in vivo targeting ability of HSPC-lipo, ICG-labeled HSPC-Lipo, liposomes, and free ICG were injected into the leukemic mice via tail vein with a dose of 400 µg/kg. After 2, 6, 12, 24 hours, the mice were sacrificed to obtain bone marrow, heart, liver, spleen, lung, and kidney for analysis. All organs were immediately photographed and analyzed using an in vivo imaging system (Maestro, Cambridge Research & Instrumentation, USA).

### Western Blot

The prepared HSPC cell membrane and HSPC-Lipo were lysed in a cold cell lysis buffer and protease inhibitor mixture for 10 min. Centrifuge at 4 degrees 12000 r for 10 minutes. The protein concentration of sample was determined using the BCA detection kit (BCA Protein Assay Kit, Beyotime, Cat. P00010) and prepared at a protein concentration of 600 µg mL$^{-1}$, and then add 5 × Load buffer solution and incubate at 99 °C for 10 minutes, and store at -80 °C. The experimental steps of western blotting are as follows. All samples were loaded onto bis-tris protein concentrated gels in a volume of 20 µL. The protein electrophoresis condition was 60 volts constant pressure for 90 minutes. The PVDF membrane were used for membrane transfer under the condition of 300 mA for 90 minutes. 5% skimmed milk was used for blocking at room temperature for 1 h. Then wash with TBST three times, each time for 10 minutes. The incubation condition for the primary antibody were at 4 °C overnight at a dilution ratio of 1:1000 (CD44, Boster, Cat. A00052; ITGB2, Beyotime, Cat. AF6399; CXCR4, Beyotime, Cat. AF6621; GAPDH, Sangon Biotech, Cat. D190090-0100, dilution 1:5000). The dilution ratio of secondary antibody (Goat Anti-Mouse IgG(H + L)(peroxidase/ HRP conjugated), Elabscience, Cat. E-AB-1001; Goat Anti-Rabbit IgG (H + L)(peroxidase/HRP conjugated), Elabscience, Cat. E-AB-1003) was 1:5000 at room temperature for about 1 hour. The ECL imaging and figures were collected by the Bio-Rad GelDoc Go with automatic exposure time.

### PCR detection of Mycoplasma in progenitor cell line (32D) cells

Samples were obtained after cell cultured for more than 3 days. 1–6 mL of the cell culture supernatant to be assayed was centrifuged at 300 $g$ for 3 min to precipitate cell debris. The supernatant was transferred to a EP tube and centrifuged at 15,000 $g$ for 10–15 min to precipitate mycoplasma. The supernatant was discarded and approximately 50 µL of the supernatant was retained to resuspend the precipitate. Because the medium RPMI1640 inhibited the PCR reaction, the samples were diluted 8-fold and used for detection. Samples were boiled at 95°C for 10 min, and 1 µL of the liquid was used as PCR template. For positive controls, MycoBlue Mycoplasma detector (Cat. #101, Vazyme) was used and diluted 30-fold with ddH20 before use. Sterile deionized water was used as a negative control (Cat. E607017-0100, Sangon). The volume required for single reaction detection was 1 µL of template, 6.25 µL of 2xPCR Master Mix, 0.5 µL of Mycoplasma F (10 µM) Mycoplasma R (10 µM), and 4.25 µL of ddH$_2$0. The primer sequence: Mycoplasma F: tgcaccatctgtcactctgttaacctc, Mycoplasma R: gggagcaaacaggattagataccct). PCR amplification was performed using a Biorad PCR apparatus. The program was 94°C for 10 min, and the 34-cycle program was 94°C for 1 min, 55°C for 30 seconds, 72°C for 30 seconds, and finally 72°C for 1 min. 2% agarose gel was used for imaging detection. When the Gel was cooled to about 60°C, 5 µL of Gel-Red dye was added and cooled for more than 15 minutes. 4 µL of the PCR amplification products were loaded. 4 µL DNA ladder (GoldBand 2000 DNA Marker, Cat. 10501ES60, Yeasen) was added to the far left of the gel. After electrophoresis (140 V) for 20 min, samples with specific bands appearing at 280 bp were defined as mycoplasma contamination cells.

### In vivo immunofluorescence staining of HSPC-Lipo

To detect the targeting ability of HSPC-Lipo in vivo, frozen slices of bone marrow from leukemic mice were used for immunofluorescence staining. Hyaluronic acid Antibody (Cat. PAA182Ge01, Cloud-Clone, US) was diluted at 1:500 and ICAM-1 antibody (Cat. AF1774, Beyotime) was diluted at 1:1000. The secondary antibody was PE-CY3-labeled fluorescent secondary antibody. The nuclei were stained with DAPI and diluted at 1:1000. After staining, CLSM (Carzeiss, LSM-800, Germany) was used for observation. Image J was used to analyze the fluorescence intensity of images.

### In vitro toxicity

For in vitro cytotoxicity experiments, cell viability was detected using a cell counter (CountStar). Specifically, cells after different treatment were taken and added with 10 µl of trypan blue dye for cell viability counting.

### Flow cytometry analysis

In the flow cytometry for cell apoptosis, about $5 \times 10^5$ cells were taken from each treatment group for Annexin V staining with 200 µL 1× binding buffer, after incubated on ice for 30 minutes without light, cleaned 3 times with PBS, and detected by flow cytometry. In the cell differentiation flow cytometry experiment, about $5 \times 10^5$ cells from different treatment groups were taken for flow cytometry staining. After 30 minutes of dark staining, clean 3 times with PBS, and perform flow cytometry on the CytoFLEX. Similarly, in flow cytometry testing of leukemia stem cells, 3 million cells from different treatment groups were stained. After incubating in dark for 40 minutes, clean three times with PBS and perform flow cytometry analysis.

The antibodies used in flow cytometry (FCM) were as follow. Anti-mouse CD45 (BV510, BioLegend, #103138), Ly-6G/Ly-6C Monoclonal Antibody (RB6-8C5, APC-Cyanine7,BioLegend, #108424), CD11b Antibody (M1/70, PE-Cyanine7, eBioscience, #25-0112-82), CD3e Monoclonal Antibody (145-2C11, APC, eBioscience, #17-0031-83), CD4 Monoclonal Antibody (RM4-5, PE-Cyanine5, eBioscience,#15-0042-83), CD8a Monoclonal Antibody (53-6.7, PE-Cyanine5, eBioscience, #15-

0081-83), CD3e Monoclonal Antibody (145-2C11, PE-Cyanine5, eBioscience, #15-0031-83), CD4 Monoclonal Antibody (RM4-5, PE-Cyanine5, eBioscience,#15-0042-83), CD8a Monoclonal Antibody (53-6.7, PE-Cyanine5, eBioscience, #15-0081-83), CD11b Monoclonal Antibody (M1/70, PE-Cyanine5, eBioscience, #15-0112-83), Ly-6G/Ly-6C Monoclonal Antibody (RB6-8C5, PE-Cyanine5, eBioscience, #15-5931-83), CD45R (B220) Monoclonal Antibody (RA3-6B2, PE-Cyanine5,eBioscience, #15-0452-83), IgM Monoclonal Antibody (II/41, PE-Cyanine5, eBioscience, #15-5790-82), TER-119 Monoclonal Antibody (TER-119, PE-Cyanine5, eBioscience, #15-5921-83), Ly-6A/E (Sca-1) Monoclonal Antibody (D7, PE-Cyanine7,BioLegend, #108114), CD117 (c-Kit) Monoclonal Antibody (2B8, APC,eBioscience, #17-1171-83), anti-mouse CD150 (SLAM) Antibody (TC15-12F12.2, PE, BioLegend, #115904), CD48 Monoclonal Antibody (HM48-1, eFluor 450, eBioscience,#48-0481-82), Rat Anti-mouse CD34 (RAM34, Alexa Fluor R 647, BD Pharm, #560230), Anti-mouse CD117 (c-kit, APC/cy7, BioLegend, #105826), Ly-6A/E (Sca-1) Monoclonal Antibody (D7, PE, eBioscience, #12-5981-82), CD127 Monoclonal Antibody (A7R34, eFluor 450, eBioscience, #48-1271-82), CD16/CD32 Monoclonal Antibody (93, PE-Cyanine7, eBioscience, #25-0161-82), Anti-human/mouse CD45R (B220) Antibody (RA3-6B2, PE, eBioscience,#12-0452-83), Hamster IgG1, λ1 Isotype Control (Clone: G235-2356, BV510, BD Horizon, #562954), Rat IgG2b kappa Isotype Control (APC-eFluor 780, eBioscience, #47-4031-82), Rat IgG2a kappa Isotype Control (PE-Cyanine7, eBioscience, #25-4321-82), Mouse IgG1 kappa Isotype Control (P3.6.2.8.1, APC, eBioscience, #17-4714-81), Rat IgG2a kappa Isotype Control (PE, eBioscience, #12-4321-83), Rat IgG2a kappa Isotype Control (PE-Cyanine5, eBioscience, #15-4321-82), Armenian Hamster IgG Isotype Ctrl Antibody (PE, BioLegend, #400908), Rat IgG1, k, Isotype Control (x40, BV421, BD Horizon, #562438), Rat IgG2b kappa Isotype Control (APC-eFluor 780, eBioscience, #47-4031-82), Hamster IgG1, λ1 Isotype Control (Clone: G235-2356, BV510, BD Horizon, #562954). The above antibodies were diluted 1:100 and used according to the procedure provided by the supplier.

### RNA-seq

For the extraction of total RNA, $1 \times 10^6$ of Ka539 and C1498 cells were pre-treated with Ara-C@HSPC-Lipo and Ara-C@Lipo (5 μg/mL) for 24 h and then harvested. Total RNA was isolated with Trizol, and then first-strand cDNA synthesis and cDNA libraries were constructed using the NEBNext UltraTM II RNA Library Prep Kit (New England Biolabs). cDNA quality was determined on Agilent 2100 BioAnalyzer (Agilent Technologies), and then sequenced on NovaSeq 6000 device (Illumina) to obtain pair-end 150 bp reads.

Raw data were trimmed by Trimmomatic (v0.39). Clean data were aligned to the mouse reference genome mm10 by HISAT2 (v2.1.0) with default setting. The resulting SAM files were converted to sorted BAM files using SAMtools (version 1.7). The aligned reads were quantified at the gene level using htseq-count (version 0.13.5).

Differential expressed genes (DEGs) were defined with a log2 fold change cutoff of |0.3| and a $p$-value < 0.05. by edgeR (v3.40.0). The significant DEGs were subjected to Gene Ontology (GO) and Kyoto Encyclopedia of Genes and Genomes (KEGG) enrichment analyzes using the clusterProfiler package (version 4.7.1). The resulting enriched terms and pathways were considered significant with a $p$-value < 0.05. Gene set enrichment analysis (GSEA) was performed using the clusterProfiler package in R to analyze differentially enriched gene sets between samples.

### Mass spectrometry experiments

Proteomic analyzes were performed using a Q Exactive HF X mass spectrometer (Thermo Fisher Scientific, San Jose, CA). Two sample groups were analyzed: HSPC and HSPC-lipo. In the HSPC group, a total of 57,622 spectra were acquired, with 5898 spectra matched and resulting in the identification of 1374 proteins and 5106 peptides. For the HSPC-lipo group, 58,635 spectra were acquired, with 7,867 spectra matched, identifying 1937 proteins and 6657 peptides.

Protein identification began with the separation of sample proteins via gel electrophoresis. Protein bands were excised from the gel, enzymatically digested, and the resulting peptides were extracted for mass spectrometric analysis. Peptides were ionized using a nanoESI source and analyzed in a Q Exactive HF X tandem mass spectrometer in Data Dependent Acquisition (DDA) mode. The primary mass spectrometry settings were as follows: ion source voltage was set to 1.9 kV; the scan range for the first mass spectrum was 350–1500 m/z with a resolution of 60,000; the second mass spectrum started at 100 m/z with a resolution of 15,000. Parent ions for fragmentation were selected based on a charge state of 2+ to 6+ and an intensity threshold of 10,000, prioritizing the top 30 parent ions. Ion fragmentation was performed using HCD with the fragments detected in the Orbitrap. Dynamic exclusion was set for 30 seconds with AGC targets of 3E6 for the first level and 1E5 for the second.

Raw mass spectrometry data were converted into peak files and searched against a database using protein identification software. The results were pre-processed and re-scored by Percolator to enhance the discrimination accuracy between correct and random matches. Spectral matches were filtered at a spectral-level False Discovery Rate (FDR) of 1% (PSM-level FDR ≤ 0.01) to yield a list of significantly identified spectra and peptides. Protein inference was then conducted based on the parsimony principle, resulting in a comprehensive proteome list. Functional annotation of the identified proteins was performed through Gene Ontology (GO) analysis (Supplementary Data 2).

### Murine leukemia model

C57BL/6 female mice aged 6–8 weeks were randomly divided into 5 groups. Mice in each group were treated with 4.5 Gy whole body irradiation, and then injected with 2 million Ka539 tumor cells through the tail vein. Two weeks later, the patients were given caudal intravenous drugs in the order of PBS, HSPC-Lipo, Ara-C, Ara-C@Lipo, and Ara-C@HSPC-Lipo. The injection was given every 72 hours for a total of 3 times. In vivo imaging of small animals was performed according to the timeline to assess tumor burden. Mice were sacrificed for analysis based on the imaging results and timeline. Briefly, the proportion of leukemia cells in peripheral blood, spleen and bone marrow was analyzed by flow cytometry. In the MLL-AF9 leukemia mouse model experiment, we selected lineage negative cells from mouse bone marrow, and then infected them with MLL-AF9-GFP retrovirus. After 48 hours, GFP positive leukemia cells were injected into 4.5 Gy irradiated mice ($2 \times 10^5$ cells per mouse) to construct the first-generation leukemia mouse model. After the onset of the first generation of mice, leukemia cells from the bone marrow of first-generation leukemia mouse were taken for second-generation tumor transplantation for subsequent animal experiments. The numbers of mice in each group of animal experiments were indicated in the corresponding figure legends.

### The maximal tumor burden statement

In the hematological malignancies, direct observation of tumor size or load is not feasible. To ensure animal welfare, euthanasia is commonly carried out at advanced stages of cancer, as signaled by symptoms such as limb paralysis, somnolence, and loss of appetite.

### H&E staining

After the mice were sacrificed, tibia and femur were taken for decalcification treatment, followed by tissue sections and H&E staining. The detailed operation was carried out by Wuhan PINUOFEI Biological Technology Co., Ltd. Sections were photographed using an inverted microscope.

# Article

## Safety evaluation of HSPC-Lipo

In the safety evaluation experiment, the weight of the mice was measured every 2 days. 2 weeks after injection, mice were sacrificed for analysis and bone marrow cells were collected, and the proportion of hematopoietic stem progenitor cells and lineage cells was analyzed. In the hemogram analysis of peripheral blood, 50 µl of tail vein blood was taken for the experiment.

## Statistics and reproducibility

The TEM and FTIR experiments in Fig. 2a and Supplementary Fig. 1d were independently repeated three times with similar results. The immunofluorescence experiments in Fig. 2c and Supplementary Figs. 2a, b were independently repeated three times with similar results. The western blot and gel electrophoresis experiments in Fig. 3e and Supplementary Fig. 5a–c were independently repeated three times with similar results. The H&E staining experiments in Supplementary Fig. 8, 9i, 10, and 13a,b were independently repeated three times with similar results.

The experiments in this study were repeated 3 times independently with similar results unless otherwise noted. No statistical method was used to predetermine sample size. No data were excluded from the analyzes. The experiments were not randomized. The investigators were double blinded to allocation during experiments and outcome assessment. The data were expressed as mean ±s.d (standard deviation). Statistical significance of $P$ values was calculated via a two-tailed, unpaired Student's t-test and were indicated as $*P < 0.05$, $**P < 0.01$, and $***P < 0.001$. GraphPad Prism 8.0 software was used for statistical analysis.

## Reporting summary

Further information on research design is available in the Nature Portfolio Reporting Summary linked to this article.

# Data availability

The authors declare that all relevant raw data presented in main figures and Supplementary Figs. has been provided in the Source Data file. Uncropped and unprocessed scans of blots have been provided as in the Source Data file. The raw data of RNA-seq have been deposited in the gene expression omnibus (GEO) repository under the accession number GEO: GSE232029. The mass spectrometry proteomics data have been deposited to the ProteomeXchange Consortium via the iProX partner repository with the dataset identifier PXD052979 The protein mass spectrometry data generated in this study are provided in the Supplementary Data 1–2. Source data are provided with this paper.

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

## Acknowledgements

This work was supported by grants from the National Key Research and Development Program of China (2022YFA1103500 to P.Q.), the National Natural Science Foundation of China (82222003, 92268117, 82161138028 to P.Q., 82370105, 82000149 to Q.W., U22A20383 to J.G.), the Key R&D Program of Zhejiang (2024SSYS0024 to P.Q.), the Zhejiang Provincial Natural Science Foundation of China (Z24H080001 to P.Q., LQ21H180006 to Q.W., LD22H300002 to J.G.), the Department of Science and Technology of Zhejiang Province (2023R01012 to P.Q.), and the Fundamental Research Funds for the Central Universities (226-2024-00007 to P.Q.), the China Postdoctoral Science Foundation (2021M702853 to Q.W.) and Zhejiang Province Postdoctoral Research Excellence Funding Project (ZJ2023151 to H.W.). The author, P.Q., gratefully acknowledges the support of the K.C. Wong Education Foundation. We thank G.C. and J.X. from the Core Facilities, Zhejiang University School of Medicine, and X.B. and J.W. from Liangzhu Laboratory for their technical support. The Figs. 1, 3a, 5a, 6a, 8a and Supplementary Fig. 9a were created with BioRender.com.

## Author contributions

J.X.L. and H.W. collaborated to complete the study. Z.Y. provided the necessary bioinformatics analysis for this study. Q.W. and X.Z. provided technique help for this study. L.J., J.Y.L., S.L., M.Z., and Y.H. aided in animal experiments. W.Q. provided the necessary help for the immunofluorescence experiment in this study. Conceptualization, reviewing, and supervision by J.G. and P.Q. All authors have read and agreed to the published version of the manuscript.

## Competing interests

The authors declare no competing interests.
