## [Peer Review File · Nature Communications]

REVIEWER COMMENTS

Reviewer #1 (Remarks to the Author):

MAJOR COMMENTS.

1. Novelty.

One of the highlights of this manuscript is the use of haematopoietic progenitor (HSPC)-coated liposomes for cytarabine drug delivery compared to other conventional liposomes. However the use of leukocyte - membrane, eg monocyte/neutrophil or red cell membrane liposomes has become more common as these liposomes have longer in vivo half-life (eg. reviewed in PMID; 36103910) and are also reported to preferentially home across vasculature to inflamed tissues, an advantage that appears to be shared with the HSPC-coated liposomes in this manuscript. In fact leukocyte membrane-coated particles have already been shown to successfully target tumours and carry anti-cancer drugs to tumours in vivo (eg. NatNaotech 2013, PMID:2341654, or 2015 PMID:25883092).

Liposomes carrying chemotherapy agents are also already well established for treatment of malignancies clinically including leukaemia (eg. 2018 PMID:30024784 CTX-351). The main novelty in this manuscript seems to be the use of HSPC membranes for coating of chemotherapy containing particles instead of other leukocyte-membranes / particles. Indeed this may offer greater BM homing potential compared to other leukocyte-membrane coated particles however this was not tested. Together these prior publications somewhat diminishes the novelty of the findings.

Can the authors mention how their HSPC coated liposomes compare with other leukocyte membrane-coated liposomes and comparative advantage, such as in efficiency of homing to bone marrow or other?

2.Methology and Figure legends.

The key findings in this manuscript are that AraC-loaded liposomes with HSPC coating provide superior anti-leukaemia efficacy in mice. Exactly how the authors achieve this and any critical interpretation of their data or controls is hard to determine given the ultra brief methods and figure legends provided (eg. this author could not find the doses of AraC administered to mice anywhere).

Authors are encouraged to read 'Ncomms submission guide' pdf document. Particularly figure legends section, last sentence "Legends should be detailed enough so that each figure and caption can, as far as possible, be understood in isolation from the main text." and provide more details to aid reader understanding of what is being shown. All figure legends require editing.

Some figure legends also appear to describe more an opinion, not a “short statement of what is depicted in the figure, not the results” a ncomms recommend.

Two examples (of several);

Fig 3k legend (‘HSPC-Lipo showed higher binding ability with leukemia cells’)

Fig 4c legend (‘Drug-loaded HSPC-lip nanoparticles significantly increased the apoptosis of leukaemia cells’). These legends are not appropriate and should be reworded in more objective form.

Although the authors frequently write “the experiment were repeated 3 times independently”, there is no mention of the number of mice per group (is it n=1?), nor what the shown data represents (technical replicates from one experiment? Pooled data from the 3 independent experiments?). (eg. Fig 4 in vitro study figures. Do the 3 dots shown in d,f,h,j represent three technical repeats of n=1 (variability seems very small), depict data from three independent repeats performed on separate dates, or does this show data from a single experiment selected from three performed? Please specify the number of replicates per experiment and what is shown, please correct for each figure).

Nature publishing also provides a “reporting life sciences research” pdf that may help provide guidelines around expectations reporting of experimental design and statistics (see first page).

OTHER COMMENTS.

STUDIES RESTRICTED TO TWO LYMPHOBLASTIC CELL LINES.

Use of two lymphoblastic cell lines (one established and cultured since 1973).

All studies in this manuscript were performed using two mouse lymphoblastic leukaemia cell lines (one established in 1973_ for all in vitro and in vivo assays. These two cell lines are;

-B lymphoma cell line (Ka539) and

-lymphoblastic cell line (C1498) originally isolated from a mouse with acute myeloid leukaemia. The cell line itself has lymphoblastic features.

The concern is these cell lines may not be very representative of ‘leukaemia’ in patients or mice.

For example several leukaemias (eg. AML) contain relatively rare leukaemia stems. It is these leukaemia stem cells that survive chemotherapy and re-establish relapse, not the bulk tumour cell population (which are relatively susceptible to chemotherapy). Long established and cultured cell lines may no longer contain any rare leukaemia stem cell populations. Also the assays used in these experiments are

not designed to consider impact of therapy on leukaemia stem cell populations after therapy compared to bulk tumour.

Furthermore both these two leukaemia cell lines share lymphoblastic features. Ideally a range of 'freshly derived' leukaemias would be tested (lymphoid and myeloid). Ideally the driving oncogenes/changes are diverse and known. A more naturally-forming leukaemia would be welcome. Many are now widely available (such as using mice transgenic for specific leukaemia oncogenes, or at least oncogene driven leukaemia eg. using methodologies first described in PMID: 19339691). Human leukaemias in NSG mice another option.

FIGURE COMMENTS.

Fig 2c. The authors use DiI staining (labels hydrophobic / lipophilic structures) to ascertain if liposomes were coated with Hematopoietic progenitor membrane. As liposomes themselves are amphipathic (have both hydrophilic and hydrophobic parts), it would be appropriate for the authors to include control (liposome only) staining as well. this is needed to confirm the 'HSPC membrane' shown in Fig 1c is only labelling cell membranes, not also liposomes.

Fig 2e. Femurs and tibias shown. Are these both from same mouse (ie. 1 mouse shown per treatment group?). If so how was this mouse chosen. How many mice per group per experiment? Legend states experiment was repeated 3 times. Does this mean 3 separate times with n=1 mice per group or a total of n=3 different mice once?

Fig 2e,g. Please specify timepoint post injection that these organs were collected. Please label the organs shown. In Fig 1g, it is not clear if these images are from control mice or mice with leukemic cells. Please specify. Could the authors also include their data for the other group (leukemic or non-leukaemia mice) as well for comparison (of liposome location in various tissues).

Fig 2 f,h,i. The values shown in Fig 2f, h, i (3 dots) have remarkably small variation. In contrast the images of femurs shown in Fig 2e, show some variation between femurs, or between tibias within each group (or within each mouse, this is not specified). Could the data shown in Fig 2f (with minimal variation) actually be three technical readouts from same mouse, or does each dot represent distinct data from separate mice? Are the data shown in Fig 2f from the same mice as in Fig 2e or different mice?. Could authors explain how they actually represented data from three biologically different mice with experiment performed 3 times in this figure. Improving the amount of detail provided in figure legend detail would greatly aid understanding of these points.

Figure 3.

Fig 3e. western blot Protein analysis. It is very difficult to understand relevance of what is being shown though schematic diagram shows comparison of 'HSPC cell membrane and HSPC-lipo'

Methods section; only give gives extremely brief description of HSPC membrane collection ("cells were repeatedly freeze-thawed for 3 times, centrifuged at 5000g for 10 min, and the cell membrane was obtained"). This super-quick method appears to not be a standard, nor a very pure membrane preparation method. There is no mention of homogenisation, of washing membrane particles to remove soluble proteins, or removal of unbroken cells from lysate or solubilisation of membrane proteins at end. Considering the importance of these membrane preps to this manuscript. A clearer method for membrane preparation is warranted.

Similarly method for Fig 3e, western blot verification" (of Fig 3a protein analysis). Methods section states; " HSPC-Lipo and HSPC cell membranes were suspended with 100uL protein lysate and were collected into 1.5mL EP tubes... centrifuge at 4 degrees... add 5x loading buffer". This method appears to be potentially also missing some key steps - otherwise both lysate groups shown in Fig 3e would be anticipated to look the same after western blotting.

Fig 4. Technical comments. In vitro toxicity studies in AML cell lines.

Fig4b. legend does not mention length of culture results making these data very hard to interpret.

Fig 4c. Please be aware that PI+ AnnexV+ staining may be either due to apoptosis or pyroptosis in these cells. It is not a specific stain for apoptosis only.

Also the % differences on axis are too small to read. Please make figure text legible. More importantly there doesn't appear to be the same number of gated cells within each group. Specifically there seems to be more cells shown in A@HSPC-Lipo group compared to others. This certainly makes it appear to have more dead cells. However these data should be shown without bias. That is with same number of cells loaded per group and clearly described method on how long these cultures were performed.

Fig 4 f,h,j. 'Drug-loaded HSPC-lipo significantly increased the differentiation of C1498 leukaemia cells. The histograms (Fig 4f,h,j) appear to have very little variation for " repeated 3 times independently" data. Could technical replicates have been accidentally shown. Could authors provide details on the data for each separate independent experiment and number of technical replicates performed within each.

Fig 5 d-i. Proportion of tumour (GFP+) cells in various tissues shown (BM, PB, Spl). Which timepoint where all these tissues taken? This is not specified in legend. Fig 5a timeline figure suggests liposomes

first injected on d14, with d28 and d42 'tumor burden monitoring' and a potential later non-specified time of 'sacrifice'. Can authors please precisely note the time at which the blood, BM and spleen tissues were analysed post liposome injection in Fig 5 d-i. . Similarly the exact dose of AraC loaded in liposomes or given as free agent should be provided at some stage in this manuscript. This reviewer has not been able to find this information in methods, figure legends or text.

The control therapeutic AraC dose administered to mice should also be a standard dose used in these type of mouse experiments (that is the maximum mice can tolerate without adverse consequences) to more closely mimic current practise in patients. Without knowing the positive control chemotherapy dose used, it is very hard to ascertain anticipated efficacy of liposome delivery compared to standard treatment.

Surprisingly in the single femur images provided in Fig 5 i. AraC (of unknown dose) administered alone does not appear to disrupt femur cellularity. Knowing the timepoint and dose of AraC used in these assays is absolutely necessary before any interpretation can be made.

Reviewer #2 (Remarks to the Author):

This work describes liposomes fused with the membrane of HSPCs to deliver a drug to the bone marrow in order to kill leukemia. CD44 is suggested to mediate enhanced homing to BM through enhanced levels of hyaluronic acid in Leukemia, and ITGB2 is proposed to bind ICAM-1 on liposomes. In general, this is an interesting study but details are severely lacking in order for me to fully evaluate the manuscript.

I have the following questions and comments:

1) Is up-regulation of HA in leukemia relevant to humans?

2) No idea what we are look at in Figure 2C (mislabeled as 1c). Description in the figure legend is poor. I had to go to the methods to figure it out. It is after incubating the HSPC-liposomes with leukemia cells. I still do not understand what the image demonstrates.

3) Use of CD44^{-/-} or ITGB2^{-/-} HSPCs would be more convincing that these molecules are critical to homing and targeting.

4) The details on how certain things are carried out are extremely minimal:

a) no description about how much fluorophore is loaded

b) no description about how the drug was loaded

c) ICG not defined.

d) Figure legend for Fig 3j needs much more.

5) What does the GFP in the BM represent? Is it the leukemia cells? There is nothing in the manuscript where this is described.

6) Poor word choice in many instances: 'Unleash' on L171. 'primitive' on L287

Reviewer #3 (Remarks to the Author):

In the study the authors develop new biomimetic vesicles, which are utilized for more localized drug delivery to the bone marrow of leukemia bearing mice. The authors use HSPC cell membranes fused to liposomes to enhance homing of the created vesicles into the marrow where leukemic stem cells reside and can avoid treatment leading to relapse. Using proteomics approach, the authors pinpoint Itgb2-Icam1 and CD44-HA as receptor-ligand pairs responsible for targeting leukemic cells with higher affinity, compared to the healthy cells. Finally, the authors perform in vivo experiments employing the HSCP-Lipo particles filled with Ara-C in leukemia mouse models and show improved elimination of leukemic cells and increased survival of tumor bearing mice.

This study addresses an important problem of "faulty" treatments for hematological malignancies offered currently to patients. The authors show improved results with their HSCP-Lipo vesicles in mouse leukemia models. However, the presented mechanism based on CD44 and Itgb2 is not sufficiently validated. Moreover, some results presented in the study could use strengthening and manuscript needs to address some issues that are currently not discussed.

Major comments:

1. The author claim that CD44-HA interactions mediate specific targeting of leukemic cells. However, CD44-HA is not the major axis responsible for homing (rather Cxcr4-Cxcl12) and the authors don't provide a convincing control to prove their point. Using cell membranes from CD44 deficient HSPC and showing reduced targeting of leukemic cells could be one of possible controls. The authors show

increased content of HA in leukemic mice. Nevertheless, this is not a convincing explanation considering overall high HA content in the ECM of healthy bone marrow and other connective tissues.

2. The affinity ratio in Fig. 3j should be compared to healthy HSC or at least LSK and not Lin+ cells. The authors show Icam1 MFI for Lin-, LSK and HSC in Fig. 3i (the highest Icam1 is on HSCs) and the goal is to target leukemic stem cells. Hence, the Lin+ cells are not appropriate control here. Moreover, Icam1 is highly expressed on bone marrow vasculature and stromal cells, it might be beneficial to address this issue in the manuscript text. Similarly to the comment above for the CD44, some type of Itgb2 blocking on HSPC-Lipo and showing reduced targeting of leukemic cells would be beneficial for proposed mechanism.

3. The discrepancy in cell viability (Fig. 4b) and level of cell death (Fig. 4d) is confusing. The authors show almost 0% viability for Ara-C@HSPC-Lipo compared to the other control groups, but the apoptosis is around 40%, which doesn't account for complete loss of viability. Also, how the authors can see increased differentiation if most of the cells are not viable (differentiated cells should be viable)? The authors should clarify.

4. The gating of LSC population in Fig. 4i is not convincing. The authors also don't comment on the B220 high population that completely disappears in Ara-C-Lipo and Ara-C@HSPC-Lipo.

5. It is surprising that there is no improvement in survival of mice in Ara-C group (green line) in Fig. 5c considering that other parameters such as frequency of tumors cells (Fig. 5d-g) and spleen weight (Fig. 5h) are improved by Ara-C only. The authors should address this discrepancy.

6. While the reviewer highly appreciates the biosafety studies presented in Fig. 7, the results for HSC frequency and absolute numbers are highly misleading (Fig. 7c and 7f). HSCs are not more abundant population compared to the Lin- or LSKs, they are only a fraction of Lin- and LSK populations. This is probably due to poor staining and gating of HSCs as presented in supplemental Fig. 3h. Similar situation regards absolute cell number of erythroid cells in the bone marrow.

Minor comments:

1. It is often not clear what leukemia model the authors use in their studies and the reader must look for this information in the methods of legends. Please clearly indicate the leukemia model in each appropriate figure (2d, 4i-j, 5).

2. The image in Fig. 2a clearly shows particles of vastly different sizes. However, the size quantification in Fig. 2b presents only one peak (not multiple peaks that would refer to different sizes from the image). The authors should comment on this. Also, what the DAPI signal in Fig. 2c represents? The authors use only cells membranes isolated from HSPCs. Please clarify.

3. Since the level of Itgb2 is quite different between cell membrane control and HSPC-Lipo sample (Fig. 2e), the presented western blot could benefit from a loading control or additional membrane protein which is important for homing process, such as Cxcr4.

4. The staining and gating of cell populations in supplemental figure 3h should be improved, the LSK and HSC plots/gates is not convincing.

5. The paragraph regarding developmental hematopoiesis in the discussion (lines 287-294) should be removed since it is not pertinent to the studies presented in the manuscript.

Point-by-point response to reviewers

Reviewer #1 (Remarks to the Author):

MAJOR COMMENTS.

1. Novelty.

One of the highlights of this manuscript is the use of haematopoietic progenitor (HSPC)-coated liposomes for cytarabine drug delivery compared to other conventional liposomes. However the use of leukocyte - membrane, eg monocyte/neutrophil or red cell membrane liposomes has become more common as these liposomes have longer in vivo half-life (eg. reviewed in PMID; 36103910) and are also reported to preferentially home across vasculature to inflamed tissues, an advantage that appears to be shared with the HSPC-coated liposomes in this manuscript. In fact leukocyte membrane-coated particles have already been shown to successfully target tumours and carry anti-cancer drugs to tumours in vivo (eg. NatNaotech 2013, PMID:2341654, or 2015 PMID:25883092).

Liposomes carrying chemotherapy agents are also already well established for treatment of malignancies clinically including leukaemia (eg. 2018 PMID:30024784 CTX-351). The main novelty in this manuscript seems to be the use of HSPC membranes for coating of chemotherapy containing particles instead of other leukocyte-membranes / particles. Indeed this may offer greater BM homing potential compared to other leukocyte-membrane coated particles however this was not tested. Together these prior publications somewhat diminishes the novelty of the findings.

Can the authors mention how their HSPC coated liposomes compare with other leukocyte membrane-coated liposomes and comparative advantage, such as in efficiency of homing to bone marrow or other?

Response: Thank you for the critical suggestions. Based on your constructive comments, to demonstrate the novelty and specificity of bone marrow targeting, we have conducted experiments from three aspects: A) Distribution advantages of different cell membranes; B) Advantages of cancer cell targeting; C) Drug loading advantages.

A) Distribution advantages of different cell membranes

Firstly, we compared the bone marrow homing ability of four different leukocyte membrane-coated liposomes with HSPC-Lipo. Specifically, we used four different sources of cell membranes, including mesenchymal stem cells, bone marrow-derived macrophages, neutrophils, and red blood cells, as controls. The results showed that compared with the other four different sources of cell membranes, the carrier prepared by HSPC cell membrane had the most significant bone marrow homing ability (**Fig. 2d-f**).

Fig. 2d-f. Bone marrow homing results of ICG fluorescent labeled different leukocyte membrane-coated liposomes in mice. **d** Bone marrow homing results of ICG fluorescent labeled different leukocyte membrane-coated liposomes in mice in a Ka539 model. Mice were subjected to *in vivo* imaging detection at different time intervals (2, 6, 12, 24 hours) after tail vein injection of different liposomes. $n=3$ biologically independent samples in each group. **e** The bone marrow homing ability of different leukocyte membrane-coated liposomes in the tibia and femur of mice. Mice were sacrificed at 24 h after tail vein injection of different liposomes, the tibia and femur were taken for *in vivo* imaging detection. $n=3$ biologically independent samples in each group. **f** Quantitative analysis of different liposomes in mouse tibia and femur. Data were presented as mean \pm s.d. ($n=3$ biologically independent samples). Significant differences of P values were calculated by two-tailed t-test and were indicated as * $P < 0.05$, ** $P < 0.01$ and *** $P < 0.001$.

B) Advantages of cancer cell targeting

Secondly, to test the specificity to target cancer cells, we used four different leukocyte membrane-coated liposomes as controls for experiments. Specifically, we prepared biomimetic carriers using cell membranes of mesenchymal stem cells (MSC), bone marrow-derived macrophages (BMDM), neutrophils (NEU), and red blood cells (RBC). Then, we tested the binding affinity of different leukocyte membrane infused liposomes to leukemic cells, and results showed that the HSPC cell membrane infused liposomes had the highest binding affinity for leukemic cells (**Fig. 3j**). These results also partially demonstrate the superiority and novelty of our study.

Fig. 3j. Detection of leukemic cell targeting ability of different leukocyte membrane-coated liposomes. Leukemic cells were harvested for detection at 0.5 hours after incubation with different liposomes. Data were presented as mean \pm s.d. (n=3 biologically independent samples). Significant differences of P values were calculated by two-tailed t-test and were indicated as * P < 0.05, ** P < 0.01 and *** P < 0.001.

C) Drug loading advantages

Thirdly, comparing with HSPC membrane vesicles, the HSPC-Lipo exhibited better drug loading efficiency of cytarabine (**Supplementary Table 1**). Due to the water-solubility, it is hard to load by the film dispersion method. Although it was possible to load cytarabine into HSPC membrane vesicles by room temperature incubation and pH gradient loading drug method, the encapsulation efficiency and drug loading capacity were too low to meet the demand for injection. Therefore, HSPC-Lipo had advantages of drug loading.

	Encapsulation efficiency (%)		Loading capacity (%)	
	HSPC-Lipo	HSPC membrane vesicle	HSPC-Lipo	HSPC membrane vesicle
Reverse-phase evaporation vesicle method	35.27 \pm 0.15	/	26.07 \pm 0.08	/
Film dispersion method	13.04 \pm 0.01	/	7.61 \pm 0.009	/
Incubation at room temperature method	10.24 \pm 0.005	0.96 \pm 0.004	7.20 \pm 0.003	0.65 \pm 0.004
pH gradient drug loading method	11.10 \pm 0.007	1.30 \pm 0.003	9.31 \pm 0.004	0.87 \pm 0.002

Supplementary Table 1. Encapsulation efficiency and loading capacity of HSPC-Lipo and HSPC membrane vesicles. The Ara-C was loaded by different methods including reverse-phase evaporation vesicle method, film dispersion method, incubation at room temperature method and pH gradient drug loading method.

OTHER COMMENTS.

STUDIES RESTRICTED TO TWO LYMPHOBLASTIC CELL LINES.

Use of two lymphoblastic cell lines (one established and cultured since 1973).

All studies in this manuscript were performed using two mouse lymphoblastic leukaemia cell lines (one established in 1973 _for all in vitro and in vivo assays. These two cell lines are;

-B lymphoma cell line (Ka539) and

-lymphoblastic cell line (C1498) originally isolated from a mouse with acute myeloid leukaemia. The cell line itself has lymphoblastic features.

The concern is these cell lines may not be very representative of 'leukaemia' in patients or mice.

For example several leukaemias (eg. AML) contain relatively rare leukaemia stems. It is these leukaemia stem cells that survive chemotherapy and re-establish relapse, not the bulk tumour cell population (which are relatively susceptible to chemotherapy). Long established and cultured cell lines may no longer contain any rare leukaemia stem cell populations. Also the assays used in these experiments are not designed to consider impact of therapy on leukaemia stem cell populations after therapy compared to bulk tumour.

Furthermore both these two leukaemia cell lines share lymphoblastic features. Ideally a range of 'freshly derived' leukaemias would be tested (lymphoid and myeloid). Ideally the driving oncogenes/changes are diverse and known. A more naturally-forming leukaemia would be welcome. Many are now widely available (such as using mice transgenic for specific leukaemia oncogenes, or at least oncogene driven leukaemia eg. using methodologies first described in PMID: 19339691). Human leukaemias in NSG mice another option.

Response: We appreciate the critical and insightful comments from the reviewer. According to the reviewer's suggestions, we have utilized a wildly-used and more naturally-forming MLL-AF9 leukemia mouse model for the *in vivo* treatment experiments (**Fig. 5**). The results further validate that the Ara-C@HSPC-Lipo showed good anti-leukemia effect *in vivo*. We have now added these data in the revised manuscript, which makes our conclusion more robust.

Fig. 5 | The anti-leukemic effect of Ara-C@HSPC-Lipo in MLL-AF9 leukemia mouse model. **a** Schematic diagram of animal experiment setup. **b** Kaplan-Meier survival curve of the leukemic mice received different treatments. (n=6 biologically independent samples for control, HSPC-Lipo and Ara-C group, and n=7 biologically independent samples for Ara-C@Lipo group, and n=11 biologically independent samples for Ara-C@HSPC-Lipo group). **c** The proportion of leukemia cells (GFP positive cells) in bone marrow. Leukemic mice were euthanized on the day 21 to collect bone marrow cells for flow cytometry analysis. **d** Quantitative analysis of flow cytometry results in Fig. 5c. Data were presented as mean ± s.d. (n=5 biologically independent samples). **e** The proportion of leukemia stem cells (CD34+CD16/32+) in bone marrow after different treatments. **f** Quantitative analysis of flow

cytometry results of leukemia stem cells. **g** The proportion of leukemic cells (GFP positive cells) in spleen. **h** Quantitative analysis of leukemic cells in spleen. Data were presented as mean \pm s.d. (n=5 biologically independent samples). **i** The proportion of leukemia cells (GFP positive cells) in peripheral blood. Data were presented as mean \pm s.d. (n=5 biologically independent samples). **j** The morphology of the spleen in leukemic mice after different treatment. **k** Weight of Spleen. Data were presented as mean \pm s.d. (n=5 biologically independent samples). Significant differences of P values were calculated by two-tailed, unpaired t-test and were indicated as * P < 0.05, ** P < 0.01 and *** P < 0.001.

FIGURE COMMENTS.

1. Fig 2c. The authors use DiD staining (labels hydrophobic / lipophilic structures) to ascertain if liposomes were coated with Hematopoietic progenitor membrane. As liposomes themselves are amphipathic (have both hydrophilic and hydrophobic parts), it would be appropriate for the authors to include control (liposome only) staining as well. this is needed to confirm the 'HSPC membrane' shown in Fig 1c is only labelling cell membranes, not also liposomes.

Response: We appreciate reviewer's suggestions. To avoid cross staining of two components with hydrophobic dyes, we have utilized anti-CD44 antibody (Boster, A00052, dilution ratio: 1:500) and secondary antibodies (Cy3-labeled Goat Anti-Rabbit IgG (H+L)) to label the HSPC membrane (represent in green). The CD44 expressed on the HSPC membrane but not on the liposomes, so this labeling has specificity for HSPC membranes. And the liposomes were labeled by DiD dye. As it showed that, green and red fluorescence signals have achieved co-localization both HSPC-Lipo alone (**Supplementary Fig. 2a**) and when combined with leukemia cells (**Supplementary Fig. 2b**). The Pearson's R value represents the strength and direction of the relationship between two variables. The value is greater than 0.5 illustrated high colocalization of two variables. Therefore, we calculated the Pearson's R value of HSPC membrane (green) and liposomes (red) and it was 0.79, which represented strong colocalization of two fluorescence signals.

Supplementary Figure 2. The colocalization of HSPC membrane and Liposomes. (a) HSPC membrane was labeled by anti-CD44 antibody and secondary antibodies (Cy3-labeled Goat Anti-Rabbit IgG (H+L)) (represent in green). Liposomes were labeled by DiD dye at 25 $\mu\text{g mL}^{-1}$ (represent in red). Scale bar, 30 μm . (b) Colocalization of HSPC-Lipo labeled by anti-CD44 antibody when combined with leukemia cells. Scale bar, 10 μm . The Pearson's R value was calculated by Fuji image J (version 153 with java 8).

In line 147-154: We have made revisions in the manuscript as “*To avoid cross staining of two components with hydrophobic dyes, we utilized anti-CD44 antibody and secondary antibodies (Cy3-labeled Goat Anti-Rabbit IgG (H+L)) to label the HSPC membrane (represent in green). The CD44 expressed on the HSPC membrane but not on the liposomes, so this labeling has specificity for HSPC membranes. And the liposomes were labeled by DiD dye. As it showed that, green and red fluorescence signals have achieved co-localization both HSPC-Lipo alone (Supplementary Fig. 2a) and when combined with leukemia cells (Supplementary Fig. 2b)*”.

2. Fig 2e. Femurs and tibias shown. Are these both from same mouse (ie. 1 mouse shown per treatment group?). If so how was this mouse chosen. How many mice per group per experiment? Legend states experiment was repeated 3 times. Does this mean 3 separate times with $n=1$ mice per group or a total of $n=3$ different mice once?

Response: We apologize that it is not clearly described in initial manuscript. The tibia and femur shown as pairs in the **Fig. 2e** were from the same mouse. In this study, each experimental group had 3 biologically different mice as one experiment. Besides, we have repeated this experiment for another two times, and we only chose one experiment as representative images. We have now made the corresponding changes in the figure legend of the revised manuscript.

3. Fig 2e,g. Please specify timepoint post injection that these organs were collected. Please label the organs shown. In Fig 1g, it is not clear if these images are from control mice or mice with leukemic cells. Please specify. Could the authors also include their data for the other group (leukemic or non-leukaemia mice) as well for comparison (of liposome location in various tissues).

Response: Thank you for your suggestion. Based on your constructive comments, we have made revisions in the manuscript and supplemented the distribution experiment in leukemic mice. These organs were collected 24 hours after injection. We have already labeled the organs in **Supplementary Fig. 3a and c**. These images were from normal mice. We also conducted the same experiment in leukemic mice. Similarly, in a leukemic mouse model, the enrichment of HSPC-Lipo in the bone marrow was significantly increased (**Supplementary Fig. 3c, d**). The detailed experimental results

and related descriptions were shown below and high lighted in red of revised manuscript.

Supplementary Figure 3. The distribution of HSPC-Lipo in different organs after intravenous injection. (a) The distribution of HSPC-Lipo in different organs of normal mice. Mice were euthanized for analysis at 24 hours after ICG labeled HSPC-Lipo was injected into the tail vein of mice. **(b)** Quantitative analysis of fluorescence signals in different organs of normal mice. **(c)** The distribution of HSPC-Lipo in different organs of leukemic mice. Mice were euthanized for analysis at 24 hours after ICG labeled HSPC-Lipo was injected into the tail vein of mice. **(d)** Quantitative analysis of fluorescence signals in different organs of leukemic mice. Data were presented as mean \pm s.d. (n=3 biologically independent samples). Significant differences of P values were calculated by two-tailed t-test and were indicated as * $P < 0.05$, ** $P < 0.01$ and *** $P < 0.001$.

4. Fig 2f,h,i. The values shown in Fig 2f, h, i (3 dots) have remarkably small variation. In contrast the images of femurs shown in Fig 2e, show some variation between femurs, or between tibias within each group (or within each mouse, this is not specified). Could the data shown in Fig 2f (with minimal variation) actually be three technical readouts from same mouse, or does each dot represent distinct data from separate mice? Are the data shown in Fig 2f from the same mice as in Fig 2e or different mice?. Could authors explain how they actually represented data from three biologically different mice with

experiment performed 3 times in this figure. Improving the amount of detail provided in figure legend detail would greatly aid understanding of these points.

Response: Thank you for your suggestion. Each dot in **Fig. 2f** represents distinct data from separate mice, and the data in **Fig. 2f** were from the same mice in **Fig. 2e**. In this study, each experimental group had 3 biologically different mice as one experiment. Besides, we have repeated this experiment for another two times, and we only chose one experiment as representative images. We have now made the corresponding changes in the figure legend of the revised manuscript.

5. Figure 3.

Fig 3e. western blot Protein analysis. It is very difficult to understand relevance of what is being shown though schematic diagram shows comparison of 'HSPC cell membrane and HSPC-lipo'.

Response: Thank you for your suggestion. By further analyzing the proteins co-enriched by the HSPC cell membrane and HSPC-Lipo vesicles, we identified the membrane proteins CD44 and ITGB2 that have been reported to mediate cell homing and adhesion (**Fig. 3d**). We further verified the mass spectrometry data by performing western blot to demonstrate the two cell adhesion molecules CD44 and ITGB2 were expressed both exist on the HSPC cell membrane and HSPC-Lipo (**Fig. 3e**).

Fig. 3d,e. **d** Proteins that coexist in HSPC cell membrane and HSPC-Lipo that associated with the adhesion process. **e** Western blot verification of CD44 and ITGB2.

In line 197-200: We have made revisions in the manuscript as “We further verified the mass spectrometry data by performing western blot to demonstrate the two classical cell adhesion molecules CD44 and ITGB2 were both exists on the cell membrane and HSPC-Lipo (Fig. 3e)”.

Methods section; only give gives extremely brief description of HSPC membrane collection (“cells were repeatedly freeze-thawed for 3 times, centrifuged at 5000g for 10 min, and the cell membrane was obtained”). This super-quick method appears to

not be a standard, nor a very pure membrane preparation method. There is no mention of homogenisation, of washing membrane particles to remove soluble proteins, or removal of unbroken cells from lysate or solubilisation of membrane proteins at end. Considering the importance of these membrane preps to this manuscript. A clearer method for membrane preparation is warranted.

Response: Thanks for your helpful advices. The HSPC membrane isolation method was provided as follow and in the revised manuscript. The cell membrane isolation method originates from existing reports [1, 2]. The purified HSPCs underwent freezing and thawing repeatedly from -80°C to 37°C three times, followed by getting enucleated in a hand-held Dounce homogenizer (20 passes while on ice) to remove the cell organelles such as nucleus. Then, HSPC membrane were resuspended with a lysis buffer containing a mixture of 75 mM NaCl, 6 mM NaHCO₃ (Fisher Chemical), 1.5 mM KCl, 0.17 mM Na₂HPO₄ (Fisher Chemical), 0.5 mM MgCl₂ (Aladdin), 20 mM HEPES, 1 mM ethylenediaminetetraacetic acid (Fisher Chemical), and protease inhibitors (Thermo Scientific). After centrifugation at 3200 g for 5 min at 4 °C, the resulting supernatant was collected and centrifuged at 211000 g for 30 min for HSPC membrane collection. The collected the HSPC membrane was suspended and stored in distilled water at 4°C.

Reference:

1. Bahmani B, *et al.* Intratumoral immunotherapy using platelet-cloaked nanoparticles enhances antitumor immunity in solid tumors. *Nature communications* **12**, 1999 (2021).
2. Zhang Q, *et al.* Neutrophil membrane-coated nanoparticles inhibit synovial inflammation and alleviate joint damage in inflammatory arthritis. *Nature nanotechnology* **13**, 1182-1190 (2018).

Similarly method for Fig 3e, western blot verification” (of Fig 3a protein analysis). Methods section states; “ HSPC-Lipo and HSPC cell membranes were suspended with 100uL protein lysate and were collected into 1.5mL EP tubes... centrifuge at 4 degrees... add 5x loading buffer”. This method appears to be potentially also missing some key steps - otherwise both lysate groups shown in Fig 3e would be anticipated to look the same after western blotting.

Response: Thank you for your suggestion. Based on your constructive comments, we have provided a detailed description of the experimental steps in the methodology section of revised manuscript and displayed as following.

In line 574-593: The prepared HSPC cell membrane and HSPC-Lipo were lysed in a cold cell lysis buffer and protease inhibitor mixture for 10min. Centrifuge at 4 degrees 12000r for 10 minutes. The protein concentration of sample was determined using the BCA detection kit (BCA Protein Assay Kit, Beyotime) and prepared at a protein concentration of 600 µg/mL, and then add 5 × Load buffer solution and incubate at 99 °C for 10 minutes, and store at -80 °C. The experimental steps of western blotting are as

follows. All samples were loaded onto bis-tris protein concentrated gels in a volume of 20 μ L. The protein electrophoresis condition was 60 volts constant pressure for 90 minutes. The PVDF membrane were used for membrane transfer under the condition of 300 mA for 90 minutes. 5% skimmed milk was used for blocking at room temperature for 1h. Then wash with TBST three times, each time for 10 minutes. The incubation condition for the primary antibody were at 4 °C overnight at a dilution ratio of 1:1000 (CD44, Boster, A00052; ITGB2, Beyotime, AF6399; CXCR4, Beyotime, AF6621; GAPDH, Sangon Biotech, D190090-0100). The dilution ratio of secondary antibody (Goat Anti-Mouse IgG(H+L) (peroxidase/HRP conjugated), Elabscience, E-AB-1001; Goat Anti-Rabbit IgG(H+L) (peroxidase/HRP conjugated), Elabscience, E-AB-1003) was 1:5000 at room temperature for about 1 hour. The ECL imaging and figures were collected by the Bio-Rad GelDoc Go with automatic exposure time.

Fig4b. legend does not mention length of culture results making these data very hard to interpret.

Response: Thank you for your suggestion. We have provided a more detailed description in the revised figure legend on the issue of incubation time *in vitro* experiments. In the revised version, the incubation time of cell viability is 48 hours (**Fig. 4b**). For apoptosis experiments, the incubation time is 48 hours (**Fig. 4c, d**). For cell differentiation experiments, the incubation time is 48 hours (**Fig. 4g, h**).

Fig 4c. Please be aware that PI+ AnnexV+ staining may be either due to apoptosis or pyroptosis in these cells. It is not a specific stain for apoptosis only.

Also the % differences on axis are too small to read. Please make figure text legible. More importantly there doesn't appear to be the same number of gated cells within each group. Specifically there seems to be more cells shown in A@HSPC-Lipo group compared to others. This certainly makes it appear to have more dead cells. However these data should be shown without bias. That is with same number of cells loaded per group and clearly described method on how long these cultures were performed.

Response: Thank you for your suggestion. We agree that PI+ AnnexV+ staining may be either due to apoptosis or pyroptosis in these cells, and have changed the y-axis from "Apoptosis rate" to "Annexin V+ cells" to avoid any misunderstanding. Besides, we have enlarged the axis labels to make the figure text legible. Moreover, according to the reviewer's nice suggestion, we have collected the same number of cells in each group for the further analysis, which showed the same results as before (**Fig. 4c-d**). Finally, we have made changes in the revised manuscript and methods to make it clearer to the readers.

Fig. 4. c The proportion of Annexin V positive cells after 48 hours of different treatment by flow cytometry in Ka539 leukemia cells. **d** The proportion of Annexin V positive cells after 48 hours of different treatment by flow cytometry in C1498 leukemia cells. The leukemia cells were harvested at 48h after different treatments and displays the same number of cells in each flow cytometry result. Data were presented as mean \pm s.d. (n=3 biologically independent samples). Significant differences of P values were calculated by two-tailed, unpaired t-test and were indicated as * P < 0.05, ** P < 0.01 and *** P < 0.001.

Fig 4f,h,j. 'Drug-loaded HSPC-lipo significantly increased the differentiation of C1498 leukaemia cells. The histograms (Fig 4f,h,j) appear to have very little variation for "repeated 3 times independently" data. Could technical replicates have been accidentally shown. Could authors provide details on the data for each separate independent experiment and number of technical replicates performed within each.

Response: Thank you for your suggestion. In this study, we used three technical replicates in one experiment, and repeated the experiments in three independent experiments. We provide the data details of three experiments in the cell differentiation experiment as shown in the figure below. We performed the statistical analysis by calculating mean \pm s.d. and presented the representative data in the **Supplementary Fig. 7c, d**. We have made changes in the revised figure legends to make it clearer to the readers.

Fig 5 d-i. Proportion of tumour (GFP+) cells in various tissues shown (BM, PB, Spl). Which timepoint where all these tissues taken? This is not specified in legend. Fig 5a timeline figure suggests liposomes first injected on d14, with d28 and d42 'tumor burden monitoring' and a potential later non-specified time of 'sacrifice'. Can authors please precisely note the time at which the blood, BM and spleen tissues were analysed post liposome injection in Fig 5 d-i. . Similarly the exact dose of AraC loaded in liposomes or given as free agent should be provided at some stage in this manuscript. This reviewer has not been able to find this information in methods, figure legends or text.

Response: Thank you for your suggestion. Regarding the timing of tissue collection, we collected tissue for analysis on the 21th day after tumor inoculation. Mice were euthanized and their bone marrow, spleen, and peripheral blood were collected for analysis. For the dose of Ara-C injection, we previously used a dose of 50mg/kg and injected it through the tail vein of mice. To better understanding of the time points and experimental settings, we have plotted a more accurate experimental timeline (Supplementary Fig. 8a).

Supplementary Figure 8a. Schematic diagram of animal experiment setup.

The control therapeutic AraC dose administered to mice should also be a standard dose used in these type of mouse experiments (that is the maximum mice can tolerate without adverse consequences) to more closely mimic current practise in patients. Without knowing the positive control chemotherapy dose used, it is very hard to ascertain anticipated efficacy of liposome delivery compared to standard treatment.

Surprisingly in the single femur images provided in Fig 5 i. AraC (of unknown dose) administered alone does not appear to disrupt femur cellularity. Knowing the timepoint and dose of AraC used in these assays is absolutely necessary before any interpretation can be made.

Response: Thank you for your suggestion. We have provided more detailed description on the timing and dosage of chemotherapy drug Ara-C in the experimental methods section. Specifically, we previously used a dosage of 50 mg/kg for Ara-C in the leukemia animal model. For the injection time of Ara-C, we administered the first injection 14 days after tumor inoculation in mice, every 72 hours, for a total of 3 injections (Supplementary Fig. 8a).

In addition, we previously used a low-dosage of Ara-C (50 mg/kg) in treating the leukemia mice, and have now increased the dosage to 100 mg/kg for Ara-C in the control group and conducted animal experiments. The results showed that although the high dosage of free chemotherapy drugs could kill leukemia cells compared with the control HSPC-Lipo group, the Ara-C@HSPC-Lipo treatment groups showed the best therapeutic effects (**Fig. 6**).

Fig. 6 | The anti-leukemic effect of Ara-C@HSPC-Lipo in Ka539 leukemia mouse model.

a Schematic diagram of animal experiment setup. **b** The proportion of leukemia cells (GFP positive cells) in bone marrow. Leukemic mice were euthanized at day 28 to collect bone marrow cells for flow cytometry analysis. **c** Quantitative analysis of flow cytometry results in Fig. 6b. Data were presented as mean \pm s.d. (n=5 biologically independent samples). **d** The proportion of leukemic cells (GFP positive cells) in spleen. **e** Weight of Spleen. Data were presented as mean \pm s.d. (n=5 biologically independent samples). **f** Kaplan-Meier survival curve of the leukemic mice received different treatments. (n=6 biologically independent samples for control, HSPC-Lipo and Ara-C group, and n=8 biologically independent samples for Ara-C@Lipo, and n=12 biologically independent samples for Ara-C@HSPC-Lipo group). Significant differences of P values were calculated by two-tailed, unpaired t-test and were indicated as * P < 0.05, ** P < 0.01 and *** P < 0.001.

Reviewer #2 (Remarks to the Author):

This work describes liposomes fused with the membrane of HSPCs to deliver a drug to the bone marrow in order to kill leukemia. CD44 is suggested to mediate enhanced homing to BM through enhanced levels of hyaluronic acid in Leukemia, and ITGB2 is proposed to bind ICAM-1 on liposomes. In general, this is an interesting study but details are severely lacking in order for me to fully evaluate the manuscript.

Response: We thank the reviewer for his/her positive general remarks on our study. According to the reviewer's suggestions, we have made the corresponding changes in the revised manuscript.

I have the following questions and comments:

1) Is up-regulation of HA in leukemia relevant to humans?

Response: Thank you for your suggestion. Based on your constructive comments, we have added relevant literature support. According to the literature (Med Oncol (2010) 27:618-623), "*In this study the patients with malignant diseases, mostly haematological, demonstrated an increase of HA in bone marrow involved of disease and to a less extent in non-involved marrows.*" And the main conclusions of this literature well support our own experimental data.

Reference:

Sundstrom G, Hultdin M, Engstrom-Laurent A, Dahl IM. Bone marrow hyaluronan and reticulin in patients with malignant disorders. *Medical oncology* **27**, 618-623 (2010).

In line 202-204: We have made revisions in the manuscript as "*Previous study has documented that the content of hyaluronic acid is upregulated in the bone marrow of leukemia patients compared to their healthy control²¹*".

2) No idea what we are look at in Figure 2C (mislabeled as 1c). Description in the figure legend is poor. I had to go to the methods to figure it out. It is after incubating the HSPC-liposomes with leukemia cells. I still do not understand what the image demonstrates.

Response: We apologize for the lack of clarity in our previous manuscript. In the previous Fig. 2c, we aimed to verify the fusion structure of HSPC cell membrane (green, labeled with DiO) and liposome (red, labeled with DiD) by performing immunofluorescence co-localization experiment (**Fig. 2c**). In addition, based on the suggestions of the reviewer, we have improved the method of labeling HSPC cell membranes. The liposome was labeled with DiD (red color), the HSPC membrane was labeled with anti-CD44 antibody and secondary antibodies (Cy3-labeled Goat Anti-Rabbit IgG (H+L)) (represent in green), the nuclei was labelled by DAPI (blue color), and the co-localization region was shown in merged color (yellow color). We have

provided new images by splitting different colors, which showed that HSPC-Lipo maintained a complete structure when binding to leukemic cells, and liposome and HSPC cell membrane exhibited excellent co-localization (**Supplementary Fig. 2b**).

Fig. 2c. Immunofluorescence co-localization of HSPC cell membrane and liposome. HSPC cell membrane was labeled with DiO (green), liposome was labeled with DiD (red).

Supplementary Figure 2. The colocalization of HSPC membrane and Liposomes. (a) HSPC membrane was labeled by anti-CD44 antibody and secondary antibodies (Cy3-labeled Goat Anti-Rabbit IgG (H+L)) (represent in green). Liposomes were labeled by DiD dye at $25 \mu\text{g mL}^{-1}$ (represent in red). Scale bar, $30 \mu\text{m}$. (b) Colocalization of HSPC-Lipo labeled by anti-CD44 antibody when combined with leukemia cells. Scale bar, $10 \mu\text{m}$. The Pearson's R value was calculated by Fuji image J (version 153 with java 8).

3) Use of CD44^{-/-} or ITGB2^{-/-} HSPCs would be more convincing that these molecules are critical to homing and targeting.

Response: Thank you for your suggestion. Based on the reviewer's constructive comments, we initially constructed CD44 knockdown by inducing HSPC with lentiviruses carrying CD44 shRNA, then collected HSPC cell membranes and conducted *in vivo* distribution experiments in mice. The results showed that compared with the control group, the CD44 knockdown significantly reduced the bone marrow targeting ability of HSPC-Lipo (**Fig. 3h-i**). In addition, we also constructed ITGB2 knockdown in HSPC cells, and found that compared with the control group, the carrier prepared using ITGB2 knockdown cell membranes significantly reduced its ability to target leukemic cells. (**Fig. 3n-o**).

Fig. 3h-i. Evaluation of the bone marrow targeting ability of HSPC-Lipo and CD44 knockdown HSPC-Lipo. **h** *In vivo* imaging detection of the bone marrow targeting ability of CD44 knockdown HSPC-Lipo at 24 hours after tail vein injection. (n=3 biologically independent samples). **i** Quantitative analysis results of *in vivo* bone marrow targeted imaging analysis. Data were presented as mean \pm s.d. (n=3 biologically independent samples). Significant differences of P values were calculated by two-tailed, unpaired t-test and were indicated as * P < 0.05, ** P < 0.01 and *** P < 0.001.

Fig. 3n-o. Evaluation of the cancer cell affinity ratio of normal HSPC-Lipo and ITGB2 knockdown HSPC-Lipo. **n** Flow cytometry detection of the leukemic cells targeting ability of ITGB2 knockdown HSPC-Lipo. Leukemic cells were harvested for detection 0.5 hours after incubation with different liposomes. **o** Quantitative analysis of the leukemic cells targeting ability of ITGB2 knockdown HSPC-Lipo. Data were presented as mean \pm s.d. (n=3 biologically

independent samples). Significant differences of P values were calculated by two-tailed t-test and were indicated as * $P < 0.05$, ** $P < 0.01$ and *** $P < 0.001$.

4) *The details on how certain things are carried out are extremely minimal:*

a) *no description about how much flurophore is loaded*

b) *no description about how the drug was loaded*

c) *ICG not defined.*

d) *Figure legend for Fig 3j needs much more.*

5) *What does the GFP in the BM represent? Is it the leukemia cells? There is nothing in the manuscript where this is described.*

6) *Poor word choice in many instances: 'Unleash' on L171. 'primitive' on L287*

a) *no description about how much flurophore is loaded*

Response: We apologize for the lack of clarity in our previous manuscript. Since DiD (Beyotime Biotech) was hydrophobic dye, the thin film dispersion method was adopted for DiD dye loading. In brief, the dye-carrying films were prepared by weighing 50, 125, 250, 500, 750, and 1000 μg of DiD dye with 20 mg of lecithin (Aladdin) dissolved in anhydrous ethanol, and rotarily evaporated the solvent at 35 $^{\circ}\text{C}$, 120 rpm. The suspension containing HSPC membranes was added and hydrated, and then cell membrane fusion was performed by probe sonication (60 w) for 6 min. After ultracentrifuge centrifugation at 4 $^{\circ}\text{C}$, 150,000 g conditions for 3 h, we separated the free dye from the loaded drug and removed the supernatant. The precipitate of the HSPC-Lipo carrying DiD dye was then suspended into 2 mL of PBS buffer. As **Supplementary Fig. 7a** displayed, 20 mg of phospholipids carried about 400 μg of DiD dye, which was much higher than the 50 μg ($25 \mu\text{g mL}^{-1}$) we used for labeling. Further, we examined DiD encapsulation rates at different time points after carryover, and found that the HSPC-Lipo could carry DiD stably for more than 72 hours (**Supplementary Fig. 7b**). We have made changes in the revised Methods to make it clearer to the readers.

a

Supplementary Figure 7a. DiD's encapsulation rate at different DiD weights of 50, 125, 250, 500, 750, and 1000 μg . Data represent the mean \pm s.d. (n=3 biologically independent samples).

Supplementary Figure 7b. Encapsulation rate of DiD at different points in time including 0, 6, 12, 24, 48 and 72 h after loading. Data represent the mean \pm s.d. (n=3 biologically independent samples).

b) no description about how the drug was loaded

Response: We apologize for the lack of clarity in our previous manuscript. Due to the water-solubility of cytarabine, HSPC-Lipo carrying cytarabine was prepared by reverse evaporation method. 30 mg of cytarabine (MCE) was dissolved in 1 mL of ultrapure water, 20 mg of lecithin (Aladdin) was dissolved in 3 mL of ether, and then the microemulsion was prepared by ultrasonication. The solvent was evaporated at room temperature until the solution was viscous when PBS buffer was added, and the residual ether was removed by rotary evaporator to obtain the drug-carrying liposome. Suspension containing HSPC membranes was added and cell membrane fusion was performed by probe sonication (60 w) for 6 min. After ultracentrifuge centrifugation at 4 °C, 150,000 g conditions for 3 h, we separated the free dye from the loaded drug and removed the supernatant. The precipitate of the HSPC-Lipo carrying DiD dye was then suspended into 2 mL of PBS buffer for experiments. We have made changes in the revised Methods to make it clearer to the readers.

c) ICG not defined.

Response: We apologize for the lack of clarity in our previous manuscript. ICG refers to indocyaninegreen, a near-infrared dye that penetrates skin and muscle and reacts to the *in vivo* distribution of the loading carriers. In our study, ICG (Aladdin) was loaded into HSPC-Lipo for *in vivo* biodistribution detection. We have added the description in the revised Methods to make it clearer to the readers.

d) Figure legend for Fig 3j needs much more.

Response: Thanks for your suggestion. We have provided a more detailed description of the annotation of Fig. 3j in the revised version. Specifically, we labeled the HSPC-Lipo and different leukocyte membrane-coated liposomes with DiD fluorescence and

incubated them with leukemic cells and HSPC cells to evaluate their binding efficiency to leukemic cells and normal cells (**Fig. 3j**).

In previous versions of Fig. 3j, we demonstrated the specificity of HSPC-Lipo targeting leukemia cells through ITGB2 through antibody blockade. Based on your critical suggestions, in the revised version, we conducted more rigorous experiments to verify the specificity of HSPC-Lipo binding to leukemic cells. We further constructed ITGB2 knockdown HSPC cell membranes using shRNA in HSPC cell lines (32D, mouse hematopoietic stem progenitor cell line). Compared with the control group, the group using ITGB2 knockdown HSPC cell membranes showed significantly reduced targeting ability to leukemic cells (**Fig. 3n-o**). The detailed experimental results and related descriptions were shown below and high lighted in red of revised manuscript.

Figure 3n-o. Flow cytometry detection of the leukemic cells targeting ability of ITGB2 knockdown HSPC-Lipo. **n** Flow cytometry detection of the leukemic cells targeting ability of ITGB2 knockdown HSPC-Lipo. Leukemic cells were harvested for detection 0.5 hours after incubation with different liposomes. **o** Quantitative analysis of the leukemic cells targeting ability of ITGB2 knockdown HSPC-Lipo. Data were presented as mean \pm s.d. (n=3 biologically independent samples). Significant differences of P values were calculated by two-tailed t-test and were indicated as * P < 0.05, ** P < 0.01 and *** P < 0.001.

e) *What does the GFP in the BM represent? Is it the leukemia cells? There is nothing in the manuscript where this is described.*

Response: We apologize for the lack of clarity in our previous manuscript. In our study, murine Ka539 leukemia cells were infected with lentiviruses carrying GFP and used for *in vivo* transplantation. GFP positive cells represented tumor cells. We have made corresponding annotations in the experimental methods and revised manuscript.

f) *Poor word choice in many instances: ‘Unleash’ on L171. ‘primitive’ on L287*

Response: We apologize for the lack of clarity in our previous manuscript. We have changed “Unleash” to “investigate” in Line 185 in the revised manuscript and highlight it in red. Regarding the word “primitive”, we have deleted that paragraph in the revised manuscript according to the reviewer #3’s suggestion.

Reviewer #3 (Remarks to the Author):

In the study the authors develop new biomimetic vesicles, which are utilized for more localized drug delivery to the bone marrow of leukemia bearing mice. The authors use HSPC cell membranes fused to liposomes to enhance homing of the created vesicles into the marrow where leukemic stem cells reside and can avoid treatment leading to relapse. Using proteomics approach, the authors pinpoint Itgb2-Icam1 and CD44-HA as receptor-ligand pairs responsible for targeting leukemic cells with higher affinity, compared to the healthy cells. Finally, the authors perform in vivo experiments employing the HSCP-Lipo particles filled with Ara-C in leukemia mouse models and show improved elimination of leukemic cells and increased survival of tumor bearing mice.

This study addresses an important problem of “faulty” treatments for hematological malignancies offered currently to patients. The authors show improved results with their HSCP-Lipo vesicles in mouse leukemia models. However, the presented mechanism based on CD44 and Itgb2 is not sufficiently validated. Moreover, some results presented in the study could use strengthening and manuscript needs to address some issues that are currently not discussed.

Response: We appreciate the positive and insightful comments of reviewer. The revised manuscript is supplemented with further pertinent experimental data. We address these concerns below and also made corresponding changes in the revised manuscript.

Major comments:

1. The author claim that CD44-HA interactions mediate specific targeting of leukemic cells. However, CD44-HA is not the major axis responsible for homing (rather Cxcr4-Cxcl12) and the authors don't provide a convincing control to prove their point. Using cell membranes from CD44 deficient HSPC and showing reduced targeting of leukemic cells could be one of possible controls. The authors show increased content of HA in leukemic mice. Nevertheless, this is not a convincing explanation considering overall high HA content in the ECM of healthy bone marrow and other connective tissues.

Response: We agree with the reviewer and now provide additional evidences to address these concerns, which greatly help improve the quality of the manuscript.

(1) Firstly, we agree that CD44-HA is not the major axis responsible for homing and we did find that CXCR4 was expressed in the HSPC membrane and HSCP-Lipo (**Supplementary Fig. 4c**). However, the action of Cxcr4-Cxcl12 requires the phosphorylation of CXCR4 and activation of downstream signaling, which were absent in the HSPC membrane liposomes. Rather, CD44 is an adhesion molecular and could directly interacts with the bone marrow microenvironment. According to the literature (G Sundström et al, Med Oncol (2010) 27:618-623) in human patients samples as well as our own data (**Fig. 3f**), the content of hyaluronic acid is upregulated in the bone marrow of leukemic mice and human patients compared to their healthy control.

Supplementary Figure 4c. Western blot experiments to validate mass spectrometry data.

Fig. 3f. Determination of hyaluronic acid content in mouse bone marrow. Data were presented as mean \pm s.d. ($n=3$ biologically independent samples). Significant differences of P values were calculated by two-tailed, unpaired t-test and were indicated as * $P < 0.05$, ** $P < 0.01$ and *** $P < 0.001$.

(2) Secondly, based on the reviewer's constructive comments, we constructed CD44 knockdown by inducing HSPC with lentiviruses carrying with CD44 shRNA, then collected HSPC cell membranes and conducted *in vivo* distribution experiments in mice. The results showed that compared with the control group, the CD44 knockdown significantly reduced the bone marrow targeting ability of HSPC-Lipo (**Fig. 3h-i**).

Fig. 3h-i. Evaluation of the bone marrow targeting ability of HSPC-Lipo and CD44 knockdown HSPC-Lipo. **h** *In vivo* imaging detection of the bone marrow targeting ability of CD44 knockdown HSPC-Lipo at 24 hours after tail vein injection. (n=3 biologically independent samples). **i** Quantitative analysis results of *in vivo* bone marrow targeted imaging analysis. Data were presented as mean \pm s.d. (n=3 biologically independent samples). Significant differences of P values were calculated by two-tailed, unpaired t-test and were indicated as * P < 0.05, ** P < 0.01 and *** P < 0.001.

2. The affinity ratio in Fig. 3j should be compared to healthy HSC or at least LSK and not Lin⁺ cells. The authors show *Icam1* MFI for Lin⁻, LSK and HSC in Fig. 3i (the highest *Icam1* is on HSCs) and the goal is to target leukemic stem cells. Hence, the Lin⁺ cells are not appropriate control here. Moreover, *Icam1* is highly expressed on bone marrow vasculature and stromal cells, it might be beneficial to address this issue in the manuscript text. Similarly to the comment above for the CD44, some type of *Itgb2* blocking on HSPC-Lipo and showing reduced targeting of leukemic cells would be beneficial for proposed mechanism.

Response: We appreciate the critical and insightful comments from the reviewer.

(1) According to the reviewer's suggestion, we have tested the affinity of HSPC-Lipo compared to normal HSPC cells. We used murine HSPC cells as the control (32D, a mouse hematopoietic stem progenitor cell line), and found that the affinity of HSPC-Lipo to normal HSPC cells was significantly lower than that of leukemia cells (**Fig. 3j**), which indicates the advantage of HSPC-Lipo.

Fig. 3j. Detection of leukemic cell targeting ability of different leukocyte membrane-coated liposomes. Leukemic cells were harvested for detection at 0.5 hours after incubation with different liposomes. Data were presented as mean \pm s.d. (n=3 biologically independent samples). Significant differences of P values were calculated by two-tailed t-test and were indicated as * P < 0.05, ** P < 0.01 and *** P < 0.001.

(2) We agree with the reviewer that ICAM-1 is highly expressed in bone marrow stroma and vascular cells. In our study, we propose that ICAM-1 is highly expressed in leukemic cells and interacts with ITGB2 on HSPC membrane liposome (**Supplementary Fig. 6a-g**), whereas CD44 on HSPC-Lipo mediates the specific targeting to the bone marrow of leukemic mice by interacting with hyaluronic acid. To validate the underlying mechanism of targeting leukemic cells, we further constructed ITGB2 knockdown HSPC cell membranes using shRNA in HSPC cell lines (32D, mouse hematopoietic stem progenitor cell line). The results showed that, compared with the control group, the group using ITGB2 knockdown HSPC cell membranes showed significantly reduced affinity with leukemic cells (**Fig. 3n and o**).

Supplementary Fig. 6. Expression profile of ICAM-1 in leukemia cell lines and HSPC cells. (a-f) Expression of ICAM-1 in human leukemia cell lines. NB4 (a), OCI-AML-2 (b), OCI-

AML-3 (c), SKM-1 (d), HL-60 (e). (f) Quantitative analysis of frequency of ICAM-1⁺ cells in Supplementary Fig. 6a-e. (g) Bioinformatics analysis of ICAM-1 expression (<http://www.bloodspot.eu>). Data were presented as mean ± s.d. (n=3 biologically independent samples). Significant differences of P values were calculated by two-tailed t-test and were indicated as * P < 0.05, ** P < 0.01 and *** P < 0.001.

Figure 3n-o. Flow cytometry detection of the leukemic cells targeting ability of ITGB2 knockdown HSPC-Lipo. n Flow cytometry detection of the leukemic cells targeting ability of ITGB2 knockdown HSPC-Lipo. Leukemic cells were harvested for detection 0.5 hours after incubation with different liposomes. o Quantitative analysis of the leukemic cells targeting ability of ITGB2 knockdown HSPC-Lipo. Data were presented as mean ± s.d. (n=3 biologically independent samples). Significant differences of P values were calculated by two-tailed t-test and were indicated as * P < 0.05, ** P < 0.01 and *** P < 0.001.

3. The discrepancy in cell viability (Fig. 4b) and level of cell death (Fig. 4d) is confusing. The authors show almost 0% viability for Ara-C@HSPC-Lipo compared to the other control groups, but the apoptosis is around 40%, which doesn't account for complete loss of viability. Also, how the authors can see increased differentiation if most of the cells are not viable (differentiated cells should be viable)? The authors should clarify.

Response: We thank the reviewer for pointing out these inconsistent data. Regarding the question about cell viability and apoptosis, we have checked the details of our *in vitro* experiments. The cell viability was detected at 96 hours after drug treatment previously, while apoptosis and differentiation were detected at 72 hours after drug treatment. We reckon that might be the reason for the inconsistency. In addition, in order to make the *in vitro* experimental data more consistent, we conducted a unified cell viability experiment at 48 hours after different treatments, and the experimental results were shown in the Fig. 4b. Thus, we have made changes in the revised figure legends to make it clearer to the readers.

Fig. 4b. Cell viability of leukemic cells after different treatments at 48 hours *in vitro*. Data were presented as mean \pm s.d. (n=3 biologically independent samples). Significant differences of P values were calculated by two-tailed, unpaired t-test and were indicated as * P < 0.05, ** P < 0.01 and *** P < 0.001.

4. The gating of LSC population in Fig. 4i is not convincing. The authors also don't comment on the B220 high population that completely disappears in Ara-C-Lipo and Ara-C@HSPC-Lipo.

Response: We appreciate these important critiques. Based on your suggestion, we have conducted further gating analysis on this B220 high population cells, and the results were shown below. As shown in the following figure, this group of B220 high cells (red boxes in Fig. a) was lineage negative (red dots in Fig. b) and highly expressed the cell marker B220 (red dots in Fig. c). In addition, we have reviewed relevant literature that reported B220 as a marker for leukemia stem cells and used it as a therapeutic target (PMID: 29343865, 2018; PMID: 28525262, 2017), which is consistent with our results that the frequency of leukemia stem cells were significantly decreased after Ara-C@HSPC-Lipo treatment (**Supplementary Fig. 7g, h**).

Reference:

Mandal T, Beck M, Kirsten N, Linden M, Buske C. Targeting murine leukemic stem cells by antibody functionalized mesoporous silica nanoparticles. *Scientific reports* **8**, 989 (2018).

Beck M, Mandal T, Buske C, Linden M. Serum Protein Adsorption Enhances Active Leukemia Stem Cell Targeting of Mesoporous Silica Nanoparticles. *ACS applied materials & interfaces* **9**, 18566-18574 (2017).

Figure. Gating analysis on this B220 high population cells.

Supplementary Figure 7. (g) The frequency of leukemia stem cells after different treatments at 72h in Ka539 cells flow cytometry analysis. **(h)** Quantitative analysis of frequency of leukemia stem cells in Ka539 cells. Data were presented as mean \pm s.d. (n=3 technical replicates). Significant differences of P values were calculated by two-tailed, unpaired t-test and were indicated as * P < 0.05, ** P < 0.01 and *** P < 0.001.

5. It is surprising that there is no improvement in survival of mice in Ara-C group (green line) in Fig. 5c considering that other parameters such as frequency of tumors cells (Fig. 5d-g) and spleen weight (Fig. 5h) are improved by Ara-C only. The authors should address this discrepancy.

Response: Thank you for your suggestion. Our data showed that Ara-C injection had therapeutic effect at certain extent, but did not significantly improve the survival of leukemia mice. We reckon that the dosage of Ara-C was too low to prolong the survival of leukemic mice (previously used 50 mg/kg). To address this discrepancy, we have increased the dosage of chemotherapy drugs to 100 mg/kg for Ara-C group and repeated the animal experiment in the Ka539 leukemia model (**Fig. 6a**). In the added animal experimental results, the increased dose of free chemotherapy drugs significantly improved the survival of leukemic mice compared with the control and HSPC-Lipo group, and the Ara-C@HSPC-Lipo group showed the longest survival period and the best therapeutic effects (**Fig. 6f**).

Fig. 6 | The anti-leukemic effect of Ara-C@HSPC-Lipo in Ka539 leukemia mouse model. **a** Schematic diagram of animal experiment setup. **b** The proportion of leukemia cells (GFP positive cells) in bone marrow. Leukemic mice were euthanized at day 28 to collect bone marrow cells for flow cytometry analysis. **c** Quantitative analysis of flow cytometry results in Fig. 6b. Data were presented as mean \pm s.d. (n=5 biologically independent samples). **d** The proportion of leukemia cells (GFP positive cells) in spleen. **e** Weight of Spleen. Data were presented as mean \pm s.d. (n=5 biologically independent samples). **f** Kaplan-Meier survival curve of the leukemic mice received different treatments. (n=6 biologically independent samples for control, HSPC-Lipo and Ara-C group, and n=8 biologically independent samples for Ara-C@Lipo, and n=12 biologically independent samples for Ara-C@HSPC-Lipo group). Significant differences of P values were calculated by two-tailed, unpaired t-test and were indicated as * $P < 0.05$, ** $P < 0.01$ and *** $P < 0.001$.

6. While the reviewer highly appreciates the biosafety studies presented in Fig. 7, the results for HSC frequency and absolute numbers are highly misleading (Fig. 7c and 7f). HSCs are not more abundant population compared to the Lin- or LSKs, they are only a fraction of Lin- and LSK populations. This is probably due to poor staining and gating of HSCs as presented in supplemental Fig 3h. Similar situation regards absolute cell number of erythroid cells in the bone marrow.

Response: Thank you for raising the crucial question. Based on your suggestion, we have conducted more rigorous gating (Supplementary Fig. 6i) and statistic analysis on the data, and the results were shown in the figure below (Fig. 8c-h). As for the issue of the proportion of HSCs, we previously calculated the proportion of HSCs in the parent group, in the revised version, we have recalculated the proportion of HSCs in the total nucleated cells (TNC) (Fig. 8c-e). Regarding the absolute number of erythroid cells in the bone marrow, we counted the enucleated erythroid cells and the results were shown in the following figure (Fig. 8h).

Supplementary Figure 6. (i) The gating strategy of HSPC in flow cytometry analysis.

Fig. 8 | Safety evaluation and analysis of HSPC-Lipo. **a** Schematic diagram of animal experiment setup. **b** Body weight of mice in different treatment groups. **c** The frequency of hematopoietic stem progenitor cells in TNC. **d** The frequency of progenitor cells in TNC. **e** The frequency of lineage cells in TNC. **f** The absolute number of hematopoietic stem progenitor

cells per legs of tibia and femur. **g** The absolute number of progenitor cells per legs of tibia and femur. **h** The absolute number of lineage cells per legs of tibia and femur. Data represent the mean \pm s.d. (n=6 biologically independent samples for each group. Significant differences of P values were calculated by two-tailed t-test and were indicated as * P < 0.05, * * P < 0.01 and * * * P < 0.001.

Minor comments:

1. It is often not clear what leukemia model the authors use in their studies and the reader must look for this information in the methods of legends. Please clearly indicate the leukemia model in each appropriate figure (2d, 4i-j, 5).

Response: Thank you for your suggestion. Based on your constructive comments regarding the indication of leukemia models, we have provided detailed annotations in the figures and figure legends in the revised version. Specifically, in the experiment shown in **Fig. 2d**, we used a mouse Ka539 transplantation model. In **Supplementary Fig. 7g, h** (previously Fig. 4i, j), the leukemic cells were mouse Ka539 cells. In the animal experiment shown in **Supplementary Fig. 8** (previously Fig. 5), we used a mouse Ka539 transplantation model. In the revised version, we have made corresponding annotations in the figure legends.

2. *The image in Fig. 2a clearly shows particles of vastly different sizes. However, the size quantification in Fig. 2b presents only one peak (not multiple peaks that would refer to different sizes from the image). The authors should comment on this. Also, what the DAPI signal in Fig. 2c represents? The authors use only cells membranes isolated from HSPCs. Please clarify.*

Response: Thank you for your advices.

(1) The plot in Fig. 2b was a fitted curve and showed the intensity of distribution in the different particle size intervals. According to the different intensity of distribution in the different particle size intervals, we provided the intensity means and curve of different particle size distribution intervals as **Fig. 2b** displayed. It showed that the majority of HSPC-Lipo was distributed in the range of 150-180 nm, which was consistent with the TEM results of **Fig. 2a**.

Fig. 2. a Transmission electron microscope results of HSPC-Lipo. **b** Average particle size distribution of HSPC-Lipo. Measured by Dynamic Light Scattering.

(2) The DAPI signal represents the nuclei of the leukemic cell (C1498). For the convenience of reading, we have revised the description in Fig. 2c and the revised manuscript. In addition, to avoid cross staining of two components with hydrophobic dyes, we have utilized anti-CD44 antibody (Boster, dilution ratio: 1:500) and secondary antibodies (Cy3-labeled Goat Anti-Rabbit IgG (H+L)) to label the HSPC membrane (represent in green). The CD44 expressed on the HSPC membrane but not on the liposomes, so this labeling has specificity for HSPC membranes. And the liposomes were labeled by DiD dye. As it showed that, green and red fluorescence signals have achieved co-localization both HSPC-Lipo alone (**Supplementary Fig. 2a**) and when combined with leukemia cells (**Supplementary Fig. 2b**).

Supplementary Figure 2. The colocalization of HSPC membrane and Liposomes. (a) HSPC membrane was labeled by anti-CD44 antibody and secondary antibodies (Cy3-labeled Goat Anti-Rabbit IgG (H+L)) (represent in green). Liposomes were labeled by DiD dye at $25 \mu\text{g mL}^{-1}$ (represent in red). Scale bar, $30 \mu\text{m}$. **(b)** Colocalization of HSPC-Lipo labeled by anti-CD44 antibody when combined with leukemia cells. Scale bar, $10 \mu\text{m}$. The Pearson's R value was calculated by Fuji image J (version 153 with java 8).

3. Since the level of *Itgb2* is quite different between cell membrane control and HSPC-Lipo sample (Fig. 2e), the presented western blot could benefit from a loading control or additional membrane protein which is important for homing process, such as *Cxcr4*.

Answer: Thank you for your suggestion. Based on your constructive comments, we have added the control GAPDH and CXCR4 protein in the western blotting experiment. The experimental results were shown below (**Supplementary Fig. 4c**).

Supplementary Figure 4c. Western blot verification of HSPC-Lipo.

4. The staining and gating of cell populations in supplemental figure 3h should be improved, the LSK and HSC plots/gates is not convincing.

Response: Thank you for your suggestion. Based on your constructive comments, we have conducted more rigorous gating and statistic analysis on the data and the results were shown in the figure below (**Supplementary Fig.6i**).

i The gating strategy for hematopoietic stem cells in flow cytometry.

Supplementary Figure 6. (i) The gating strategy of HSPC in flow cytometry analysis.

5. The paragraph regarding developmental hematopoiesis in the discussion (lines 287-294) should be removed since it is not pertinent to the studies presented in the manuscript.

Response: Thank you for your suggestion. Based on your constructive comment, we have removed this section of the discussion from the manuscript.

REVIEWER COMMENTS

Reviewer #1 (Remarks to the Author):

I would like to thank the authors for their inclusion of additional studies addressing reviewer concerns and much additional methodology. However some of these additional data (as shown in rebuttal) do not at this stage contain enough detail .

Most importantly;

Rebuttal Fig 2d-f.

If one assumes the same quantity of liposomes and a similar quantity of membrane coating were loaded in each mouse per group then the results shown indeed appear interesting. However as several details are not explained in legend text one cannot be sure.

Please include a sentence near start of figure legend, stating how much membrane was loaded per unit liposomes (per membrane type) and how much of each loaded per mouse (specifically confirming same amount of membrane prep/liposome) was administered to each mouse).

Providing a clear and detailed methodology on the source of HSPC used (not yet provided), how liposomes were created and coated (now mostly provided) is absolutely needed to ensure other researchers can replicate and confirm the authors experiments after publication. The authors have already responded with more details around this however providing the whole method from start to finish (including all reagent suppliers and catalog numbers) in detail is important. Eg. Source of HSPC and other cells used for membrane preps remained not defined. Are these primary cells? Are they cell lines? How much may they actually differ from actual HSPC? Does this make the manuscript title misleading? (possibly).

Not all 'protease inhibitor mixtures' are the same. Some very different. Providing Source and cat# essential for reproducibility. These details can go in Supplementary files if too long for text.

Manuscript title.

1) It is clear that 'hematopoietic stem cell' membrane were not actually used, instead ' this should be changed to 'hematopoietic progenitor cell membranes '. Even that is not sure, as these details (on HSPC source used) are not yet provided. It could possibly even be 'membranes from long-term cultured HSPC cell lines (originally derived from, or by introducing leukaemia associated genes)' . Some are even from a different mouse strain from the host mouse (compatibility mismatches).

2) Also the authors cannot state 'elimination of leukemia cells' in title. A substantial reduction is shown. However none of the experimental data shown in this manuscript demonstrate leukaemia 'elimination.' Please change title accordingly

I also note with concern that the 'HSC membrane preparation' in some figures (eg. second rebuttal figure) were generated using the '32D' cell line (as 'HSPC' source). My recollection is 32D is a murine neutrophil leukaemia cell line often used as an in vitro neutrophil model (following incubation with G-CSF). Indeed, 32D was generated in 1983 by Greenberger by infecting bone marrow cells from C3H/HeJ mice with Friend Murine Leukemia Virus). Thus 32D itself is a leukemic progenitor cell line, and is also not HLA matched with the C57BL/6 host mice used in these assays. These two differences may confound some of the results shown. Despite this concern, the use of 32D cells for generating "HSPC' membranes to coat liposomes, appear to have been mostly used for the studies in Fig 1 ? and Fig 3. The source of the HSPC's used to prepare membranes to coat liposomes in the other Figures / data shown in this publication does not appear to be specified. This detail is missing from methods section as well. An important omission. Please add these details in full. That is the exact source of cells for each liposome coating, where they derive from, how they were isolated or cultured, their mycoplasma status if cell lines and mouse strain they were derived from (to ensure no MHC mismatch with the C57BL/6 hosts used in these experiments.

Furthermore my concern is that in fact all the continuously cultured 'HSPC-like' cell lines I am aware of have been originally derived from leukemic patients/animals, or carry leukaemia like oncogenes/changes (such as generated by injecting leukaemia virus in donor mice to immortalise distinct leukocyte populations,), as do virtually all macrophage and neutrophil cell lines as well (eg. RAW264.7, commonly used as a murine 'macrophage' cell line). For this reason, please clearly specify source of origin for all cells used in these assays, along with source mouse background, MHC compatibility and mycoplasma testing results.

In summary, authors are encouraged to provide even more explicit detailed methods including details on the source of their HSPC for membrane preps, exactly how they were processed and loaded on liposomes, plus the controls taken to confirm same amount of membrane prep per liposome per mouse.

MINOR COMMENTS

(Fig 3 d,e)

In the (32D) 'HSPC' Cell Membrane and (32D) HSPC-lipo co-enriched protein preps, the authors show 74 proteins were identified by mass spec to be co-enriched. Of these they highlighted CD44 and ITGB2 as both well known as important in cell adhesion.

Are the 72 other co-enriched proteins also listed in the appendix for interested readers? Could authors also include list of the 16 cell surface proteins that do not transfer from HSPC membranes to liposome coating. Conversely would the authors like to discuss the apparent presence of 40 HSPC (32D) cell membrane proteins exclusively identified on HSPC-lipo, but in the original HSPC preps by mass spec.

Fig 3h. knockdown of CD44 data. Please also include your 'non-specific or scrambled control knockdown data as well, to give the readers confidence in the specificity of your CD44 knockdown data.

Figure 3o, 5c, 6f. etc. These are Kaplan-Meier survival curves. Please can the authors explain how the statistics and P-values were calculated. Are they Kaplan-Meier statistics or other? Specifically is this the analysis being referred to by the text 'Significant differences of P values were calculated by two-tailed, unpaired t-test.' If so then most of the figures appear to contain non-parametric data (ie. ttest is likely not the appropriate statistic. Best to consult with a statistician. Other possible tests include: (i) those to determine if data is normally distributed (ie. parametric) eg. Shapiro-Wilk . (ii) Potential non-parametric ttest alternatives include Kruskal-Wallis or Mann-Whitney.

Figure Y-axis label.

"Frequency of Tumor cells" is currently the term used. The label 'frequency of leukemia cells' may be more appropriate. Indeed this reviewer initially mistook these data (eg. Fig 4h etc) to actually be referring to tumor cell frequency, not leukemia cells.

'NEU" often means neuraminidase, while NEUT is more common for neutrophil.

Fig 2b. please add 'nm' to X-axis (as this refers to size of liposomes)

Reviewer #2 (Remarks to the Author):

I'm satisfied with the authors responses to my previous comments and suggestions.

Reviewer #3 (Remarks to the Author):

The authors addressed all my comments and concerns. The manuscript is greatly improved. The reviewer highly appreciates the new in vivo data presented in Fig. 6f.

Reviewer #4 (Remarks to the Author):

To the authors:

In the revised ms, the authors have provided the data comparing efficacy of targeting leukemia and homing of liposomes of different cell sources. In vivo imaging of bones over time convincingly demonstrated that HSPC-lipo exhibits statistically better efficacy than the other four cell sources.

To strengthen the authors' claim, in vivo treatment using MLL-AF9 AML model was a good addition to the CDX using two cell lines (Ka539 and C1498) originally presented at initial submission. If the authors performed histopathological examination using the treated mice, HE stained bone sections can be included as supplementary figures to show how different residual leukemia cells are present in specific locations such as endosteum or peri-vascular regions among different treatment options.

Point-by-point response to reviewers

REVIEWER COMMENTS

Reviewer #1 (Remarks to the Author):

I would like to thank the authors for their inclusion of additional studies addressing reviewer concerns and much additional methodology. However, some of these additional data (as shown in rebuttal) do not at this stage contain enough detail.

Most importantly;

Rebuttal Fig 2d-f.

If one assumes the same quantity of liposomes and a similar quantity of membrane coating were loaded in each mouse per group then the results shown indeed appear interesting. However as several details are not explained in legend text one cannot be sure.

Please include a sentence near start of figure legend, stating how much membrane was loaded per unit liposomes (per membrane type) and how much of each loaded per mouse (specifically confirming same amount of membrane prep/liposome) was administered to each mouse).

Response: Thanks for your helpful suggestions. In light of the current studies, the challenge in quantifying the quality of phospholipids in cell membrane has led researchers to adopt total cell membrane protein as a readily measurable proxy for membrane quantification in cell membrane-based biomimetic nanoparticle preparation studies¹⁻³. The total protein weight can be accurately determined using BCA assay kit. Our previous investigation revealed that when the weight ratio of liposomes to total cell membrane proteins was maintained at 1:0.5, an optimal fusion efficiency between the cell membranes and liposomes was achieved, the cell membrane-based biomimetic particles with a size approximating 150 nm, which is suitable for intravenous administration^{4,5}.

Regarding the exploration of *in vivo* targeting capabilities, all cell membrane-based biomimetic vesicles were administered intravenously to leukemic mice *via* tail vein injection at a dose equivalent to 400 $\mu\text{g kg}^{-1}$ of ICG content. With an encapsulation rate of $35.27\pm 0.15\%$ as detailed in **Supplementary Table 1**, each mouse received approximately 20 μg of liposomes and 10 μg of cell membrane (indicated by total cell membrane protein weight). These pertinent details have been integrated into the revised manuscript.

1. Krishnan, N. et al. A modular approach to enhancing cell membrane-coated nanoparticle functionality using genetic engineering. *Nature Nanotechnology* (2023). <https://doi.org/10.1038/s41565-023-01533-w>.

2. Zhang, F. et al. Nanoparticle-modified microrobots for in vivo antibiotic delivery to treat acute bacterial pneumonia. *Nature Materials* **21**, 1324-1332 (2022).
3. Zhou, J. et al. Nanotoxoid vaccination protects against opportunistic bacterial infections arising from immunodeficiency. *Science Advances* **8**, eabq5492 (2022).
4. Wu, H. et al. Hybrid stem cell-derived bioresponsive vesicles for effective inflamed blood-brain barrier targeting delivery. *Nano Today* **49**, 101800 (2023).
5. Wu, H. et al. Engineering Stem Cell Derived Biomimetic Vesicles for Versatility and Effective Targeted Delivery. *Advanced Functional Materials* **30**, 2006169 (2020).

Providing a clear and detailed methodology on the source of HSPC used (not yet provided), how liposomes were created and coated (now mostly provided) is absolutely needed to ensure other researchers can replicate and confirm the authors experiments after publication. The authors have already responded with more details around this however providing the whole method from start to finish (including all reagent suppliers and catalog numbers) in detail is important. Eg. Source of HSPC and other cells used for membrane preps remained not defined. Are these primary cells? Are they cell lines? How much may they actually differ from actual HSPC? Does this make the manuscript title misleading? (possibly).

Response: Thank you for the critical suggestions. We have now provided more details on the source and isolation of HSPCs in the Methods section of the manuscript. The detailed separation method is described below.

The isolation of hematopoietic progenitor cells was using the Mouse Hematopoietic Progenitor Cell Isolation Kit (MojoSort™, Cat. 480004, BioLegend). The detailed methodology followed these steps. Initially, mouse bone marrow cells were flushed out using a 3-mL syringe and centrifuged at 300 g for 5 minutes. Subsequently, 2 mL of red blood cell lysis buffer was introduced and the mixture was lysed at room temperature for 4 minutes. Following this, they were reconstituted in PBS and subjected to cell counting.

At 4°C, the above cells were centrifuged again at 300 g for an extended period of 10 minutes. Post-centrifugation, the cells were resuspended in a solution composed of precooled PBS+2% BSA+2 mM EDTA, with a ratio of 100 µL per every 10 million cells. Each batch of 10 million cells was then incubated with 10 µL of the Biotin-Antibody cocktail at 4 degrees for 15 minutes.

Following this step, 10 µL Streptavidin Nanobeads were added per 10 million cells and the mixture was again incubated at 4°C for another 15 minutes. After incubation, cells underwent washing with 4 mL of PBS under centrifugation conditions of 300 g for 10 min. The supernatant was carefully discarded, and the cells were resuspended in 2.5 mL PBS within a 5 mL polypropylene tube (12 x 75 mm). This tube was then placed in a magnetic field for a period of 5 minutes. This entire process was meticulously

repeated once more, culminating in a total of two separation cycles. Upon completion, the liquid was decanted, and the hematopoietic progenitor cells contained therein were collected into fresh, clean tubes for subsequent applications.

Preparation of HSPC-Lipo

As for the isolation of HSPC cells, mouse bone marrow cells were flushed with a 3 mL syringe, and the HSPC cells and cell membranes were collected according to the above method. The protein quantification of the obtained cell membranes was executed using the BCA Protein Assay Kit (Beyotime Biotechnology, Catalog No. P0010). The thin film method was adopted for liposome preparation. Dissolve 20 mg of lecithin in ethanol, spin dry ethanol under negative pressure at 35 °C and 120 rpm to prepare a thin film. The thin film was rehydrated with PBS buffer containing the cell membrane suspension, following which the phospholipid-to-cell membrane protein mass ratio was adjusted to 1:0.5 based on our previously established protocols^{15, 16}. Then, put the mixture of liposome and cell membrane into ultrasonic disrupter (SCIENTZ-IID, SCIENTZ, Chima) under ice bath and cell membrane and liposome fusion were performed by ultrasonic cavitation (60 w) for 6 min. Ultracentrifuge centrifugation at 4 °C, 150,000 g conditions for 3 h, remove the supernatant, the precipitate is the HSPC-Lipo.

The sources of the other different cells used in this study were provided more detailed in the revised manuscript. Specifically, in bone marrow and leukemia cell targeting assays, Neutrophils were from primary bone marrow derived neutrophils from C57BL/6 mice. Macrophages were from primary bone marrow derived cells induced macrophages from C57BL/6 mice. MSCs were derived from primary bone marrow isolated Mesenchymal stem cells of SD mice. Erythrocytes were derived from peripheral blood of C57BL/6 mice. The detailed isolation methods of the different cells and the source of the mouse have been marked in the revised manuscript.

Not all 'protease inhibitor mixtures' are the same. Some very different. Providing Source and cat# essential for reproducibility. These details can go in Supplementary files if too long for text.

Response: Thank you for the critical suggestions. According to your suggestion, we have added relevant ingredient description and cat# in the manuscript. The protease inhibitor we used was a generic protease inhibitor mixture used for cell or tissue protein extraction, etc., with the item No: (Cat#, P1005, Protease inhibitor cocktail for general use, 100X, Beyotime), and the ingredients were as follows: (200mM AEBSF, 30μM Aprotinin, 13mM Bestatin, 1.4mM E64 and 1mM Leupeptin in DMSO).

Manuscript title.

1) It is clear that 'hematopoietic stem cell' membrane were not actually used, instead

' this should be changed to 'hematopoietic progenitor cell membranes '. Even that is not sure, as these details (on HSPC source used) are not yet provided. It could possibly even be 'membranes from long-term cultured HSPC cell lines (originally derived from, or by introducing leukaemia associated genes) '. Some are even from a different mouse strain from the host mouse (compatibility mismatches).

2) Also the authors cannot state 'elimination of leukemia cells' in title. A substantial reduction is shown. However none of the experimental data shown in this manuscript demonstrate leukaemia 'elimination.' Please change title accordingly.

Response: Thank you for the critical suggestions. According to your two suggestions, we have revised the title of this article as follows to: **“Hematopoietic progenitor cell membranes-coated vesicles for targeted drug delivery to the bone marrow and inhibition of leukemogenesis.”**

I also note with concern that the 'HSC membrane preparation' in some figures (eg. second rebuttal figure) were generated using the '32D' cell line (as 'HSPC' source). My recollection is 32D is a murine neutrophil leukaemia cell line often used as an in vitro neutrophil model (following incubation with G-CSF). Indeed, 32D was generated in 1983 by Greenberger by infecting bone marrow cells from C3H/HeJ mice with Friend Murine Leukemia Virus). Thus 32D itself is a leukemic progenitor cell line, and is also not HLA matched with the C57BL/6 host mice used in these assays. These two differences may confound some of the results shown. Despite this concern, the use of 32D cells for generating "HSPC" membranes to coat liposomes, appear to have been mostly used for the studies in Fig 1? and Fig 3. The source of the HSPC's used to prepare membranes to coat liposomes in the other Figures / data shown in this publication does not appear to be specified. This detail is missing from methods section as well. An important omission. Please add these details in full. That is the exact source of cells for each liposome coating, where they derive from, how they were isolated or cultured, their mycoplasma status if cell lines and mouse strain they were derived from (to ensure no MHC mismatch with the C57BL/6 hosts used in these experiments.

Response: Thank you for the critical suggestions.

1) Reasons for using the 32D cell line in this study.

Thank you for your critical suggestions. First and foremost, we acknowledge that the 32D cell line is an "HSPC-like" cell line. In this article, we have meticulously outlined and justified our rationale for utilizing the 32D cell line with enhanced clarity.

The role of the 32D cells was confined to elucidating the underlying molecular mechanisms of targeting. Importantly, it should be highlighted that all subsequent *in vitro* toxicity assessments and *in vivo* animal experiments were conducted using primary isolated HSPC cells, ensuring biological relevance.

Furthermore, we recognize that primary HSPCs inherently present a higher degree of complexity compared to conventional cell lines when it comes to viral transduction efficiency and the establishment of stable knockdown cell lines. Thus, due to these practical challenges, we pragmatically employed the 32D cell line as a preliminary platform to validate the targeting mechanism. And in the subsequent *in vitro* toxicity and *in vivo* animal experiments, we used the primary isolated HSPC cells.

2) MHC compatibility of HSPCs.

Thank you for your advice. Regarding the MHC compatibility, we ensured consistency by utilizing primary HSPCs isolated from C57BL/6 mice in our subsequent *in vivo* therapeutic experiments. This choice aligned with the mouse strain employed in our leukemia mouse model, thus ensuring homogeneity in the cell source.

Moreover, we conducted a meticulous bioinformatics analysis to investigate the expression patterns of MHC molecules across the murine hematopoietic system and various leukocyte subsets. In this analysis, we focused on key MHC components such as H2-Ab1, H2-Aa, H2-Eb1, and H2-Eb2 within the context of mouse HSPCs as well as different leukocyte types. The comprehensive findings derived from this analysis are presented graphically in **Figure R1**, offering profound insights into the distribution and relevance of these MHC molecules in the targeted cell populations

Figure R1. The major MHC molecules such as H2-Ab1, H2-Aa, H2-Eb1 and H2-Eb2 in mouse HSPCs and different leukocyte types. (The data were analyzed online by <http://www.bloodspot.eu>)

3) Results of mycoplasma detection after continuous culture of 32D cells.

Thank you for your critical suggestions. We performed mycoplasma assays on the 32D cell line used in our experiments. The test results were shown in the figure below. The results showed that no mycoplasma contamination was observed in our cells. For positive controls, MycoBlue Mycoplasma detector (Cat. #101, Vazyme) was used and diluted 30-fold with ddH₂O before use. Sterile deionized water was used as a negative control (Cat. E607017-0100, Sangon). Detailed procedures for mycoplasma detection were presented in the Methods section of the manuscript.

Supplementary Fig. 5c The mycoplasma detection of 32D cells. Specific bands appearing at 280 bp were defined as mycoplasma contamination cells.

4) Description of the membrane origin of HSPCs in different experiments.

Thank you for your critical suggestions. Regarding the membrane origin of HSPCs in different experiments, we have annotated in more detail in the experimental methods. To be specific, we used 32D cells only in experiments demonstrating the molecular mechanism of targeting (**Fig.3h and n**). In the subsequent *in vitro* toxicity and *in vivo* animal treatment experiments, we all used primary HSPCs extracted from C57BL/6 mice.

Furthermore my concern is that in fact all the continuously cultured 'HSPC-like' cell lines I am aware of have been originally derived from leukemic patients/animals, or carry leukaemia like oncogenes/changes (such as generated by injecting leukaemia virus in donor mice to immortalise distinct leukocyte populations.), as do virtually all macrophage and neutrophil cell lines as well (eg. RAW264.7, commonly used as a murine 'macrophage' cell line). For this reason, please clearly specify source of origin for all cells used in these assays, along with source mouse background, MHC compatibility and mycoplasma testing results.

In summary, authors are encouraged to provide even more explicit detailed methods including details on the source of their HSPC for membrane preps, exactly how they were processed and loaded on liposomes, plus the controls taken to confirm same amount of membrane prep per liposome per mouse.

Response: Thank you for the critical suggestions. The sources of the different cells used in this study and the more detailed steps of how the membrane-coated liposomes were prepared were more detailed provided in the revised manuscript. Specifically, in bone marrow and leukemia cell targeting assays, Neutrophils were from primary bone marrow derived neutrophils from C57BL/6 mice. Macrophages were from primary bone marrow derived cells induced macrophages from C57BL/6 mice. MSCs were derived from primary bone marrow isolated Mesenchymal stem cells of SD mice. Erythrocytes were derived from peripheral blood of C57BL/6 mice. The detailed isolation methods of the different cells and the source of the mouse have been marked in the revised manuscript.

MINOR COMMENTS

(Fig 3 d,e)

In the (32D) 'HSPC' Cell Membrane and (32D) HSPC-lipo co-enriched protein preps, the authors show 74 proteins were identified by mass spec to be co-enriched. Of these they highlighted CD44 and ITGB2 as both well known as important in cell adhesion. Are the 72 other co-enriched proteins also listed in the appendix for interested readers? Could authors also include list of the 16 cell surface proteins that do not transfer from HSPC membranes to liposome coating. Conversely would the authors like to discuss the apparent presence of 40 HSPC (32D) cell membrane proteins exclusively identified on HSPC-lipo, but in the original HSPC preps by mass spec.

Response: Thank you for the critical suggestions. We used primary isolated HSPCs cells from the bone marrow of C57BL/6 mice for mass spectrometry experiments. According to your suggestion, we performed further KEGG enrichment analysis on the MS data according to different groups (**Supplementary Fig. 4d-f**). And the detailed list of proteins were presented in the **Supplementary Table 4-6**. We have also discussed in the revised manuscript as follows: “To verify the mechanism by which HSPC-Lipo targets leukemia cells, we further analyzed the mass spectrometry data and found that the adhesion-related proteins were present in both the cell membrane and HSPC-Lipo (**Fig. 3c, d and Supplementary Fig. 4a, b**). We further performed KEGG pathway enrichment analysis among different groups and found that adhesion-related pathways, such as focal adhesion, were both found in HSPC-Lipo and co-enriched groups (**Supplementary Fig. 4d-f**). The detailed protein lists were presented in **Supplementary Table 4-6**. By further analyzing the proteins enriched by the cell membrane and HSPC-Lipo vesicles, we identified ITGB2 (CD18), the classic proteins involved in cell adhesion. We further verified the mass spectrometry data by performing western blot to demonstrate that the adhesion molecules ITGB2 was expressed on the cell membrane and HSPC-Lipo (**Fig. 3e**)”

Supplementary Figure 4. d-f KEGG pathway enrichment analysis of HSPC CM (**d**), HSPC-Lipo (**e**) and overlap of HSPC CM and HSPC-Lipo (**f**).

Fig 3h. knockdown of CD44 data. Please also include your 'non-specific or scrambled control knockdown data as well, to give the readers confidence in the specificity of your CD44 knockdown data.

Response: Thank you for the critical suggestions. At your suggestion, we added scrambled control as a control in the WB experiments. The relevant results were shown below, and we have also made revisions in the manuscript.

Supplementary Figure 5. Verify the knockdown efficiency of CD44 and ITGB2 in mouse HSPC cells (32D) using shRNA method. (a) Knockdown efficiency of CD44. **(b)** Knockdown efficiency of ITGB2. The ECL imaging and figures were collected by the Bio-Rad GelDoc Go with automatic exposure time.

Figure 3o, 5c, 6f. etc. These are Kaplan-Meier survival curves. Please can the authors explain how the statistics and P-values were calculated. Are they Kaplan-Meier statistics or other? Specifically is this the analysis being referred to by the text 'Significant differences of P values were calculated by two-tailed, unpaired t-test.' If so then most of the figures appear to contain non-parametric data (ie. ttest is likely not the appropriate statistic. Best to consult with a statistician. Other possible tests include: (i) those to determine if data is normally distributed (ie. parametric) eg. Shapiro-Wilk. (ii) Potential non-parametric ttest alternatives include Kruskal-Wallis or Mann-Whitney.

Response: Thank you for the critical suggestions. Based on your suggestion, we have made a more rigorous explanation of how the statistics and P-values of survival curves were calculated in this study. In this study, the comparison of survival curves and P-values of different groups were calculated using the Log-rank (Mantel-Cox) test. And GraphPad Prism 8.0.2 software was used to analyze the survival rate in this study. We have made corresponding revisions and descriptions in the figure legend in **Fig. 5b, Fig. 6f, and Supplementary Fig. 9c.**

Figure Y-axis label.

“Frequency of Tumor cells” is currently the term used. The label ‘frequency of leukemia cells’ may be more appropriate. Indeed this reviewer initially mistook these data (eg. Fig 4h etc) to actually be referring to tumor cell frequency, not leukemia cells.

Response: Thank you for the critical suggestions. According to your suggestion, we have revised in the figures manuscript accordingly. In **Fig. 5d,h,i and Fig. 6c,d and Supplementary Fig. 9e-g**, we revised “Frequency of Tumor cells” to the currently “frequency of leukemia cells.”

‘NEU’ often means neuraminidase, while NEUT is more common for neutrophil.

Response: Thank you for the critical suggestions. According to your suggestion, we have revised the figures in **Fig. 2d,e,f and Fig. 3j** in revised manuscript. The revised figure were shown below.

Fig. 2 d-e Bone marrow homing results of ICG fluorescent labeled different leukocyte membrane-coated liposomes in leukemic mice (Ka539 model). Mice were subjected to *in vivo* imaging detection at different time intervals (2, 6, 12, 24 hours) after tail vein injection of different liposomes. $n=3$ biologically independent samples in each group. **e** The bone marrow homing ability of different leukocyte membrane-coated liposomes in the tibia and femur of mice. Mice were sacrificed at 24 h after tail vein injection of different liposomes, the tibia and femur were taken for *in vivo* imaging detection. $n=3$ biologically independent samples in each group. **f** Quantitative analysis of different liposomes in mouse tibia and femur. Data were presented as mean \pm s.d. ($n=3$ biologically independent samples).

Fig. 3j Flow cytometry detection of leukemic cell targeting ability of different leukocyte membrane-coated liposomes. Leukemia cells were harvested for detection at 0.5 hours after incubation with different liposomes. Data were presented as mean \pm s.d. ($n=3$ biologically independent samples).

Fig 2b. please add 'nm' to X-axis (as this refers to size of liposomes)

Response: Thank you for the critical suggestions. According to your suggestion, we have revised the figure as follows.

Fig. 2b Average particle size distribution of HSPC-Lipo. Measured by Dynamic Light Scattering.

Reviewer #2 (Remarks to the Author):

I'm satisfied with the authors responses to my previous comments and suggestions.

Response: We thank the reviewer for his/her positive general remarks on our study.

Reviewer #3 (Remarks to the Author):

The authors addressed all my comments and concerns. The manuscript is greatly improved. The reviewer highly appreciates the new in vivo data presented in Fig. 6f.

Response: We appreciate the positive and insightful comments of reviewer.

Reviewer #4 (Remarks to the Author):

To the authors:

In the revised ms, the authors have provided the data comparing efficacy of targeting

leukemia and homing of liposomes of different cell sources. In vivo imaging of bones over time convincingly demonstrated that HSPC-lipo exhibits statistically better efficacy than the other four cell sources.

To strengthen the authors' claim, in vivo treatment using MLL-AF9 AML model was a good addition to the CDX using two cell lines (Ka539 and C1498) originally presented at initial submission. If the authors performed histopathological examination using the treated mice, HE stained bone sections can be included as supplementary figures to show how different residual leukemia cells are present in specific locations such as endosteum or peri-vascular regions among different treatment options.

Response: We appreciate the positive and insightful comments of reviewer. The revised manuscript is supplemented with further pertinent experimental data. We address these concerns below and also made corresponding changes in the revised. According to your suggestion, we have added the results of H&E stained sections of mouse bone marrow in MLL-AF9 AML model after treatment with different drugs in the manuscript (**Supplementary Fig. 8**). From the staining results, the group of Ara-C@HSPC-Lipo showed a good inhibitory effect of leukemogenesis.

Supplementary Figure 8. H&E staining of BM in MLL-AF9 leukemia model. The infiltration of leukemia cells in BM after different treatment. The purple staining indicates the leukemia cell infiltration.

REVIEWERS' COMMENTS

Reviewer #1 (Remarks to the Author):

I would like to thank the authors again for their constructive approach in responding to reviewers question. The readability, data presentation, explanations and methods section are now greatly improved.

Remaining comments are mostly minor. Throughout text I strongly encourage the authors to only write what their data actually shows without use of attractive but scientifically inaccurate words such as 'eliminates leukemia' (when it clearly doesn't as all their mice died of leukemia despite treatment), or 'specifically targets' when the data indicates a few fold enrichment in targeting occurs (ie an increase in specificity only) is shown. Another proof-read of the authors own text would be much appreciated and would save considerable reviewers time. Some of the comments below have already been mentioned in by this reviewer. This time I specifically request they are made throughout the text.

1. Through authors hard work authors, the manuscript title and figures now accurately reflect the data shown. However this reviewer noticed that some of the wording in the text describing same data could also be more precise. This reviewer would like to suggest that the authors actually look at their own results data carefully before writing manuscript title, abstract and text. Each sentence should accurately reflect the data shown. The most important examples to change are:

Numerous comments in the text mention that the HSPC coated vesicles exhibited 'specific' affinity to leukemia cells, and are 'specific' to ICAM-1 on leukemia cells. Based on the data shown, neither of these are statements are precisely correct.

Suggestion is for the authors to replace such inaccurate wording with either the actual data they have generated (preferred approach) or by alternative wording (eg. In abstract where word limits apply).

..for example in the ABSTRACT:

(i) please modify the sentence from 'biomimetic vesicles exhibited specific affinity to leukemia cells'.....with something like a slightly more accurate ..."superior affinity to the leukemia cells" Instead.

Alternatively the same data in Fig 3o (describing role of ICAM-1 receptor ITGB2 on membrane-coated liposomes in mediating leukemia cell attachment in vitro) could be even more accurately described as

' HSPC lipo increased affinity of liposomes for leukemia (Ka539) cell line ~2.7-fold in vitro (from a baseline of ~13% when ITGB2 knocked down, up to ~35%). Maybe even using term ..."showed 2.7-fold higher specificity for..." would be a shorter and still more accurate way to explain the actual data.

This reviewer has very similar comments around the authors reported increased anti-leukemia efficiency of AraC containing HSPC-liposomes, compared to AraC alone liposomes in mice. (results/discussion sections referring to Fig 5e (residual BM CD34+ CD16/32 cells following treatment). The actual data (Fig 5e) appear to suggest 40% LSC in BM of non-treated leukemia mice, can be reduced to ~25% LSC in BM following injection of AraC alone, down to ~10% with AraC-liposomes, and maybe 2% with HSPC-AraC-liposomes. Thus suggesting the addition of HSPC membrane to AraC-liposomes potentially promotes a ~5-fold greater targeting specificity for LSC in the BM, and maybe 1.4-fold greater specificity in blood (data in Fig 5c, from ~40% GFP+ leukaemia cells in blood of AraC-liposome mice down to maybe 25% in HSPC-coated AraC-liposome treated mice). The data are good, however the term 'elimination' of leukemia as mentioned in abstract and suggested several other various places in text should be clarified.

Re. word elimination. All mice died of their leukemia regardless of which treatment group they were (Fig 6f) – Thus the leukemia was clearly not 'eradicated'. Please change such words throughout manuscript.

Accordingly please modify the words used in:

ABSTRACT: line 48, Specific affinity line 51, superior line 58.

INTRO: line specific targeting 124, superior biosafety 126

RESULTS; (there are many, please correct) including titles eg. Line 185 'specifically targeted' which in this specific case could be replaced by 'increased targeting' (refers to data in Fig 2f (HSPC-liposome targeting, maybe 5-fold), and in Fig 3i (role of CD44 on HSPC-liposome in BM targeting, maybe 1.4-fold greater specificity, ie. HPSC-lipo from CD44 ko gives BM readout 5, while HPSC-lipo from WT is 8 radiant efficiency).

*also note possible mistake in Fig 3i legend text – this figure appears to refer to CD44 targeting while legend (line 965) mentions 'ICAM-1 analysis'

Line 363, referring to CD44-HA BM homing. Change the word from 'CD44-HA axis is 'indispensable to' to more accurate 'is involved in' - unless the authors are referring to data other than the ~1.4-fold greater BM homing shown in Fig 3i.

(ii) Similar comments around the term “we verified ‘superior’ biosafety of the biomimetic vesicles” in abstract, which refers to results data (text line 328, Fig 8) . I agree the authors indeed checked the HSPC-coated liposomes showed no toxicity in mice in terms of no change in blood hemogram indices, weight loss, and H&E organ staining (Fig 8). However these readouts remained almost identical for all three agents tested for biosafety (PBS, liposome and HSPC liposomes). Thus Fig 8 data confirms safety, but gives no evidence of ‘superior biosafety. Please change sentence accordingly (ie’ the HSPC liposomes are safe in mice’) or completely remove sentence from abstract.

2. In all places where 32D cells used in text referred to please refer to them as a ‘progenitor cell line (CD32D)’ or ‘HPC line (32D)’ instead of current use of term HSPC (32D)’. Reason is 32D are a cell line and it is confusing for reader for them to continually be refer to as HPC or HSPC in manuscript text, when real KIT+ BM HPSC -coated liposomes are also used. Note Cell lines are very different to primary cells.

This should be corrected throughout all text and figure legends.

I also remind the authors that “32D were generated in C3H/Hej mouse strain (that is are not MHC matched with the C57BL/6 used in this manuscript) and were immortalized by murine leukemia virus infection. “This information should specifically be added to the METHODS section when 32D are first mentioned which presumably should be METHODS – CELL CULTURE section (line 398) - but appears they are not currently mentioned.

Specifically please Add this sentence to METHODS – Cell Culture section ...’ 32D a murine hematopoietic progenitor cell line, originally generated in C3H/Hej mouse strain and immortalized by murine leukemia virus were cultured in media... were used as source of HPC for in Figures 2,3,7 (if correct).

3. METHODS. Line 459. The authors mention ‘film method’ . Could they please also include a reference at end of this sentence that describes the ‘film method’.

4. In response to a prior question, the authors confirmed that they repeated the studies three times. As the studies in Fig 5 and Fig 6 (efficacy in reducing leukaemia burden in mice) may attract much reader attention, could the authors include the data of their repeat independent experiments (conducted using a different batch of HSPC-AraC-liposomes on different batch of mice) as an additional supplementary

figure? This would aid interested readers in understanding the variation and reproducibility within the whole method in boosting long-term survival of leukemic mice.

5. Figures containing flow cytometry dot plots. Minor edits. Fig 5c,e,f. Fig6 b.. Please ensure X and Y axis marks/numbers are visible on all flow cytometry dot plots.

Point-by-point response to reviewers

REVIEWER COMMENTS

Reviewer #1 (Remarks to the Author):

I would like to thank the authors again for their constructive approach in responding to reviewers question. The readability, data presentation, explanations and methods section are now greatly improved.

Remaining comments are mostly minor. Throughout text I strongly encourage the authors to only write what their data actually shows without use of attractive but scientifically inaccurate words such as 'eliminates leukemia' (when it clearly doesn't as all their mice died of leukemia despite treatment), or 'specifically targets' when the data indicates a few fold enrichment in targeting occurs (ie an increase in specificity only) is shown. Another proof-read of the authors own text would be much appreciated and would save considerable reviewers time. Some of the comments below have already been mentioned in by this reviewer. This time I specifically request they are made throughout the text.

Response: Thank you for the critical suggestions. Based on your suggestions, we have revised the inappropriate words in the full text to make the description of the article more accurate.

1. Through authors hard work authors, the manuscript title and figures now accurately reflect the data shown. However this reviewer noticed that some of the wording in the text describing same data could also be more precise. This reviewer would like to suggest that the authors actually look at their own results data carefully before writing manuscript title, abstract and text. Each sentence should accurately reflect the data shown. The most important examples to change are::

Numerous comments in the text mention that the HSPC coated vesicles exhibited 'specific' affinity to leukemia cells, and are 'specific' to ICAM-1 on leukemia cells. Based on the data shown, neither of these are statements are precisely correct.

Suggestion is for the authors to replace such inaccurate wording with either the actual data they have generated (preferred approach) or by alternative wording (eg. In abstract where word limits apply).

..for example in the ABSTRACT:

(i) please modify the sentence from 'biomimetic vesicles exhibited specific affinity to leukemia cells'with something like a slightly more accurate ... "superior affinity to the leukemia cells" Instead.

Response: Thanks for your helpful suggestions. We have revised “*biomimetic vesicles exhibited specific affinity to leukemia cells*” into “*superior affinity to the leukemia cells*” in the abstract.

Alternatively the same data in Fig 3o (describing role of ICAM-1 receptor ITGB2 on membrane-coated liposomes in mediating leukemia cell attachment in vitro) could be even more accurately described as ‘HSPC lipo increased affinity of liposomes for leukemia (Ka539) cell line ~2.7-fold in vitro (from a baseline of ~13% when ITGB2 knocked down, up to ~35%). Maybe even using term ... ”showed 2.7-fold higher specificity for...” would be a shorter and still more accurate way to explain the actual data.

Response: Thanks for your helpful suggestions. We have added the corresponding descriptions in the manuscript as follow: “*HSPC-Lipo increased affinity of liposomes for leukemia (Ka539) cell line ~2.7-fold in vitro (from a baseline of ~13% when ITGB2 knocked down, up to ~35%)*”.

This reviewer has very similar comments around the authors reported increased anti-leukemia efficiency of AraC containing HSPC-liposomes, compared to AraC alone liposomes in mice. (results/discussion sections referring to Fig 5e (residual BM CD34+ CD16/32 cells following treatment). The actual data (Fig 5e) appear to suggest 40% LSC in BM of non-treated leukemia mice, can be reduced to ~25% LSC in BM following injection of AraC alone, down to ~10% with AraC-liposomes, and maybe 2% with HSPC-AraC-liposomes. Thus suggesting the addition of HSPC membrane to AraC-liposomes potentially promotes a ~5-fold greater targeting specificity for LSC in the BM, and maybe 1.4-fold greater specificity in blood (data in Fig 5c, from ~40% GFP+ leukaemia cells in blood of AraC-liposome mice down to maybe 25% in HSPC-coated AraC-liposome treated mice). The data are good, however the term ‘elimination’ of leukemia as mentioned in abstract and suggested several other various places in text should be clarified.

Re. word elimination. All mice died of their leukemia regardless of which treatment group they were (Fig 6f) – Thus the leukemia was clearly not ‘eradicated’. Please change such words throughout manuscript.

Response: Thanks for your helpful suggestions.

(1) Line 287: We have added the corresponding descriptions in the manuscript as follow: “*The above results suggest the addition of HSPC membrane to Ara-C loading liposomes potentially promotes a ~5-fold greater targeting specificity for LSC in the BM, and approximately 1.4-fold greater specificity in peripheral blood for leukemic cells.*”

(2) Line 48: We have removed the relevant words “*elimination of leukemia*”.

(3) Line 50: We have revised the word “*eliminated*” into “*decreased the number of*”.

Accordingly please modify the words used in:

ABSTRACT: line 48, Specific affinity line 51, superior line 58.

INTRO: line specific targeting 124, superior biosafety 126

RESULTS; (there are many, please correct) including titles eg. Line 185 ‘specifically targeted’ which in this specific case could be replaced by ‘increased targeting’ (refers to data in Fig 2f (HSPC-liposome targeting, maybe 5-fold), and in Fig 3i (role of CD44 on HSPC-liposome in BM targeting, maybe 1.4-fold greater specificity, ie. HPSC-lipo from CD44 ko gives BM readout 5, while HPSC-lipo from WT is 8 radiant efficiency)).

Response: Thanks for your helpful suggestions.

(1) Line 48: We have removed the relevant words “*elimination of leukemia*”.

(2) Line 46: We have revised “*specific affinity to leukemia cells*” into “*superior affinity to the leukemia cells*”.

(3) Line 52: We have revised “*Finally, we verified the superior biosafety of the biomimetic vesicles*” into “*Finally, we verify that HSPC liposomes are safe in mice*”.

(4) Line 124: We have revised “*specific targeting*” into “*the targeting*”.

(5) Line 125: We have revised “*Finally, we verified the superior biosafety of the biomimetic vesicles.*” into “*Finally, we verify that HSPC liposomes are safe in mice*”.

(6) Line 183: We have revised “*specifically targeted*” into “*facilitates targeting*”.

**also note possible mistake in Fig 3i legend text – this figure appears to refer to CD44 targeting while legend (line 965) mentions ‘ICAM-1 analysis’*

Response: Thank you for the critical suggestions.

Line 1075: We have revised the figure legend as “*i Quantitative analysis of fluorescence images of CD44 knockdown HSPC-Lipo. Data were presented as mean ± s.d. (n=3 biologically independent mice)*”.

Line 363, referring to CD44-HA BM homing. Change the word from ‘CD44-HA axis is indispensable to’ to more accurate ‘is involved in’ - unless the authors are referring to data other than the ~1.4-fold greater BM homing shown in Fig 3i.

Response: Thank you for the critical suggestions.

Line 366: We have revised “*was indispensable in*” into “*is involved in*”.

(ii) Similar comments around the term “we verified ‘superior’ biosafety of the biomimetic vesicles” in abstract, which refers to results data (text line 328, Fig 8) . I agree the authors indeed checked the HSPC-coated liposomes showed no toxicity in mice in terms of no change in blood hemogram indices, weight loss, and H&E organ staining (Fig 8). However these readouts remained almost identical for all three agents tested for biosafety (PBS, liposome and HSPC liposomes). Thus Fig 8 data confirms safety, but gives no evidence of ‘superior biosafety. Please change sentence accordingly (ie’ the HSPC liposomes are safe in mice’) or completely remove sentence from abstract.

Response: Thank you for the critical suggestions.

(1) Line 52: We have revised “*Finally, we verified the superior biosafety of the biomimetic vesicles*” into “*Finally, we verify that HSPC liposomes are safe in mice*”.

(2) Line 125: We have revised “*Finally, we verified the superior biosafety of the biomimetic vesicles.*” into “*Finally, we verify that HSPC liposomes are safe in mice*”.

(3) Line 343: We have revised “*Taken together, these results show that the HSPC-lipo vesicles have superior biological safety.*” into “*Taken together, these results show that the HSPC liposomes are safe in mice*”.

2. In all places where 32D cells used in text referred to please refer to them as a ‘progenitor cell line (CD32D)’ or ‘HPC line (32D)’ instead of current use of term HSPC (32D)’. Reason is 32D are a cell line and it is confusing for reader for them to continually be refer to as HPC or HSPC in manuscript text, when real KIT+ BM HPSC -coated liposomes are also used. Note Cell lines are very different to primary cells.

This should be corrected throughout all text and figure legends.

Response: Thank you for the critical suggestions. We have revised throughout the manuscript and figure legend as follows based on your suggestions.

(1) Line 223: We have revised “*HSPC cells (32D)*” into “*progenitor cell line (32D)*”.

(2) Line 471: We have revised “*HSPCs cells (32D)*” into “*progenitor cell line (32D)*”.

(3) Line 671: We have revised “*32D cells*” into “*progenitor cell line (32D) cells*”.

(4) Line 1074: We have revised “*cell membrane derived from 32D cells*” into “*cell membrane derived from progenitor cell line 32D cells*”.

(5) Line 1081: We have revised “*mouse hematopoietic stem progenitor cells (32D)*” into “*mouse progenitor cell line (32D)*”.

(6) Line 1085: We have revised “cell membrane derived from 32D cells” into “mouse progenitor cell line (32D)”.

I also remind the authors that “32D were generated in C3H/Hej mouse strain (that is are not MHC matched with the C57BL/6 used in this manuscript) and were immortalized by murine leukemia virus infection. “This information should specifically be added to the METHODS section when 32D are first mentioned which presumably should be METHODS – CELL CULTURE section (line 398) - but appears they are not currently mentioned.

Specifically please Add this sentence to METHODS – Cell Culture section ...’ 32D a murine hematopoietic progenitor cell line, originally generated in C3H/Hej mouse strain and immortalized by murine leukemia virus were cultured in media... were used as source of HPC for in Figures 2,3,7 (if correct).

Response: Thank you for the critical suggestions. We have added the corresponding descriptions in METHODS–CELL CULTURE section in the manuscript as follow: “32D a murine hematopoietic progenitor cell line, originally generated in C3H/Hej mouse strain and immortalized by murine leukemia virus were cultured in 90% 1640 +10% FBS +10 ng/ml mIL-3 media, were used as source of HPC for knockdown experiments in Fig. 3 and Supplementary Fig. 5”.

3. METHODS. Line 459. The authors mention ‘film method’. Could they please also include a reference at end of this sentence that describes the ‘film method’.

Response: Thank you for the critical suggestions. We have added the corresponding references in the methods section of the manuscript in Line 468 as follows.

37. Zhang H. Thin-Film Hydration Followed by Extrusion Method for Liposome Preparation. *Liposomes* **1522**, pp 17–22 (2016).

4. In response to a prior question, the authors confirmed that they repeated the studies three times. As the studies in Fig 5 and Fig 6 (efficacy in reducing leukaemia burden in mice) may attract much reader attention, could the authors include the data of their repeat independent experiments (conducted using a different batch of HSPC-AraC-liposomes on different batch of mice) as an additional supplementary figure? This would aid interested readers in understanding the variation and reproducibility within the whole method in boosting long-term survival of leukemic mice.

Response: Thanks for your comments and suggestions. Regarding the concerns about the batch effect of the biomimetic liposome and the repeatability of animal experiments, the results from three animal treatment trials across two types of

leukemia mouse models presented in our manuscript have corroborated the effectiveness and repeatability of the HSPC biomimetic liposome therapy to a certain extent. Specifically, we conducted a total of three *in vivo* therapeutic efficacy experiments in mice using retrovirus-mediated leukemia models (MLL-AF9 model) and CDX models of B-cell acute lymphoblastic leukemia (Ka539 cells), testing both low and high dosage regimens of chemotherapeutic agents, which demonstrated the consistent therapeutic efficacy in mice.

Regarding the issue of independent repetition of *in vivo* and *in vitro* experiments in this study, we have made an explanation in the corresponding legends. Most of the experiments in this study were repeated three times independently, and the animal experiment was repeated once, but a total of three *in vivo* experiments were performed. In addition, the number of mice in each group was marked in the corresponding figure legends and Methods section in the manuscript.

5. Figures containing flow cytometry dot plots. Minor edits. Fig 5c,e,f. Fig 6 b.. Please ensure X and Y axis marks/numbers are visible on all flow cytometry dot plots.

Response: Thanks for your critical suggestion. We have modified the marks/numbers of the X-axis and Y-axis can be seen.

Fig. 5 | The anti-leukemic effect of Ara-C@HSPC-Lipo in mLL-AF9 leukemia mouse model. c Representative flow cytometry plots of leukemia cells (GFP positive

cells) in bone marrow. Leukemic mice were euthanized on the day 21 to collect bone marrow cells for flow cytometry analysis. **d** Quantitative analysis of leukemia cells in bone marrow. Data were presented as mean \pm s.d. (n=5 biologically independent mice). **e** Representative flow cytometry plots of leukemia stem cells (CD34+CD16/32+) in bone marrow after different treatments. **f** Quantitative analysis of leukemia stem cells in BM. Data were presented as mean \pm s.d. (n=5 biologically independent mice). **g** Representative flow cytometry plots of leukemic cells (GFP positive cells) in spleen. **h** Quantitative analysis of leukemia cells in spleen. Data were presented as mean \pm s.d. (n=5 biologically independent mice).

Fig. 6 | The anti-leukemic effect of Ara-C@HSPC-Lipo in Ka539 leukemia mouse model. **b** Representative flow cytometry plots of leukemia cells (GFP positive cells) in bone marrow. Leukemic mice were euthanized at day 28 to collect bone marrow cells for analysis. **c** Quantitative analysis of leukemia cells in bone marrow. Data were presented as mean \pm s.d. (n=5 biologically independent mice).